# EMMA-500: Enhancing Massively Multilingual Adaptation of Large Language Models

## Abstract

In this work, we introduce **EMMA-500**, a large-scale multilingual language model continue-trained on texts across 546 languages designed for enhanced multilingual performance, with a focus on improving language coverage for low-resource languages. To facilitate continual pre-training, we compile **the MaLA corpus**, a comprehensive multilingual dataset and enrich it with curated datasets across diverse domains. Leveraging this corpus, we conduct extensive continual pre-training of the Llama 2 7B model, resulting in EMMA-500, which demonstrates robust performance across a wide collection of benchmarks, including a comprehensive set of multilingual tasks and **PolyWrite**, an open-ended generation benchmark developed in this study. Our results highlight the effectiveness of continual pre-training in expanding large language models' language capacity, particularly for underrepresented languages, demonstrating significant gains in cross-lingual transfer, task generalization, and language adaptability.

## 1 Introduction

Multilingual language models (MLMs) are designed to process and generate text in multiple languages. These models have evolved rapidly over the past decade, fueled by advances in deep learning, e.g., Transformer networks (Vaswani et al., 2017), pre-training techniques, and the availability of large-scale multilingual corpora such as mC4 (Raffel et al., 2020) and ROOTS (Laurençon et al., 2022). The development of models like BERT (Devlin et al., 2019), GPT, and T5 (Raffel et al., 2020) opened the door for multilingual counterparts such as mBERT, XLM-R (Conneau et al., 2020), mGPT (Shliazhko et al., 2022), and mT5 (Xue et al., 2021). These models were trained on massive multilingual corpora, allowing text in dozens of languages to be processed with the same set of model weights. Multilingual language models have shown impressive performance across various tasks like text classification and machine translation by leveraging cross-lingual transfer from high-resource languages such as English and Chinese. However, many low-resource languages, with limited available data, remain underrepresented. While large corpora are abundant for high-resource languages like English, French, and Spanish, languages like Xhosa and Inuktitut suffer from scarce or fragmented data, leading to imbalanced training and models that prioritize high-resource languages.

Recent studies adopt continual pre-training to enhance the language coverage of large language models on low-resource languages. For example, Glot500 (Imani et al., 2023) and MaLA-500 (Lin et al., 2024) use continual pre-training and vocabulary extension using XLM-R and LLaMA, respectively, on the Glot500-c corpus covering 534 languages. xLLMs-100 (Lai et al., 2024) proceeds to multilingual instruction fine-tuning to improve the multilingual performance of LLaMA and BLOOM models on 100 languages, and Aya model (Üstün et al., 2024) applies continual training to the mT5 model (Xue et al., 2021) using their constructed instruction dataset. LLaMAX (Lu et al., 2024) pushes the envelope by focusing on translation tasks via continual pre-training of LLaMA in over 100 languages.

As the field of MLMs evolves, the role of data becomes increasingly critical in enhancing the performance of the models, particularly when it comes to low-resource languages. To address this need for more and better data, we extend existing work, such as MaLA-500, by expanding the corpus for continual pre-training, coupled with large-scale training methods. We emphasize the creation of a massively multilingual corpus that not only increases the quantity of data but also diversifies the types of texts (e.g., code, books, scientific papers, and instructions). This ensures better language adaptation and broader language coverage, thus improving the representation and performance of

multilingual language models, especially for underrepresented languages, ultimately creating more inclusive and versatile language models that cater to a broader linguistic diversity. Our contribution is summarized as follows: (1) We compile a massively multilingual corpus named MaLA to facilitate continual training of large language models for enhanced language adaptation across a wide range of linguistic contexts. (2) We extend the MaLA corpus by integrating multiple curated datasets, creating a comprehensive and diverse data mix specifically for continual pre-training. (3) We perform continual pre-training with the Llama 2 7B model[1] on the multilingual corpus with 546 languages, resulting in the new EMMA-500 model. This model is rigorously evaluated on a diverse set of tasks and benchmarks, including PolyWrite, a novel open-ended generation benchmark developed as part of this work.

**The MaLA Corpus**    Our multilingual corpus features the following characteristics:

- It contains 939 languages, 546 of which have more than 100k tokens and are used for training our EMMA-500 model, and 74 billion (B) whitespace delimited tokens in total.

- It has more than 300 languages with over 1 million whitespace delimited tokens and 546 languages with over 100k tokens.

- It comes with four publicly available versions: (1) a noisy version after only basic pre-processing like extraction and harmonization; (2) a cleaned version after data cleaning; (3) a deduplicated version after approximate and exact deduplication; (4) a split version with train and valid splits.

- Our augmentation to the MaLA corpus includes different types of texts such as code, books, scientific papers and instruction data, leading to a data mix with 100B+ whitespace delimited tokens.

**Evaluation Results**    In comparison to other decoder-only LLMs with parameter sizes from 4.5B to 13B, including Llama 2-based multilingual models and the latest advanced models, our EMMA-500 7B parameter model demonstrates strong performance. It achieves the lowest negative log-likelihood in intrinsic evaluation and significantly improves commonsense reasoning, machine translation, and open-ended generation tasks. It also outperforms all Llama 2-based models and multilingual LLMs in text classification and natural language inference. While math and machine reading comprehension remain challenging, our model still enhances the Llama 2 base model, and it surpasses the base model in code generation without regression in performance.

## 2    THE MALA CORPUS

The MaLA corpus—MaLA standing for **Ma**ssive **L**anguage **A**daptation—is a diverse and extensive compilation of text data encompassing 939 languages sourced from a wide array of datasets. It is developed for continual training of multilingual large language models. The source datasets the MaLA corpus is compiled from exhibit a wide range of variance in various aspects. Examples of such elements include the data quality, the nature of the text content, how the data sources were organized into directory and file tree structures (if distributed as files rather than through an API) and the naming conventions used therein, the data formats and structures in the files or in-memory objects containing the text data, and the logic by which multilingual texts were aligned. This section introduces the efforts made in data extraction, harmonization, pre-processing, cleaning, and deduplication in order to build the corpus.

### 2.1    DATA PRE-PROCESSING

To develop the MaLA corpus for training our language model, we establish a processing workflow consisting of the following key steps: (1) loading and curating identified data sources, (2) extracting and harmonizing both textual and metadata from these diverse sources into a unified format—often with tailored filtering, and (3) performing deduplication and further filtering on the textual data. In step (1), source data are either organized and loaded into memory from a file tree structure or, if

---

[1]Choosing Llama 2 allows us to compare our model with many other models derived from it using continual pre-training. We plan to continue training models based on Llama 3/3.1 in the future.

available, accessed through an API. In step (2), the output is designed to be `JSON` Lines (`JSONL`) files with extracted text data and other desired content. A `JSONL` file contains multiple JSON records for storing data, each separated by a newline character. We selectively process only data annotated as training or development (validation) data, deliberately excluding test data.

### 2.1.1 EXTRACTION AND HARMONIZATION

As mentioned, there is significant variability in the data quality of the source datasets used for compiling the corpus. Many of the datasets exhibit data quality challenges that would have adverse effects on model training if left unaddressed. These dataset-specific challenges are typically addressed during the pre-processing stage. For example, we identify one issue involving text records consisting solely of date and timestamp information in the dataset for Languages of Russia (Corpora and Tools, n.d.), likely resulting from a web scraper failing to differentiate between these elements and actual text. We address this by implementing a logic in the pre-processing script to detect and exclude such records. This issue is resolved by introducing a rule to identify and discard text containing consecutive repeating words. Due to the extensive volume of data, exhaustive examination of every source for data quality issues is impractical. Instead, we address issues only as we encounter them in our data exploration and pre-processing pipeline development efforts. This approach likely leaves some data quality problems undetected, since we do not go through the data systematically. Despite the need for customized handling of certain dataset-specific idiosyncrasies, the core logic and structure of the pre-processing workflow remain consistent across most datasets. We develop a standardized pre-processing script that can be adapted with minor adjustments to accommodate different datasets.

### 2.1.2 LANGUAGE CODE NORMALIZATION

An essential component of our pre-processing pipeline involved converting language codes to the ISO 639-3 standard. This is crucial for ensuring consistent language identification across the source datasets. We rely on the declared language of each dataset and normalize it to the ISO standard without performing additional language identification at this step. This approach helps maintain uniformity while streamlining the pre-processing workflow. We primarily use the PyPI library `iso639-lang`[2] or `langcodes`[3]. While converting language codes to ISO 639-3, we encounter several challenges. One issue is that some languages in ISO 639-3 are divided into multiple subvarieties, but our source data does not specify which subvariety is present. Our solution is to retain the original language code from the dataset, even when it does not conform to the ISO 639-3 standard. Another issue arises when certain languages are merged into other languages in the ISO 639-3 standard. In these cases, we update the language code to reflect the merged language. Additionally, some language names or codes in the source data—referred to as "original language names" or "original language codes"—are not recognized by the conversion libraries. In some cases, the reason behind this is that the original code in fact represents language families or groups of dialects (e.g., the ISO 639-2 codes "ber" for Berber languages and "bih" for Bihari languages), rather than specific languages. If so, we then retain the original codes, despite their non-compliance with ISO 639-3. In other cases, the original language names are spelled differently from the standard recognized by the libraries. To address this mismatch, we implement a logic to detect and correct misspelled language names during pre-processing. All these "corner cases" require careful attention in the pre-processing stage to ensure correct language code identification.

### 2.1.3 WRITING SYSTEM RECOGNITION

In addition to normalizing language codes, we also identify the script or writing system used in the text data. We use the `GlotScript` library (Kargaran et al., 2024) to recognize writing systems accordant to the ISO 15924 standard. The process begins by sampling 100 random lines from each dataset (or the full dataset if it contains fewer than 100 lines). If `GlotScript` fails to identify a script from this sample, we attempt identification using just the first line of the sample. If this still does not yield a result, we set the script as "None". It is worth noting that we choose not to classify a dataset into multiple scripts, even when code-mixing (i.e., the use of multiple scripts) is present.

---

[2]https://pypi.org/project/iso639-lang/
[3]https://pypi.org/project/langcodes

If script identification is unsuccessful after the initial steps, we assume the script matched a previously detected one for that language. In cases where no previous script information exists, we refer to a mapping of languages and their default scripts provided by the Glot500 corpora collection. Through this multi-step process, we are able to determine the script for every dataset without exceptions.

During script identification, we encounter several challenges. One issue is determining an appropriate length for the text chunk used for script recognition. A chunk that is too short could lead to incorrect identification if the text contains quotes or foreign language fragments using a different writing system. Conversely, using a chunk that is too long could result in excessive resource usage, slowing down processing or even causing memory exhaustion. Another consideration is whether to assume that a single file or dataset might contain multiple scripts. Such an assumption would require identifying the script at a more granular level, such as paragraph by paragraph or even sentence by sentence. Alternatively, we could assume that each file or dataset contained only one "main" script. This assumption would allow us to identify the script from a representative sample of the text for the whole file or dataset. We adopt the latter approach, recognizing a single dominant script for each dataset. The output of this process is a label in the format language_Script, e.g., eng_Latn, where "Language" represents the ISO 639-3 language code and "Script" represents the ISO 15924 script code.

## 2.2 DATA CLEANING

Most source data has already undergone data cleaning to different extents. Nonetheless, different cleaning processes have been adopted. We continue to clean the data to ensure consistency and accuracy for monolingual and bilingual texts. Following the pipeline used by BigScience's pipeline for ROOTS corpus (Laurençon et al., 2022), we further adopt some necessary data cleaning to filter out text samples that might have undesirable quality. We first perform document modification for monolingual texts. The first step is whitespace standardization: all types of whitespace in a document are converted into a single, consistent space character. We split documents by newline characters, tabs, and spaces, strip words, and reconstruct the documents to remove very long words. However, these two steps do not apply to languages without whitespace word delimiters like Chinese, Japanese, Korean, Thai, Lao, Burmese, etc. We also remove words containing certain patterns, e.g., "http" and ".com", which are likely to be links and page source code. We then perform document filtering, including word count filtering, character repetition filtering, word repetition filtering, special characters filtering, stop words filtering, and flag words filtering. We re-identify the languages that are supported by the pre-trained `fastText`-based language identification model (Joulin et al., 2016b;a). For other languages, we assume the language identification of the original data source and language code conversion are reliable.

As we collect data from different sources, we deduplicate the data to remove the overlap between different sources using MinHash and exact deduplication (Mou et al., 2023), with details described in Appendix A.3.

## 2.3 KEY STATISTICS

This section presents the final MaLA corpus obtained after data sourcing, pre-processing, cleaning, and deduplication. Table 1 first shows some basic data statistics and compares them with other multilingual corpora for pre-training language models or language adaptation. Additional statistics are presented in Appendix B in the appendix. The MaLA corpus harvests a wide range of datasets in multiple domains. Table 7 in the appendix lists the corpora and collections we used as monolingual data sources. The token counts are based on white-space delimitation, though it might not be accurate for languages like Chinese, Japanese and Korean since the entire clause is counted as one token. Glot500-c (Imani et al., 2023) has 534 languages in total, in which 454 languages are directly distributed on Huggingface [4]. We also omit high-resource languages in the other three datasets, i.e., MADLAD (Kudugunta et al., 2024), CulturaX (Nguyen et al., 2023), and CC100 (Wenzek et al., 2020), as our main focus is continual pre-training for language adaptation. The final MaLA corpus consists of 939 languages, 546 of which have more than 100k tokens and are used for training our EMMA-500 model. Counting languages with more than 1 million tokens, the MaLA corpus and Glot500-c have more than 300, while MADLAD and CulturaX have 200 and 100 respectively. Compared with Glot500-c, the MaLA corpus contains documents with significantly higher sequence

---

[4] https://huggingface.co/datasets/cis-lmu/Glot500

Table 1: Data statistics of the MaLA corpus and comparison to other multilingual corpora. The number of documents and tokens is in millions.

| Dataset | N Lang | N Lang Counted | N Docs | N Tokens | Avg Tokens/Doc |
|---|---|---|---|---|---|
| Glot500-c | 534 | 454 | 1,815 | 35,449 | 19.53 |
| MADLAD | 419 | 414 | 1,043 | 645,111 | 618.51 |
| CulturaX | 167 | 161 | 2,141 | 1,029,810 | 480.99 |
| CC100 | 116 | 101 | 2,557 | 52,201 | 20.41 |
| MaLA | 939 | 546 | 824 | 74,255 | 90.12 |

lengths, with an average token count of 90 versus 19. This higher sequence length is advantageous for continually training LLMs because it provides more context within each training example, allowing the model to better capture long-range dependencies and patterns in the data. As a result, MaLA is more effective for language adaptation.

## 3 DATA MIXING AND MODEL TRAINING

Incorporating a diverse data mix—spanning various languages, domains, document lengths and styles—is crucial for continual training of large language models to enhance their versatility, generalization ability, and robustness across a wide range of tasks and domains. We augment the MaLA corpus with diverse data to mitigate issues such as over-fitting to specific styles or topics or underperforming on tasks outside the training distribution.

### 3.1 DATA MIXING

**Curated Data** We enhance the corpus with high-quality curated data, specifically high-resource languages in the monolingual part. We use texts from scientific papers as these provide a structured, information-dense corpus that can improve the model's ability to handle technical language and domain-specific content. They are (1) CSL (Li et al., 2022), a large-scale Chinese Scientific Literature dataset, that contains titles, abstracts, keywords and academic fields of 396,209 papers; (2) pes2o (Soldaini & Lo, 2023), a collection of full-text open-access academic papers derived from the Semantic Scholar Open Research Corpus (S2ORC) (Lo et al., 2020). We further add free e-books from the Gutenberg project[5] compiled by Faysse (2023). These texts enhance the range of literary styles and narrative forms, thus enhancing the model's versatility. Adding high-resource languages into the pre-training corpora also mitigates the forgetting in model training.

**Instruction Data** We further augmented the training corpus by incorporating instruction-based datasets, inspired by Li et al. (2023); Taylor et al. (2022); Nakamura et al. (2024). We mix two instruction data into our training corpus. They are: (1) xp3x (Crosslingual Public Pool of Prompts eXtended)[6], a multitask instruction collection in 277 languages (Muennighoff et al., 2022); (2) the Aya collection [7] that contains both human-curated and machine translated instructions in 101 languages (Singh et al., 2024). For both instruction datasets, we use their training set.

**Code** We additionally enrich the training corpus by sourcing code data from The Stack (Kocetkov et al., 2023). This is done following existing work that demonstrates the value of code data in improving the reasoning ability of language models (Zhang et al., 2024b; Ma et al., 2024) while also mitigating any catastrophic forgetting of the base model's programming knowledge.

We subsample The Stack at an effective rate of 15.2%, prioritizing high-quality source files and data science code [8]. We retain the 32 most important non-data programming languages by prevalence while also adding in all `llvm` code following prior work detailing its importance in multi-lingual code generation (Szafraniec et al., 2023; Paul et al., 2024). We also source from data-heavy formats

---

[5] https://www.gutenberg.org/

[6] https://huggingface.co/datasets/CohereForAI/xP3x

[7] https://huggingface.co/datasets/CohereForAI/aya_collection_language_split

[8] https://huggingface.co/datasets/AlgorithmicResearchGroup/arxiv_research_code

Table 2: Data mix for continual training. Code and reasoning-related data are counted by Llama 2 tokenizer and others are counted as whitespace delimited; 'inst' stands for instruction and 'mono' stands for monolingual texts.

| Data | Original Counts | Sample Rate | Final Counts | Percentage |
|------|----------------|-------------|--------------|------------|
| inst high | 42,121,055,562 | 0.1 | 4,212,105,556 | 3.08% |
| inst medium-high+ | 6,486,592,274 | 0.2 | 1,297,318,455 | 0.95% |
| inst medium-high | 30,651,187,534 | 0.5 | 15,325,593,767 | 11.21% |
| inst medium | 1,444,764,863 | 1.0 | 1,444,764,863 | 1.06% |
| inst medium-low | 47,691,495 | 5.0 | 238,457,475 | 0.17% |
| inst low | 3,064,796 | 20.0 | 61,295,920 | 0.04% |
| inst code/reasoning | 612,208,775 | 1.0 | 612,208,775 | 0.45% |
| code | 221,003,976,266 | 0.1 | 20,786,882,764 | 15.20% |
| curated (EN pes2o) | 56,297,354,921 | 0.2 | 11,241,574,489 | 8.22% |
| curated (ZH CSL & wiki) | 61,787,372 | 1.0 | 61,787,372 | 0.05% |
| curated (Gutenberg) | 5,173,357,710 | 1.0 | 5,173,357,710 | 3.78% |
| mono high EN | 3,002,029,817 | 0.1 | 300,202,982 | 0.22% |
| mono high | 40,411,201,964 | 0.5 | 20,205,600,982 | 14.78% |
| mono medium-high | 27,515,227,962 | 1.0 | 27,515,227,962 | 20.12% |
| mono medium | 2,747,484,380 | 5.0 | 13,737,421,900 | 10.05% |
| mono medium-low | 481,935,633 | 20.0 | 9,638,712,660 | 7.05% |
| mono low | 97,535,696 | 50.0 | 4,876,784,800 | 3.57% |

but follow precedent (Lozhkov et al., 2024) and subsample them more aggressively. For a more detailed read on filtering heuristics, we direct the reader to Appendix A.2.

**Data Mix** Our final data mix for continual training is listed in Table 2. The resource categorization refers to Appendix B.1 in the appendix and `inst medium-high+` is a separate category with languages with more than 500 million but less than 1B tokens. For monolingual text, we also have a seperate category for English. In continual learning, where new data is introduced to an existing model, there is a risk of "catastrophic forgetting", where the model loses knowledge from earlier training stages. Although our work's primary focus is in a low-resource regime, we enhance the training corpus with a wide range of data types, including books and scientific papers in high-resource languages, code, and instruction data in our data mix. We downsample texts in high-resource languages and upsample text in low-resource languages using different sample rates according to how resourceful the language is. We make our data mix diverse and balanced towards different resource groups of languages in order to retain the prior knowledge of the model while learning new information, especially in medium- and low-resource languages, thus maintaining high performance across both previously seen and new languages. The final data mix has around 136B tokens.

## 3.2 MODEL TRAINING

We employ continual training using the causal language modelling objective for the decoder-only Llama model and exposing the pre-trained model to new data to develop our EMMA-500 model. We adopt efficient training strategies combining optimization, memory management, precision handling, and distributed training techniques. Our EMMA-500 model is trained on the Leonardo supercomputer[9], occupying 256 Nvidia A100 GPUs, using the GPT-NeoX framework (Andonian et al., 2023). During training, we set a global batch size of 4096 and worked with sequences of 4096 tokens. The training process ran for 12,000 steps, resulting in a total of 200 billion Llama 2 tokens. We use the Adam optimizer (Kingma & Ba, 2015) with a learning rate of 0.0001, betas of [0.9, 0.95], and an epsilon of 1e-8. We use a cosine learning rate scheduler with a warm-up of 500 iterations. To reduce memory consumption, activation checkpointing is employed. Precision is managed through mixed-precision techniques, using bfloat16 for computational efficiency and maintaining FP32 for gradient accumulation.

## 4 EVALUATION

### 4.1 TASKS, BENCHMARKS, AND BASELINES

**Tasks and Benchmarks** We conduct a comprehensive evaluation to validate the usability of our processed data and data mixing for massively multilingual language adaptation. We perform the

---

[9]https://leonardo-supercomputer.cineca.eu

Table 3: Evaluation statistics. Sample/Lang: average number of test samples per language; N Lang: number of languages covered; NLL: negative log-likelihood; ACC: accuracy.

| Tasks | Dataset | Metric | Samples/Lang | N Lang | Domain |
|---|---|---|---|---|---|
| Intrinsic Evaluation (Appendix E.1) | Glot500-c test (Imani et al., 2023) | NLL | 1000 | 534 | Misc |
| | PBC (Mayer & Cysouw, 2014) | NLL | 500 | 370 | Bible |
| Text Classification (Section 4.3) | SIB200 (Adelani et al., 2023) | ACC | 204 | 205 | Misc |
| | Taxi1500 (Ma et al., 2023) | ACC | 111 | 1507 | Bible |
| Commonsense Reasoning (Appendix E.3) | XCOPA (Ponti et al., 2020) | ACC | 600 | 11 | Misc |
| | XStoryCloze (Lin et al., 2022) | ACC | 1870 | 11 | Misc |
| | XWinograd (Tikhonov & Ryabinin, 2021) | ACC | 741.5 | 6 | Misc |
| Natural Language Inference (Appendix E.5) | XNLI (Conneau et al., 2018) | ACC | 2490 | 15 | Misc |
| Machine Translation (Section 4.2) | FLORES-200 (Costa-jussà et al., 2022) | BLEU, chrF++ | 1012 | 204 | Misc |
| Open-Ended Generation (Section 4.4) | Aya (Singh et al., 2024) | BLEU, Self-BLEU | 215 | 119 | Misc |
| | PolyWrite (Ours) | Self-BLEU | 149 | 240 | Misc |
| Summarization (Appendix E.2) | XL-Sum (Hasan et al., 2021) | ROUGE-L, BERTScore | 2537 | 44 | News |
| Math (Appendix E.6) | MGSM direct (Shi et al., 2022) | ACC | 250 | 10 | Misc |
| | MGSM CoT (Shi et al., 2022) | ACC | 250 | 10 | Misc |
| Machine Comprehension (Appendix E.7) | BELEBELE (Bandarkar et al., 2023) | ACC | 900 | 122 | Misc |
| | ARC multilingual (Lai et al., 2023) | ACC | 1170 | 31 | Misc |
| Code Generation (Appendix E.8) | Multipl-E (Cassano et al., 2022) | Pass@$k$ | 164 | 7 | Misc |

intrinsic evaluation of the models' performance on next-word prediction and evaluate the model's performance on downstream tasks. Table 3 lists the datasets we used as downstream evaluation datasets in this work.

We present the evaluation results of intrinsic evaluation in Appendix E.1, commonsense reasoning in Appendix E.3, text summarization in Appendix E.2, math tasks in Appendix E.6, machine reading comprehension in Appendix E.7, and coding generation in Appendix E.8.

**Baselines** We compare our model with three groups of decoder-only models. They are (1) Llama 2 models (Touvron et al., 2023) and continual pre-trained models based on Llama 2, such as CodeL-lama (Roziere et al., 2023), MaLA-500 (Lin et al., 2024), LLaMAX (Lu et al., 2024), Tower (Alves et al., 2024), and YaYi[10]; (2) other LLMs and continual pre-trained LLMs designed to be massively multilingual, including BLOOM (Scao et al., 2022), mGPT (Shliazhko et al., 2022), XGLM (Lin et al., 2022), and Occiglot[11]; and (3) recent LLMs with superior English capabilities like Llama 3 (Dubey et al., 2024), Llama 3.1[12], Qwen 2 (Yang et al., 2024a), and Gemma 2 (Team et al., 2024). There are also some other LLMs such as OpenAI's API models [13] and xLLMs-100 (Lai et al., 2024). However, they do not release the model weights or they limit access to them through commercial API, so we did not include them. The MADLAD model (Kudugunta et al., 2024) that uses the decoder-only T5 architecture is not supported by inference engines such as the HuggingFace transformers (Wolf et al., 2019). We do not compare them in this work.

## 4.2 MACHINE TRANSLATION

FLORES-200 is an evaluation benchmark for translation tasks with 204 language pairs involving English and thus 408 translation directions, with a particular focus on low-resource languages. We assess all language models by adopting a 3-shot evaluation approach with the prompt in Appendix D.1.

The performance is measured by BLEU (Papineni et al., 2002) and chrF++ (Popović, 2015) implemented in `sacrebleu` (Post, 2018). The BLEU score is calculated with the `flores200` tokenizer applied to the texts and chrF++ uses word order 2. The choice of `flores200` tokenization ensures that languages that do not have a whitespace delimiter can be evaluated at the (sub-)word level. For reproducibility, we attach the BLEU and chrF++ signatures.[14,15]

---

[10] https://huggingface.co/wenge-research/yayi-7b-llama2
[11] https://huggingface.co/occiglot/occiglot-7b-eu5
[12] https://llama.meta.com/docs/model-cards-and-prompt-formats/llama3_1
[13] https://platform.openai.com/docs/models
[14] BLEU: nrefs:1—case:mixed—eff:no—tok:flores200—smooth:exp—version:2.4.2
[15] chrF++: nrefs:1—case:mixed—eff:yes—nc:6—nw:2—space:no—version:2.4.2

Table 4: 3-shot results on FLORES-200 (X-Eng, BLEU/chrF++). EMMA-500 Llama 2 7B has better average performance than all baselines.

| Model | Avg | High | Medium-High | Medium | Medium-Low | Low |
|---|---|---|---|---|---|---|
| Llama 2 7B | 12.93/ 30.32 | 19.91/ 39.04 | 17.56/ 35.84 | 12.49/ 29.81 | 8.27/ 24.35 | 6.96/ 23.36 |
| Llama 2 7B Chat | 12.28/ 31.72 | 18.98/ 39.65 | 17.06/ 37.03 | 11.74/ 31.1 | 7.79/ 26.34 | 6.18/ 25.03 |
| CodeLlama 2 7B | 10.82/ 28.57 | 17.39/ 37.43 | 15.27/ 33.94 | 10.39/ 28.05 | 6.45/ 22.85 | 5.04/ 21.48 |
| LLaMAX Llama 2 7B | 1.99/ 13.66 | 3.68/ 22.18 | 2.95/ 18.15 | 1.83/ 12.84 | 0.67/ 7.2 | 1.01/ 9.04 |
| LLaMAX Llama 2 7B Alpaca | 22.29/ 42.27 | 32.83/ 54.56 | 30.04/ 51.25 | 21.7/ 41.94 | 14.24/ 31.32 | 14.24/ 32.88 |
| MaLA-500 Llama 2 10B v1[‡] | 2.29/ 13.6 | 4.64/ 15.95 | 3.18/ 14.64 | 2.68/ 14.23 | 1.24/ 12.58 | 0.33/ 11.18 |
| MaLA-500 Llama 2 10B v2[‡] | 2.87/ 15.44 | 5.58/ 18.65 | 3.81/ 16.33 | 3.55/ 16.29 | 1.63/ 14.2 | 0.55/ 12.76 |
| Yayi Llama 2 7B | 12.98/ 31.38 | 19.48/ 39.58 | 17.55/ 36.71 | 12.47/ 30.79 | 8.54/ 25.63 | 7.22/ 24.84 |
| TowerBase Llama 2 7B | 13.74/ 31.47 | 21.76/ 40.96 | 18.92/ 37.27 | 13.15/ 30.9 | 8.3/ 25.05 | 7.21/ 24.1 |
| TowerInstruct Llama 2 7B | 4.81/ 25.43 | 9.18/ 34.4 | 6.66/ 30.01 | 4.62/ 25.22 | 2.64/ 20.24 | 1.8/ 18.69 |
| Occiglot Mistral 7B v0.1 | 13.12/ 31.13 | 19.53/ 38.93 | 17.57/ 36.27 | 13.07/ 31.2 | 9.03/ 26.15 | 6.86/ 23.83 |
| Occiglot Mistral 7B v0.1 Instruct | 11.61/ 31.65 | 16.72/ 39.28 | 15.06/ 36.48 | 11.7/ 31.73 | 8.48/ 26.88 | 6.54/ 24.7 |
| BLOOM 7B | 9.57/ 27.84 | 15.75/ 36.65 | 9.65/ 28.19 | 9.42/ 27.81 | 6.81/ 23.95 | 8.61/ 25.89 |
| BLOOMZ 7B[†] | 20.22/ 34.74 | 32.23/ 47.03 | 19.2/ 34.08 | 20.09/ 34.49 | 16.25/ 30.58 | 18.54/ 32.63 |
| mGPT | 5.29/ 20.69 | 9.37/ 26.64 | 8.28/ 25.29 | 3.41/ 17.87 | 2.43/ 16.07 | 2.84/ 17.28 |
| mGPT-13B | 7.42/ 24.58 | 12.61/ 31.95 | 11.11/ 30.16 | 5.72/ 22.49 | 3.57/ 18.16 | 4.11/ 20.04 |
| Yayi 7B | 4.82/ 21.36 | 5.69/ 25.18 | 4.53/ 19.97 | 4.41/ 21.52 | 3.71/ 19.18 | 6.13/ 23.12 |
| Llama 3 8B | 23.78/ 43.72 | 33.71/ 55.36 | 30.31/ 51.3 | 24.75/ 44.91 | 15.18/ 33.65 | 16.01/ 34.65 |
| Llama 3.1 8B | 24.19/ 44.1 | 34.15/ 55.7 | 30.79/ 51.7 | 24.98/ 45.26 | 15.89/ 34.24 | 16.13/ 34.85 |
| Gemma 2 9B | 23.15/ 38.87 | 33.11/ 51.36 | 30.81/ 48.53 | 25.58/ 41.23 | 15.37/ 30.03 | 11.73/ 24.15 |
| Gemma 7B | 23.79/ 43.68 | 34.23/ 55.77 | 29.87/ 50.95 | 24.0/ 44.25 | 16.16/ 34.36 | 16.03/ 34.58 |
| Qwen 1.5 7B | 15.58/ 35.87 | 24.07/ 46.29 | 19.92/ 40.74 | 15.76/ 36.27 | 9.74/ 28.81 | 9.77/ 29.13 |
| Qwen 2 7B | 17.39/ 37.61 | 27.63/ 50.06 | 22.48/ 43.28 | 18.13/ 38.63 | 9.89/ 28.54 | 10.64/ 29.99 |
| EMMA-500 Llama 2 7B | 25.37/ 45.78 | 32.24/ 53.74 | 31.39/ 52.85 | 25.72/ 46.16 | 20.32/ 39.96 | 17.18/ 36.15 |

Table 4 presents the average X-to-English (X-Eng) translation results.[16] Our EMMA-500 model outperforms all other models on average. We achieve the best performance across all language settings, except for high-resource languages where our model slightly lags behind Llama 3/3.1, Gemma 7B, and LlaMAX 7B Alpaca. In the English-to-X (Eng-X) translation direction, as shown in Table 16 in Appendix E.4, the advantage of EMMA-500 is even more pronounced. We outperform all other models even in high-resource languages, and the advantage becomes more significant in lower-resource languages. Overall, we note that our model outperforms Tower models which are explicitly adjusted to perform translation tasks in high-resource languages. Further, the much larger margin between EMMA-500 and other models in Eng-X compared with X-Eng indicates that our EMMA-500 model is particularly good at generating non-English texts.

## 4.3 TEXT CLASSIFICATION

SIB-200 (Adelani et al., 2023) and Taxi1500 (Ma et al., 2023) are two prominent topic classification datasets. SIB-200 encompasses seven categories: science/technology, travel, politics, sports, health, entertainment, and geography. Taxi1500 spans 1507 languages, involving six classes: Recommendation, Faith, Description, Sin, Grace, and Violence. We use 3-shot prompting with prompts in Appendix D.2, drawing demonstrations from the development set and testing models on the test split. The outcomes on SIB-200 and TAXI-1500 are tabulated in Table 5. For SIB-200, our EMMA-500 model outperforms all Llama2-based models, with particularly notable gains in languages with medium or fewer resources—seeing an average improvement of 47.5%. Taxi-1500 could be a more challenging task since it is in the religious domain, but our model still surpasses all Llama2-based models except for MalA-500. However, despite these improvements in both classification tasks, our models lag behind the latest models such as Llama3 and 3.1, especially in high-resource languages.

## 4.4 OPEN-ENDED GENERATION

**Aya Evaluation** We choose the two subsets `aya-human-annotated` and `dolly-machine-translated` from the Aya evaluation suite (Singh et al., 2024), which have both inputs and targets for subsequent evaluation. To quantitatively assess the quality of the generated text by the models, we employ two metrics: BLEU (Papineni et al., 2002) and Self-BLEU (Zhu et al.,

---

[16]We mark BLOOMZ with a † because it has used FLORES in its instruction tuning data; we mark MaLA-500 with a ‡ because it has used FLORES in its training data but with source and target sides split. Besides, as a remark, Tower, LLaMAX, and our EMMA-500 have intentionally used parallel data (not FLORES) in the training stage.

Table 5: 3-shot results on SIB-200 and Taxi-1500 (ACC). EMMA-500 Llama 2 7B has better average performance than Llama 2 models and comparable performance with multilingual LLMs, and has comparable performance with the compared LLMs.

| Model | SIB-200 | | | | | | Taxi-1500 | | | | | |
|---|---|---|---|---|---|---|---|---|---|---|---|---|
| | Avg | High | Med-High | Medium | Med-Low | Low | Avg | High | Med-High | Medium | Med-Low | Low |
| Llama 2 7B | 0.2241 | 0.2664 | 0.2469 | 0.2205 | 0.1968 | 0.1900 | 0.1754 | 0.1950 | 0.1949 | 0.1847 | 0.1746 | 0.1737 |
| Llama 2 7B Chat | 0.2558 | 0.2972 | 0.2811 | 0.2501 | 0.2303 | 0.2191 | 0.1544 | 0.1873 | 0.1766 | 0.1661 | 0.1559 | 0.1522 |
| CodeLlama 2 7B | 0.2335 | 0.2606 | 0.2542 | 0.2310 | 0.2142 | 0.2037 | 0.1703 | 0.1745 | 0.1741 | 0.1720 | 0.1705 | 0.1700 |
| LLaMAX Llama 2 7B | 0.1061 | 0.1242 | 0.1160 | 0.1001 | 0.0945 | 0.0954 | 0.2352 | 0.2320 | 0.2340 | 0.2376 | 0.2356 | 0.2352 |
| LLaMAX Llama 2 7B Alpaca | 0.2789 | 0.3309 | 0.3212 | 0.2716 | 0.2338 | 0.2282 | 0.1509 | 0.1870 | 0.1688 | 0.1599 | 0.1500 | 0.1491 |
| MaLA-500 Llama 2 10B v1 | 0.2325 | 0.2330 | 0.2364 | 0.2288 | 0.2276 | 0.2358 | 0.2527 | 0.2390 | 0.2402 | 0.2476 | 0.2457 | 0.2543 |
| MaLA-500 Llama 2 10B v2 | 0.1930 | 0.1893 | 0.2105 | 0.1949 | 0.1755 | 0.1846 | 0.2339 | 0.2136 | 0.2230 | 0.2132 | 0.2172 | 0.2367 |
| Yayi Llama 2 7B | 0.2457 | 0.2904 | 0.2717 | 0.2442 | 0.2144 | 0.2069 | 0.1773 | 0.1874 | 0.1846 | 0.1819 | 0.1789 | 0.1765 |
| TowerBase Llama 2 7B | 0.1934 | 0.2200 | 0.2092 | 0.1874 | 0.1790 | 0.1693 | 0.1773 | 0.1849 | 0.1881 | 0.1867 | 0.1810 | 0.1761 |
| TowerInstruct Llama 2 7B | 0.2053 | 0.2321 | 0.2196 | 0.2026 | 0.1915 | 0.1804 | 0.1729 | 0.2017 | 0.1960 | 0.1808 | 0.1740 | 0.1709 |
| Occiglot Mistral 7B v0.1 | 0.3269 | 0.3880 | 0.3582 | 0.3174 | 0.2892 | 0.2836 | 0.2226 | 0.2464 | 0.2291 | 0.2299 | 0.2233 | 0.2215 |
| Occiglot Mistral 7B v0.1 Instruct | 0.3431 | 0.3948 | 0.3716 | 0.3336 | 0.3147 | 0.3008 | 0.1876 | 0.2430 | 0.2090 | 0.1941 | 0.1918 | 0.1848 |
| BLOOM 7B | 0.1781 | 0.2313 | 0.1805 | 0.1717 | 0.1576 | 0.1702 | 0.1476 | 0.1558 | 0.1489 | 0.1456 | 0.1511 | 0.1473 |
| BLOOMZ 7B | 0.2973 | 0.3039 | 0.2963 | 0.2980 | 0.2953 | 0.2970 | 0.1696 | 0.1693 | 0.1698 | 0.1696 | 0.1699 | 0.1695 |
| mGPT | 0.2711 | 0.2858 | 0.2799 | 0.2673 | 0.2589 | 0.2648 | 0.1072 | 0.0867 | 0.0844 | 0.0992 | 0.1029 | 0.1093 |
| mGPT-13B | 0.3320 | 0.3669 | 0.3427 | 0.3448 | 0.2939 | 0.3226 | 0.1723 | 0.1798 | 0.1644 | 0.1588 | 0.1610 | 0.1738 |
| XGLM 7.5B | 0.3181 | 0.3528 | 0.3512 | 0.3169 | 0.2696 | 0.2996 | 0.2041 | 0.2421 | 0.2369 | 0.2125 | 0.2105 | 0.2010 |
| Yayi 7B | 0.3576 | 0.4057 | 0.3620 | 0.3563 | 0.3472 | 0.3318 | 0.1612 | 0.1665 | 0.1638 | 0.1645 | 0.1583 | 0.1611 |
| Llama 3 8B | 0.6369 | 0.7345 | 0.7025 | 0.6462 | 0.5316 | 0.5696 | 0.2173 | 0.3184 | 0.2708 | 0.2560 | 0.2261 | 0.2105 |
| Llama 3.1 8B | 0.6142 | 0.7070 | 0.6790 | 0.6199 | 0.5146 | 0.5475 | 0.2020 | 0.2751 | 0.2521 | 0.2443 | 0.2097 | 0.1959 |
| Gemma 7B | 0.5821 | 0.6806 | 0.6455 | 0.5816 | 0.4886 | 0.5112 | 0.1805 | 0.2868 | 0.2855 | 0.2499 | 0.2028 | 0.1689 |
| Gemma 2 9B | 0.4625 | 0.5177 | 0.4900 | 0.4692 | 0.4304 | 0.4045 | 0.1383 | 0.2429 | 0.2413 | 0.1874 | 0.1497 | 0.1283 |
| Qwen 1.5 7B | 0.4795 | 0.5600 | 0.5181 | 0.4825 | 0.4156 | 0.4286 | 0.0729 | 0.1265 | 0.1145 | 0.0878 | 0.0730 | 0.0692 |
| Qwen 2 7B | 0.5495 | 0.6637 | 0.6014 | 0.5517 | 0.4519 | 0.4925 | 0.2187 | 0.2737 | 0.2557 | 0.2401 | 0.2233 | 0.2147 |
| EMMA-500 Llama 2 7B | 0.3127 | 0.3275 | 0.3328 | 0.3099 | 0.3083 | 0.2760 | 0.1982 | 0.2366 | 0.2333 | 0.2277 | 0.2200 | 0.1930 |

2018). BLEU is crucial for assessing the linguistic accuracy and relevance of the generated text in comparison to the expected human-like text present in the dataset. Self-BLEU is used to evaluate the diversity of the text generated by a model. It measures how similar different texts from the same model are to each other by treating one generated text as the "candidate" and others as the "reference" texts. This metric is useful in scenarios where high degrees of variation are desirable, as it helps identify models that might be overfitting to particular styles or patterns of text.

The results are presented in Table 6. The BLEU metric is an indicator that measures how generated texts are close to the references. However, in open-ended generation, LLMs cannot generate identical texts to references, leading to low BLEU scores. The EMMA-500 model obtains remarkably high BLEU scores in both high-resource and low-resource settings when compared with baselines. Its performance in low-resource settings is particularly noteworthy, as it not only sustains high BLEU scores but also exhibits a Self-BLEU score of 5.09, the highest among all models evaluated. This high Self-BLEU score indicates less diversity in the generated text, suggesting that while EMMA-500 maintains consistency, it may produce less varied outputs. When compared to other high-performing models like Qwen 2 7B and Llama 3.1 8B, the EMMA-500 model exhibits a superior balance between accuracy and linguistic creativity. Unlike Qwen 2 7B, which shows a spike in performance primarily in medium-low resource settings, EMMA-500 maintains a consistently high performance across varying levels of resource availability.

**PolyWrite** This is a novel multilingual benchmark composed in this work for evaluating open-ended generation in 240 languages. We use ChatGPT to generate different prompts in English and use Google Translate to translate them into different languages for models to generate creative content. This benchmark consists of 31 writing tasks, such as storytelling and email writing, and 155 prompts in total. We back-translate the multilingual prompts to English, calculate the BLEU scores between original English prompts and back-translation, and filter out translated prompts with BLEU scores below 20, and the entire dataset contains a total of 35,751 prompts. The details of PolyWrite are described in Appendix C.1.

We use Self-BLEU (Zhu et al., 2018) to evaluate the diversity of generated texts in the PolyWrite benchmark, as presented in Table 6. A lower Self-BLEU score indicates more diverse generation, but does not means a better generation quality. Our EMMA-500 model demonstrates comparable performance across various languages. Compared to other models like Llama 3/3.1 and Qwen 1.5/2,

particularly in medium-low and low-resource languages, EMMA-500 has higher Self-BLEU scores, indicating lower diversity in its generated content.

Evaluating open-ended generation poses significant challenges, as it goes beyond simply measuring accuracy or correctness. Metrics like BLEU or Self-BLEU, while useful for assessing similarity to reference texts or the diversity of given texts, often fail to capture more nuanced aspects. Subjective factors like cultural relevance and the appropriateness of responses in low-resource languages are difficult to quantify. This makes it challenging to create evaluation benchmarks and metrics that fully capture the strengths and weaknesses of models like EMMA-500 in diverse, real-world scenarios.

Table 6: Results on Aya (BLEU/Self-BLEU) and PolyWrite (Self-BLEU). EMMA-500 Llama 2 7B has higher average BLEU scores than all baselines on Aya.

| Model | Aya | | | | | | PolyWrite | | | | | |
|---|---|---|---|---|---|---|---|---|---|---|---|---|
| | Avg | High | Med-High | Medium | Med-Low | Low | Avg | High | Med-High | Medium | Med-Low | Low |
| Llama 2 7B | 1.24/0.74 | 1.27/0.57 | 1.47/0.60 | 0.86/0.37 | 1.17/0.45 | 0.77/1.87 | 0.5358 | 0.4282 | 0.6545 | 0.3769 | 0.4766 | 0.6587 |
| Llama 2 7B Chat | 1.17/1.29 | 1.46/1.15 | 1.36/1.15 | 0.69/1.03 | 1.14/1.14 | 0.54/2.23 | 1.1550 | 0.8640 | 0.8902 | 1.1877 | 1.4435 | 1.4167 |
| CodeLlama 2 7B | 1.22/1.21 | 1.31/1.19 | 1.40/1.00 | 0.85/0.84 | 1.23/0.56 | 0.78/2.57 | 1.0313 | 1.2052 | 1.2883 | 0.9191 | 0.9092 | 0.6798 |
| LLaMAX Llama 2 7B | 1.72/1.70 | 1.80/1.37 | 2.03/1.48 | 1.30/1.24 | 1.65/0.89 | 0.98/3.74 | 0.9564 | 1.0066 | 1.1655 | 0.8786 | 0.8363 | 0.7709 |
| LLaMAX Llama 2 7B Alpaca | 1.68/1.67 | 1.82/1.22 | 1.96/1.42 | 1.28/1.14 | 1.55/1.04 | 1.00/3.94 | 0.8086 | 0.7321 | 0.9517 | 0.8322 | 0.8351 | 0.4981 |
| MaLA-500 Llama 2 10B v1 | 0.40/2.29 | 0.42/2.53 | 0.49/2.79 | 0.32/1.22 | 0.31/2.42 | 0.18/1.16 | 3.7079 | 3.7902 | 3.5066 | 2.4541 | 4.8052 | 3.5163 |
| MaLA-500 Llama 2 10B v2 | 0.41/2.31 | 0.42/2.01 | 0.50/2.65 | 0.32/1.42 | 0.33/1.02 | 0.18/3.09 | 4.0059 | 3.1737 | 4.0148 | 3.0476 | 5.0652 | 3.8916 |
| Yayi Llama 2 7B | 1.65/0.61 | 1.84/0.82 | 1.88/0.62 | 1.26/0.62 | 1.53/0.45 | 1.02/0.41 | 0.6274 | 0.6207 | 0.6921 | 0.4169 | 0.6813 | 0.6352 |
| TowerBase Llama 2 7B | 1.44/0.83 | 1.45/0.64 | 1.67/0.56 | 1.19/0.49 | 1.46/0.38 | 0.88/2.54 | 0.4938 | 0.7268 | 0.4736 | 0.3945 | 0.4417 | 0.5396 |
| TowerInstruct Llama 2 7B | 1.55/0.93 | 1.80/0.85 | 1.82/0.78 | 1.15/0.65 | 1.21/0.54 | 0.85/1.97 | 0.7124 | 0.9565 | 0.7651 | 0.6492 | 0.5998 | 0.6615 |
| Occiglot Mistral 7B v0.1 | 1.53/2.43 | 1.57/0.91 | 1.78/2.89 | 1.17/1.73 | 1.61/1.31 | 0.95/4.27 | 0.9647 | 0.8818 | 0.7975 | 0.9543 | 0.9563 | 1.4157 |
| Occiglot Mistral 7B v0.1 Instruct | 0.75/2.81 | 0.82/2.38 | 0.89/3.10 | 0.43/2.72 | 0.79/1.17 | 0.45/3.50 | 3.9033 | 5.3884 | 4.7140 | 3.7555 | 2.8444 | 2.9568 |
| BLOOM 7B | 0.85/1.17 | 0.92/1.20 | 0.96/1.32 | 0.73/1.05 | 0.78/1.27 | 0.55/0.72 | 1.3892 | 1.2845 | 1.6705 | 1.9685 | 1.1513 | 0.6600 |
| BLOOMZ 7B | 0.12/0.61 | 0.07/0.33 | 0.17/0.62 | 0.08/1.00 | 0.06/0.89 | 0.08/0.51 | 0.0024 | 0.0000 | 0.0005 | 0.0093 | 0.0000 | 0.0049 |
| mGPT | 1.24/0.55 | 1.22/0.64 | 1.47/0.59 | 0.91/0.48 | 1.21/0.60 | 0.84/0.29 | 0.7560 | 0.9222 | 0.6291 | 0.4534 | 0.8156 | 1.1134 |
| mGPT-13B | 1.42/0.57 | 1.42/0.80 | 1.63/0.58 | 1.00/0.48 | 1.53/0.44 | 1.03/0.36 | 0.7479 | 0.8483 | 0.7091 | 0.4230 | 0.7962 | 1.0201 |
| Yayi 7B | 1.05/0.39 | 1.22/0.38 | 1.18/0.41 | 0.76/0.42 | 1.01/0.54 | 0.67/0.22 | 0.5151 | 0.4266 | 0.3574 | 0.3283 | 0.6706 | 0.8574 |
| Llama 3 8B | 1.59/0.94 | 1.03/0.60 | 1.31/0.53 | 2.96/0.54 | 1.11/0.28 | 2.43/3.47 | 0.5796 | 0.5753 | 0.5921 | 0.7141 | 0.5586 | 0.4440 |
| Llama 3.1 8B | 1.85/1.08 | 1.41/0.95 | 1.60/0.64 | 3.11/0.52 | 1.33/0.39 | 2.52/3.52 | 0.7995 | 0.6805 | 0.8253 | 0.5044 | 0.6965 | 1.3585 |
| Gemma 2 9B | 1.55/0.82 | 1.59/0.94 | 1.73/0.93 | 1.38/0.70 | 1.33/0.47 | 1.21/0.65 | 1.1736 | 1.2347 | 1.2913 | 1.1616 | 0.9261 | 1.3240 |
| Gemma 7B | 1.29/0.57 | 1.41/0.62 | 1.39/0.66 | 1.16/0.40 | 1.26/0.50 | 0.93/0.40 | 1.0541 | 0.9629 | 1.1222 | 0.9275 | 1.1112 | 1.0284 |
| Qwen 1.5 7B | 1.93/1.13 | 1.66/0.85 | 1.64/0.65 | 3.14/0.79 | 1.51/0.58 | 2.45/3.69 | 0.6441 | 0.9398 | 0.7457 | 0.3797 | 0.4164 | 0.8738 |
| Qwen 2 7B | 1.99/1.13 | 1.84/0.94 | 1.69/0.70 | 3.29/0.72 | 1.36/0.44 | 2.47/3.53 | 0.5709 | 0.7763 | 0.6135 | 0.3695 | 0.4186 | 0.7989 |
| EMMA-500 Llama 2 7B | 2.93/1.54 | 2.90/1.29 | 2.75/0.83 | 3.79/0.82 | 2.86/0.95 | 2.87/5.09 | 0.9879 | 0.9833 | 0.7058 | 0.8465 | 1.3130 | 1.1702 |

## 5 CONCLUSION AND OUTLOOKS

This paper addresses critical advancements and challenges in adapting language models to more than 500 languages, focusing on enhancing their performance across diverse languages. We compile the MaLA corpus, a multilingual dataset for continual pre-training of multilingual language models. By expanding and augmenting existing corpora, we train the EMMA-500 model. It demonstrates notable improvements in a range of tasks such as next-token prediction, commonsense reasoning, machine translation, text classification, and open-ended generation, showing remarkable improvements in low-resource languages. Our results show that a well-curated, massively multilingual corpora can advance model capabilities. This work sets a new benchmark for inclusive and effective multilingual language models and paves the way for future research to address the disparities between high-resource and low-resource languages.

Multilingual language models are designed to cater to diverse linguistic and cultural backgrounds, yet many existing multilingual benchmarks rely on human or machine translations, primarily reflecting English-speaking communities and introducing imperfections that affect evaluation integrity. We emphasize the need for natively-created multilingual test sets to provide more accurate assessments. While our EMMA-500 model shows enhanced multilingual performance compared to Llama 2 7B and other variants, it falls short on certain benchmarks compared to newer models like Llama 3 and Gemma 2. Future efforts will explore multilingual extension with these models and direct pre-training using the released corpus, along with multilingual instruction tuning to improve task performance and interactions.

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

## A  DATA SOURCES

### A.1  MONOLINGUAL DATA

Table 7 lists the corpora and collections we use as monolingual data sources in this work. Monolingual data sources simply contain text data in a single language.

**Metadata** For the monolingual data sources, we define the contents of the `JSONL` output file from the pre-processing workflow to consist of the fields `url`, `text`, `collection`, `source`, and `original_code`. The contents of these fields are as follows. The field `text` contains the language data in a granularity specific to the given corpus. If the granularity was sentence-level, then we could expect the sentences in the corpus to generally be independent of each other, while only parts of the

sentences—such as phrases, clauses, and words—exhibit serial dependence. If the granularity was paragraph-level, then we could expect the sentences within paragraphs to have serial dependence, while paragraphs to largely be independent of each other. The field `url` contains a URL indicating the web address from which the text data has been extracted, if available. The field `collection` contains the name of the collection, i.e., a corpus or a set of corpora, which the text is extracted from, whereas the field `source` contains the name of a more specific part of the collection, such as the name of an individual corpus or a file in the collection the text was extracted from. Lastly, the field `original_code` contains the language code of the text data as it is designated in the data source, e.g., in the directory structure, the filenames, or the data object returned by an API call.

Table 7: Datasets used as monolingual source data.

| Name | Languages | Domains | URL | Year |
|---|---|---|---|---|
| AfriBERTa (Ogueji et al., 2021) | 10 | news | https://huggingface.co/datasets/castorini/afriberta-corpus | 2021 |
| Bloom library (Leong et al., 2022) | 363 | religious, books | https://huggingface.co/datasets/sil-ai/bloom-lm | 2022 |
| CC100 (Conneau et al., 2020) | 100 | web | https://huggingface.co/datasets/cc100 | 2020 |
| CulturaX (Nguyen et al., 2023) | 167 | web | https://huggingface.co/datasets/uonlp/CulturaX | 2023 |
| CulturaY (Thuat Nguyen & Nguyen, 2024) | 75 | web | https://huggingface.co/datasets/ontocord/CulturaY | 2024 |
| curse-of-multilinguality (Chang et al., 2023) | 200 | misc | https://huggingface.co/datasets/tylerachang/curse-of-multilinguality | 2023 |
| Evenki Life (Life, 2014) | 1 | newspapers | https://drive.google.com/file/d/1he2q6RncA_NKHPIJjSz1kK-2qgEFTiCG/view | 2014 |
| Glot500 (Imani et al., 2023) | 511 | misc | https://huggingface.co/datasets/cis-lmu/Glot500 | 2023 |
| GlotSparse (Kargaran et al., 2023) | 10 | news | https://huggingface.co/datasets/cis-lmu/GlotSparse | 2023 |
| HPLT v1.2 (de Gibert et al., 2024) | 75 | web | https://hplt-project.org/datasets/v1.2 | 2024 |
| Indigenous Languages Corpora (EdTeKLA, 2022) | 1 | UNK | https://github.com/EdTeKLA/IndigenousLanguages_Corpora | 2022 |
| Indo4B (Wilie et al., 2020) | 1 | misc | https://github.com/IndoNLP/indonlu?tab=readme-ov-file | 2020 |
| Lacuna Project (Masakhane, 2023) | 20 | UNK | https://github.com/masakhane-io/lacuna_pos_ner | 2023 |
| Languages of Russia (Corpora and Tools, n.d.) | 46 | social media, web | http://web-corpora.net/wsgi3/minorlangs/download | UNK |
| Madlad-400 (Kudugunta et al., 2024) | 419 | web | https://huggingface.co/datasets/allenai/MADLAD-400 | 2023 |
| Makerere Radio Speech Corpus (Mukiibi et al., 2022) | 1 | transcription | https://zenodo.org/records/5855017 | 2022 |
| masakhane-ner1.0 (Ifeoluwa Adelani et al., 2021) | 12 | UNK | https://github.com/masakhane-io/masakhane-ner | 2021 |
| MC2 (Zhang et al., 2024a) | 4 | web | https://huggingface.co/datasets/pkupie/mc2_corpus | 2024 |
| mC4 (Raffel et al., 2020) | 101 | web | https://huggingface.co/datasets/allenai/c4 | 2020 |
| multilingual-data-peru (Grupo de Inteligencia Artificial PUCP, n.d.) | 4 | UNK | https://github.com/iapucp/multilingual-data-peru | 2020 |
| OSCAR 2301 (OSCAR, 2023) | 152 | web | https://huggingface.co/datasets/oscar-corpus/OSCAR-2301 | 2023 |
| The Leipzig Corpora (Goldhahn et al., 2012) | 136 | newspapers, wikipedia | https://huggingface.co/datasets/imvladikon/leipzig_corpora_collection | 2012 |
| Tigrinya Language Modeling (Gaim et al., 2021) | 1 | news, blogs, books | https://zenodo.org/records/5139094 | 2021 |
| Wikipedia 20231101 (Wikimedia Foundation, n.d.) | 323 | wikipedia | https://huggingface.co/datasets/wikimedia/wikipedia | 2023 |
| Wikisource 20231201 (Wikimedia Foundation, n.d.) | 73 | books | https://huggingface.co/datasets/wikimedia/wikisource | 2023 |
| Tatoeba challenge monolingual (Tiedemann, 2020) | 280 | wikimedia | https://github.com/Helsinki-NLP/Tatoeba-Challenge/blob/master/data/MonolingualData-v2020-07-28.md | 2020 |

### A.1.1 LIST OF DATA SOURCES

The major ones include AfriBERTa (Ogueji et al., 2021), Bloom library (Leong et al., 2022), CC100 (Conneau et al., 2020; Wenzek et al., 2020) CulturaX (Nguyen et al., 2023), CulturaY (Thuat Nguyen & Nguyen, 2024), the Curse of Multilinguality (Chang et al., 2023), Evenki Life (Life, 2014), Glot500 (Imani et al., 2023), GlotSparse (Kargaran et al., 2023), monoHPLT of HPLT v1.2 (de Gibert et al., 2024) from the HPLT project (Aulamo et al., 2023), Indigenous Languages Corpora (EdTeKLA, 2022), Indo4B (Wilie et al., 2020), Lacuna Project (Masakhane, 2023), Languages of Russia (Corpora and Tools, n.d.), MADLAD-400 (Kudugunta et al., 2024), Makerere Radio Speech Corpus (Mukiibi et al., 2022), masakhane-ner1.0 (Ifeoluwa Adelani et al., 2021), MC2 (Zhang et al., 2024a), mC4 (Raffel et al., 2020), multilingual-data-peru (Grupo de Inteligencia Artificial PUCP, n.d.), OSCAR 2301 (OSCAR, 2023) from the OSCAR project [17], Tatoeba challenge monolingual collection (Tiedemann, 2020), The Leipzig Corpora (Goldhahn et al., 2012), Tigrinya Language Modeling (Gaim et al., 2021), Wikipedia 20231101 (Wikimedia Foundation, n.d.) and Wikisource 20231201 (Wikimedia Foundation, n.d.). We exclude high-resource languages in CulturaX, HPLT, MADLAD-400, CC100, mC4, and OSCAR 2301. We exclude Gahuza and Pidgin in the AfriBERTa dataset. We filter out texts that mainly contain a date or timestamp in the Languages of Russia dataset. For Glot500-c, we filter out texts, which may come from train or test tests from datasets for machine translation, such as Flores200, Tatoeba, and mtdata. Despite the translation data being split into source and target languages in the Glot500-c corpus, we decide to filter them to avoid potential data leakage, especially since we use Flores200 as the evaluation benchmark.

Glot500-c uses the following datasets: AI4Bharat,[18] AIFORTHAI-LotusCorpus,[19] Add (El-Haj et al., 2018), AfriBERTa (Ogueji et al., 2021), AfroMAFT (Adelani et al., 2022; Xue et al., 2021), Anuvaad,[20] AraBench (Sajjad et al., 2020), AUTSHUMATO,[21] Bloom (Leong et al., 2022), CC100

---

[17] https://oscar-project.org/
[18] https://ai4bharat.org/
[19] https://github.com/korakot/corpus/releases/download/v1.0/AIFORTHAI-LotusCorpus.zip
[20] https://github.com/project-anuvaad/anuvaad-parallel-corpus
[21] https://autshumato.sourceforge.net/

(Conneau et al., 2020), CCNet (Wenzek et al., 2020), CMU_Haitian_Creole,[22] CORP.NCHLT,[23] Clarin,[24] DART (Alsarsour et al., 2018), Earthlings (Dunn, 2020), FFR,[25] Flores200 (Costa-jussà et al., 2022), GiossaMedia (Góngora et al., 2022; 2021), Glosses (Camacho-Collados et al., 2016), Habibi (El-Haj, 2020), HinDialect (Bafna, 2022), HornMT,[26] IITB (Kunchukuttan et al., 2018), IndicNLP (Nakazawa et al., 2021), Indiccorp (Kakwani et al., 2020), isiZulu,[27] JParaCrawl (Morishita et al., 2020), KinyaSMT,[28] LeipzigData (Goldhahn et al., 2012), Lindat,[29] Lingala_Song_Lyrics,[30] Lyrics,[31] MC4 (Raffel et al., 2020), MTData (Gowda et al., 2021), MaCoCu (Bañón et al., 2022), Makerere MT Corpus,[32] Masakhane community,[33] Mburisano_Covid,[34] Menyo20K (Adelani et al., 2021), Minangkabau corpora (Koto & Koto, 2020), MoT (Palen-Michel et al., 2022), NLLB_seed (Costa-jussà et al., 2022), Nart/abkhaz,[35] OPUS (Tiedemann, 2012), OSCAR (Suárez et al., 2019), ParaCrawl (Bañón et al., 2020), Parallel Corpora for Ethiopian Languages (Abate et al., 2018), Phontron (Neubig, 2011), QADI (Abdelali et al., 2021), Quechua-IIC (Zevallos et al., 2022), SLI_GalWeb.1.0 (Agerri et al., 2018), Shami (Abu Kwaik et al., 2018), Stanford NLP,[36] StatMT,[37] TICO (Anastasopoulos et al., 2020), TIL (Mirzakhalov et al., 2021), Tatoeba,[38] TeDDi (Moran et al., 2022), Tilde (Rozis & Skadiņš, 2017), W2C (Majliš, 2011), WAT (Nakazawa et al., 2022), WikiMatrix (Schwenk et al., 2021a), Wikipedia,[39] Workshop on NER for South and South East Asian Languages (Singh, 2008), XLSum (Hasan et al., 2021). We filter out Flores200 when processing the Glot500-c dataset. Glot500-c includes texts in languages, i.e., Azerbaijani, Gujarati, Igbo, Oromo, Rundi, Tigrinya and Yoruba, from the XLSum dataset.

## A.2  CODE

All the programming language splits are filtered using the following conditions:

- For files forked more than 25 times, we retain them if the average line length is less than 120, the maximum line length is less than 300, and the alphanumeric fraction is more than 30%.

- For files forked between 15 and 25 times, we retain them if the average line length is less than 90, the maximum line length is less than 150, and the alphanumeric fraction is more than 40%.

- For files forked less than 15 times, we retain them if the average line length is less than 80, the maximum line length is less than 120, and the alphanumeric fraction is more than 45%.

Subsequently, an aggressive MinHash deduplication pipeline with a threshold of 0.5 and a shingle size of 20 is applied. Finally, the resultant language splits are then capped at 5 million samples each.

## A.3  DATA DEDUPLICATION

As we collect data from different sources, we deduplicate the data to remove the overlap between different sources.

---

[22] http://www.speech.cs.cmu.edu/haitian/text/
[23] https://repo.sadilar.org/handle/20.500.12185/7
[24] https://www.clarin.si/
[25] https://github.com/bonaventuredossou/ffr-v1/tree/master/FFR-Dataset
[26] https://github.com/asmelashteka/HornMT
[27] https://zenodo.org/record/5035171
[28] https://github.com/pniyongabo/kinyarwandaSMT
[29] https://lindat.cz/faq-repository
[30] https://github.com/espoirMur/songs_lyrics_webscrap
[31] https://lyricstranslate.com/
[32] https://zenodo.org/record/5089560
[33] https://github.com/masakhane-io/masakhane-community
[34] https://repo.sadilar.org/handle/20.500.12185/536
[35] https://huggingface.co/datasets/Nart/abkhaz_text
[36] https://nlp.stanford.edu/
[37] https://statmt.org/
[38] https://tatoeba.org/en/
[39] https://huggingface.co/datasets/wikipedia

**MinHash Deduplication**    For each language's dataset in each writing system, we start by using the MinHashLSH algorithm (Rajaraman & Ullman, 2011) to filter out similar documents. It is a near-deduplication technique that builds on MinHash (Broder, 1997), utilizing multiple hash functions for n-grams and the Jaccard similarity, and incorporates Locality-Sensitive Hashing to enhance efficiency. We use the implementation by `text-dedup` repository (Mou et al., 2023), applying 5-grams and a similarity threshold of 0.7 to identify similar documents based on the Jaccard index.

**Exact Deduplication**    We further deploy exact deduplication using the `text-dedup` repository again with precise matching. It takes each document through a hash function, i.e., MD5 (Rivest, 1992) in our choice, and the hash values of all documents are compared to identify duplicates.

## B    ADDITIONAL STATISTICS OF MALA CORPUS

### B.1    SUPPORTED LANGUAGES

Table 8 shows the languages codes of MaLA corpus, where "unseen" means the languages are not used for training EMMA-500. The classification system for token counts categorizes language resources based on their size into five distinct tiers: "high" for resources exceeding 1 billion tokens, indicating a vast amount of data; "medium-high" for those with more than 100 million tokens, reflecting a substantial dataset; "medium" for resources that contain over 10 million tokens, representing a moderate size; "medium-low" for datasets with over "1 million tokens", indicating a smaller yet significant amount of data; and finally, "low" for resources containing less than 1 million tokens, which suggests a minimal data presence. This hierarchy helps in understanding the scale and potential utility of the language resources available. Figure 1 shows the number of texts and tokens in different resource groups.

### B.2    DATA ANALYSIS

We examine the Unicode block distribution of each language, which counts the percentage of tokens falling into the Unicode block of each language. This aims to check whether language code conversion and writing system recognition are reasonably good. Figure 2 shows the Unicode block distribution. The result aligns with our observation of the presence of code-mixing, but in general, the majority of languages have tokens falling in their own Unicode blocks.

We also check the data source distribution as shown in Figure 3 to see where the texts in the MaLA corpus come from. The main source is common crawl, e.g., CC 2018, CC, OSCAR, and CC 20220801. A large number of documents come from Earthlings which comes from Glot500-c (Imani et al., 2023). For web-crawled data with a URL in the original metadata of corpora like CulturaX (Nguyen et al., 2023) and HPLT (de Gibert et al., 2024), we extract the domain. Thus, the final corpus has many sources with a small portion.

## C    EVALUATION BENCHMARKS

### C.1    POLYWRITE

The PolyWrite dataset has 51 writing tasks with the number of prompts per task shown in Figure 4. We use Google Translate to translate the English prompts into 240 languages and back-translate for translation quality assessment.

The metadata of PolyWrite includes several key fields. The `category` specifies the task type. The `name` field typically holds the specific identifier for each prompt, while `prompt_en` contains the English version of the prompt. `lang_script` identifies the language and script used, ensuring correct language processing. The `prompt_translated` field holds the translated prompt in the target language, and `prompt_backtranslated` contains the back-translated version to assess translation quality. Both `bleu` and `chrf++` fields provide numeric evaluation metrics, with BLEU and chrF++ scores measuring the quality of the generated text. Finally, the `uuid` ensures a unique identifier for each dataset entry, allowing for precise reference and tracking of individual prompts. To

Table 8: Languages by resource groups

| Category | Languages | Language Codes |
|---|---|---|
| high | 27 | fra_Latn, mon_Cyrl, kat_Geor, tgk_Cyrl, kaz_Cyrl, glg_Latn, hbs_Latn, kan_Knda, mal_Mlym, rus_Cyrl, cat_Latn, hye_Armn, guj_Gujr, slv_Latn, fil_Latn, bel_Cyrl, isl_Latn, nep_Deva, mlt_Latn, pan_Guru, afr_Latn, urd_Arab, mkd_Cyrl, aze_Latn, deu_Latn, eng_Latn, ind_Latn |
| low | 210 | prs_Arab, nqo_Nkoo, emp_Latn, pfl_Latn, teo_Latn, gpe_Latn, izz_Latn, shn_Mymr, hak_Latn, pls_Latn, evn_Cyrl, djk_Latn, toj_Latn, nog_Cyrl, ctu_Latn, tca_Latn, jiv_Latn, ach_Latn, mrj_Latn, ajp_Arab, apc_Arab, tab_Cyrl, hvn_Latn, tls_Latn, bak_Latn, ndc_Latn, trv_Latn, top_Latn, kjh_Cyrl, guh_Latn, mni_Mtei, csy_Latn, noa_Latn, dov_Latn, bho_Deva, kon_Latn, hne_Deva, kcg_Latn, mni_Beng, hus_Latn, pau_Latn, jbo_Latn, dtp_Latn, kmb_Latn, hau_Arab, pdc_Latn, nch_Latn, acf_Latn, bim_Latn, ixl_Latn, dty_Deva, kas_Arab, lrc_Arab, alz_Latn, lez_Cyrl, lld_Latn, tdt_Latn, acm_Arab, bih_Deva, mzh_Latn, guw_Latn, rop_Latn, rwo_Latn, ahk_Latn, qub_Latn, kri_Latn, gub_Latn, laj_Latn, sxn_Latn, luo_Latn, tly_Latn, pwn_Latn, mag_Deva, xav_Latn, bum_Latn, ubu_Latn, roa_Latn, mah_Latn, tsg_Latn, gcr_Latn, arn_Latn, csb_Latn, guc_Latn, bat_Latn, knj_Latn, cre_Latn, bus_Latn, anp_Deva, aln_Latn, nah_Latn, zai_Latn, kpv_Cyrl, enq_Latn, gvl_Latn, wal_Latn, fiu_Latn, swh_Latn, crh_Latn, nia_Latn, bqc_Latn, map_Latn, atj_Latn, npi_Deva, bru_Latn, din_Latn, pis_Latn, gur_Latn, cuk_Latn, zne_Latn, cdo_Latn, lhu_Latn, pcd_Latn, mas_Latn, bis_Latn, ncj_Latn, ibb_Latn, tay_Latn, bts_Latn, tzj_Latn, bzj_Latn, cce_Latn, jvn_Latn, ndo_Latn, rug_Latn, koi_Cyrl, mco_Latn, fat_Latn, olo_Latn, inb_Latn, mkn_Latn, qvi_Latn, mak_Latn, ktu_Latn, nrm_Latn, kua_Latn, san_Latn, nbl_Latn, kik_Latn, dyu_Latn, sgs_Latn, msm_Latn, mnw_Latn, zha_Latn, sja_Latn, xal_Cyrl, rmc_Latn, ami_Latn, sda_Latn, tdx_Latn, yap_Latn, tzh_Latn, sus_Latn, ikk_Latn, bas_Latn, nde_Latn, dsb_Latn, seh_Latn, knv_Latn, amu_Latn, dwr_Latn, iku_Cans, uig_Latn, bxr_Cyrl, tcy_Knda, mau_Latn, aoj_Latn, gor_Latn, cha_Latn, fip_Latn, chr_Cher, mdf_Cyrl, arb_Arab, quw_Latn, shp_Latn, spp_Latn, frp_Latn, ape_Latn, cbk_Latn, mnw_Mymr, mfe_Latn, lad_Latn, awa_Deva, mad_Latn, ote_Latn, shi_Latn, btx_Latn, maz_Latn, ppk_Latn, smn_Latn, twu_Latn, blk_Mymr, msi_Latn, naq_Latn, tly_Arab, wuu_Hani, mos_Latn, cab_Latn, zlm_Latn, gag_Latn, suz_Deva, ksw_Mymr, gug_Latn, nij_Latn, nov_Latn, srm_Latn, jac_Latn, nyu_Latn, yom_Latn, gui_Latn |
| medium | 68 | tha_Thai, kat_Latn, lim_Latn, tgk_Arab, che_Cyrl, lav_Latn, xho_Latn, war_Latn, nan_Latn, grc_Grek, orm_Latn, zsm_Latn, cnh_Latn, yor_Latn, arg_Latn, tgk_Latn, azj_Latn, tel_Latn, slk_Latn, pap_Latn, zho_Hani, sme_Latn, tgl_Latn, uzn_Cyrl, als_Latn, san_Deva, azb_Arab, ory_Orya, lmo_Latn, bre_Latn, mvf_Mong, fao_Latn, oci_Latn, sah_Cyrl, sco_Latn, tuk_Latn, aze_Arab, hin_Deva, haw_Latn, glk_Arab, oss_Cyrl, lug_Latn, tet_Latn, tsn_Latn, hrv_Latn, gsw_Latn, arz_Arab, vec_Latn, mon_Latn, ilo_Latn, ctd_Latn, ben_Beng, roh_Latn, kal_Latn, asm_Beng, srp_Latn, bod_Tibt, hif_Latn, rus_Latn, nds_Latn, lus_Latn, ido_Latn, lao_Laoo, tir_Ethi, chv_Cyrl, wln_Latn, kaa_Latn, epo_Latn, nya_Latn |
| medium-high | 79 | div_Thaa, som_Latn, jpn_Japn, hat_Latn, sna_Latn, heb_Hebr, bak_Cyrl, nld_Latn, tel_Telu, kin_Latn, msa_Latn, gla_Latn, bos_Latn, dan_Latn, smo_Latn, ita_Latn, mar_Deva, pus_Arab, srp_Cyrl, spa_Latn, lat_Latn, hmn_Latn, sin_Sinh, zul_Latn, bul_Cyrl, amh_Ethi, ron_Latn, tam_Taml, khm_Khmr, nno_Latn, cos_Latn, fin_Latn, ori_Orya, uig_Arab, hbs_Cyrl, gle_Latn, cym_Latn, vie_Latn, kor_Hang, lit_Latn, yid_Hebr, ara_Arab, sqi_Latn, pol_Latn, tur_Latn, swa_Latn, hau_Latn, ceb_Latn, eus_Latn, kir_Cyrl, mlg_Latn, jav_Latn, snd_Arab, sot_Latn, por_Latn, uzb_Cyrl, fas_Arab, nor_Latn, est_Latn, hun_Latn, ibo_Latn, ltz_Latn, swe_Latn, tat_Cyrl, ast_Latn, mya_Mymr, uzb_Latn, sun_Latn, ell_Grek, ces_Latn, mri_Latn, ckb_Arab, kur_Latn, kaa_Cyrl, nob_Latn, ukr_Cyrl, fry_Latn, epo_Latn, nya_Latn |
| medium-low | 162 | aym_Latn, rue_Cyrl, rom_Latn, dzo_Tibt, poh_Latn, sat_Olck, ary_Arab, fur_Latn, mbt_Latn, bpy_Beng, iso_Latn, pon_Latn, glv_Latn, new_Deva, gym_Latn, bgp_Latn, kac_Latn, abt_Latn, quc_Latn, otq_Latn, sag_Latn, cak_Latn, avk_Latn, pam_Latn, meo_Latn, tum_Latn, bam_Latn, kha_Latn, syr_Syrc, kom_Cyrl, nhe_Latn, bal_Arab, srd_Latn, krc_Cyrl, lfn_Latn, bar_Latn, rcf_Latn, nav_Latn, nnb_Latn, sdh_Arab, aka_Latn, bew_Cyrl, bbc_Latn, meu_Latn, zza_Latn, ext_Latn, yue_Hani, ekk_Latn, xmf_Geor, nap_Latn, mzn_Arab, pcm_Latn, lij_Latn, myv_Cyrl, scn_Latn, dag_Latn, ban_Latn, twi_Latn, udm_Cyrl, som_Arab, nso_Latn, pck_Latn, crs_Latn, acr_Latn, tat_Latn, afb_Arab, uzs_Arab, hil_Latn, mgh_Latn, tpi_Latn, ady_Cyrl, pag_Latn, kiu_Latn, ber_Latn, iba_Latn, ksh_Latn, plt_Latn, lin_Latn, chk_Latn, tzo_Latn, tlh_Latn, ile_Latn, lub_Latn, hui_Latn, min_Latn, bjn_Latn, szl_Latn, kbp_Latn, inh_Cyrl, que_Latn, ven_Latn, vls_Latn, kbd_Cyrl, run_Latn, wol_Latn, ace_Latn, ada_Latn, kek_Latn, yua_Latn, tbz_Latn, gom_Latn, ful_Latn, mrj_Cyrl, abk_Cyrl, tuc_Latn, stq_Latn, mwl_Latn, tvl_Latn, quh_Latn, gom_Deva, mhr_Cyrl, fij_Latn, grn_Latn, zap_Latn, mam_Latn, mps_Latn, tiv_Latn, ksd_Latn, ton_Latn, bik_Latn, vol_Latn, ava_Cyrl, tso_Latn, szy_Latn, ngu_Latn, hrw_Armn, fon_Latn, skr_Arab, kos_Latn, tyz_Latn, kur_Arab, srn_Latn, tyv_Cyrl, bci_Latn, vep_Latn, crh_Cyrl, kpg_Latn, hsb_Latn, ssw_Latn, zea_Latn, ewe_Latn, ium_Latn, diq_Latn, ltg_Latn, nzi_Latn, guj_Deva, ina_Latn, pms_Latn, bua_Cyrl, lvs_Latn, eml_Latn, hmo_Latn, kum_Cyrl, kab_Latn, chm_Cyrl, cor_Latn, cfm_Latn, alt_Cyrl, bcl_Latn, ang_Latn, frr_Latn, mai_Deva |
| unseen | 393 | rap_Latn, pmf_Latn, lsi_Latn, dje_Latn, bkx_Latn, ipk_Latn, syw_Deva, ann_Latn, bag_Latn, bat_Cyrl, chu_Cyrl, gwc_Arab, adh_Latn, szy_Hani, shi_Arab, njy_Latn, pdu_Latn, buo_Latn, cuv_Latn, udg_Mlym, bax_Latn, tio_Latn, kjb_Latn, taj_Deva, lez_Latn, olo_Cyrl, rnl_Latn, bri_Latn, inh_Latn, kas_Cyrl, wni_Latn, anp_Latn, tsc_Latn, mgg_Latn, udi_Cyrl, mdf_Latn, agr_Latn, xty_Latn, llg_Latn, nge_Latn, gan_Latn, tuv_Latn, stk_Latn, nut_Latn, thy_Thai, lgr_Latn, hnj_Latn, dar_Cyrl, aia_Latn, lwl_Thai, tnl_Latn, tvs_Latn, jra_Khmr, tay_Hani, gal_Latn, ybi_Deva, snk_Arab, gag_Cyrl, tuk_Cyrl, trv_Hani, ydd_Hebr, kea_Latn, gbm_Deva, kwi_Latn, hro_Latn, rki_Latn, quy_Latn, tdg_Deva, zha_Hani, pcg_Mlym, tom_Latn, nsn_Latn, quf_Latn, jmx_Latn, kqr_Latn, mrn_Latn, bxa_Latn, abc_Latn, mve_Arab, lfa_Latn, qup_Latn, yin_Latn, roo_Latn, mrw_Latn, nxa_Latn, yrk_Cyrl, bem_Latn, kvt_Latn, csw_Cans, bjr_Latn, mgm_Latn, ngn_Latn, pib_Latn, quz_Latn, awb_Latn, myk_Latn, otq_Arab, ino_Latn, tkd_Latn, bef_Latn, bug_Bugi, aeu_Latn, nlv_Latn, dty_Latn, bkc_Latn, mmu_Latn, hak_Hani, sea_Latn, mlk_Latn, cbr_Latn, lmp_Latn, tnn_Latn, qvz_Latn, pbt_Arab, cjs_Cyrl, mlw_Latn, mnf_Latn, bfm_Latn, dig_Latn, thk_Latn, zxx_Latn, lkb_Latn, chr_Latn, pnt_Latn, vif_Latn, fli_Latn, got_Latn, hbb_Latn, tll_Latn, bug_Latn, kxp_Arab, qaa_Latn, krr_Khmr, kjg_Laoo, isu_Latn, kmu_Latn, gof_Latn, sdk_Latn, mne_Latn, baw_Latn, idt_Latn, xkg_Latn, mgo_Latn, dtr_Latn, kms_Latn, ffm_Latn, hna_Latn, nxl_Latn, bfd_Latn, odk_Arab, miq_Latn, mhx_Latn, kam_Latn, yao_Latn, pnt_Grek, kby_Latn, kpv_Latn, kbx_Latn, cim_Latn, qvo_Latn, pih_Latn, nog_Latn, nco_Latn, rmy_Cyrl, clo_Latn, dmg_Latn, aaa_Latn, rel_Latn, ben_Latn, loh_Latn, thl_Deva, chd_Latn, cni_Latn, cjs_Latn, lbe_Latn, ybh_Deva, zxx_Zyyy, awa_Latn, gou_Latn, xmm_Latn, nqo_Latn, rut_Cyrl, kbq_Latn, tkr_Latn, dwr_Ethi, ckt_Cyrl, ady_Latn, yea_Mlym, nhx_Latn, niv_Cyrl, bwt_Latn, xmg_Latn, chy_Latn, mfj_Latn, hre_Latn, bbk_Latn, shn_Latn, lrc_Latn, qvc_Latn, muv_Mlym, mdr_Latn, luy_Latn, lzh_Hani, fuh_Latn, mle_Latn, brx_Deva, pex_Latn, kau_Latn, yrk_Latn, hin_Latn, ekm_Latn, msb_Latn, unr_Orya, cac_Latn, chp_Cans, ckt_Latn, bss_Latn, lts_Latn, bbj_Latn, ttt_Cyrl, kwu_Latn, smn_Cyrl, kpy_Cyrl, tod_Latn, wbm_Latn, tcy_Latn, arc_Syrc, nst_Latn, tuz_Latn, bob_Latn, bfn_Latn, pli_Deva, snl_Latn, kwd_Latn, lgg_Latn, nza_Latn, wbr_Deva, lan_Latn, kmz_Latn, bzi_Thai, hao_Latn, nla_Latn, qxr_Latn, ken_Latn, tbj_Latn, blk_Latn, ybb_Latn, nwe_Latn, gan_Hani, snk_Latn, kak_Latn, tpl_Latn, hla_Latn, tks_Arab, pea_Latn, bya_Latn, enc_Latn, jgo_Latn, tnp_Latn, aph_Deva, bgf_Latn, brv_Laoo, nod_Thai, niq_Latn, nwi_Latn, xmd_Latn, gbj_Orya, syr_Latn, ify_Latn, xal_Latn, bra_Deva, cgc_Latn, bhs_Latn, pwg_Latn, ang_Runr, oki_Latn, qve_Latn, qvm_Latn, bkm_Latn, bkh_Latn, niv_Latn, zuh_Latn, mry_Latn, fiu_Cyrl, ssn_Latn, rki_Mymr, sox_Latn, yav_Latn, nyo_Latn, dag_Arab, qxh_Latn, bze_Latn, myx_Latn, zaw_Latn, ddg_Latn, wnk_Latn, bwx_Latn, mqy_Latn, lad_Hebr, boz_Latn, lue_Latn, ded_Latn, pli_Latn, avk_Cyrl, wms_Latn, sgd_Latn, azn_Latn, ajz_Latn, psp_Latn, jra_Latn, smt_Latn, ags_Latn, csw_Latn, wtk_Latn, emp_Cyrl, koi_Latn, tkr_Cyrl, amp_Latn, ymp_Latn, mfh_Latn, tdb_Deva, omw_Latn, khb_Talu, doi_Deva, gld_Cyrl, ava_Latn, chu_Latn, dnw_Latn, azo_Latn, dug_Latn, bce_Latn, kmr_Latn, kpy_Armn, abq_Cyrl, trp_Latn, ewo_Latn, the_Deva, hig_Latn, pkb_Latn, mxu_Latn, oji_Latn, tnt_Latn, mzm_Latn, mns_Cyrl, lbe_Cyrl, qvh_Latn, kmg_Latn, sps_Latn, brb_Khmr, tah_Latn, sxb_Latn, mkz_Latn, mgq_Latn, got_Goth, lns_Latn, arc_Latn, akb_Latn, skr_Latn, nsk_Cans, sml_Latn, pce_Mymr, eee_Thai, lhm_Deva, yux_Cyrl, bqm_Latn, bcc_Arab, nas_Latn, agq_Latn, xog_Latn, tsb_Latn, fub_Latn, mqj_Latn, nsk_Latn, bxr_Latn, dln_Latn, ozm_Latn, rmy_Latn, cre_Cans, kim_Cyrl, cuh_Latn, ngl_Latn, yas_Latn, bud_Latn, miy_Latn, ame_Latn, pnz_Latn, raj_Deva, enb_Latn, cmo_Khmr, saq_Latn, tpu_Khmr, eve_Cyrl, cdo_Hani |

mitigate errors introduced by the machine translation process, we filter out prompts with a BLEU score of less than 20. Figure 5 shows the average BLEU score of each language in the final dataset.

# D  PROMPTS

## D.1  MACHINE TRANSLATION

We use the prompt below for machine translation.

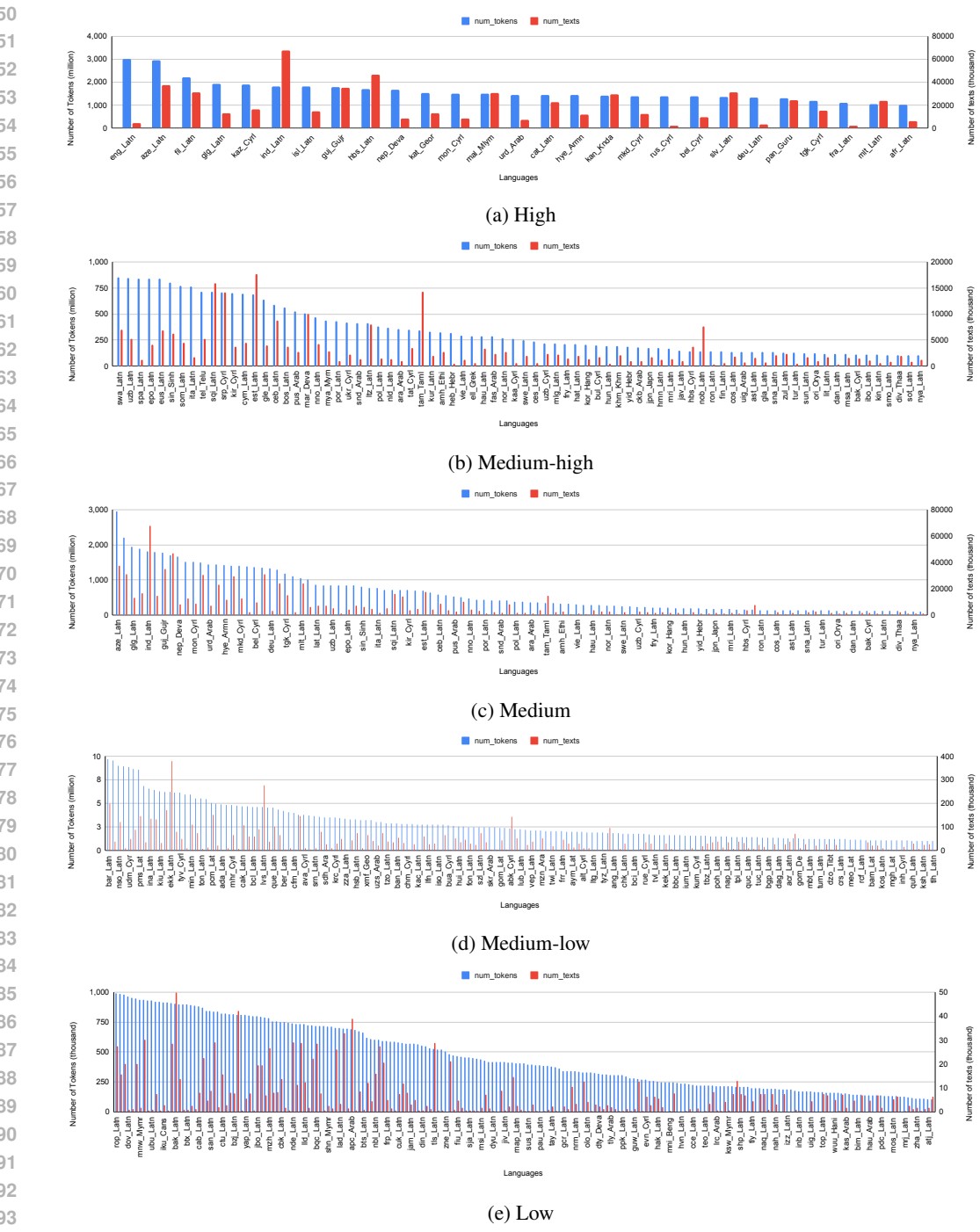

Figure 1: The number of texts and tokens of MaLA corpus in different resource groups.

```
Translate the following sentence from {src_lang} to {tgt_lang}
[{src_lang}]: {src_sent}
[{tgt_lang}]:
```

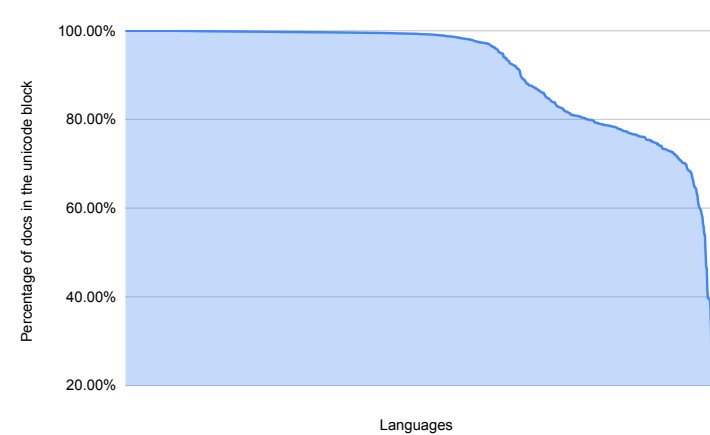

Figure 2: Unicode block distribution that measures the percentage of token counts falling into the Unicode block of each language

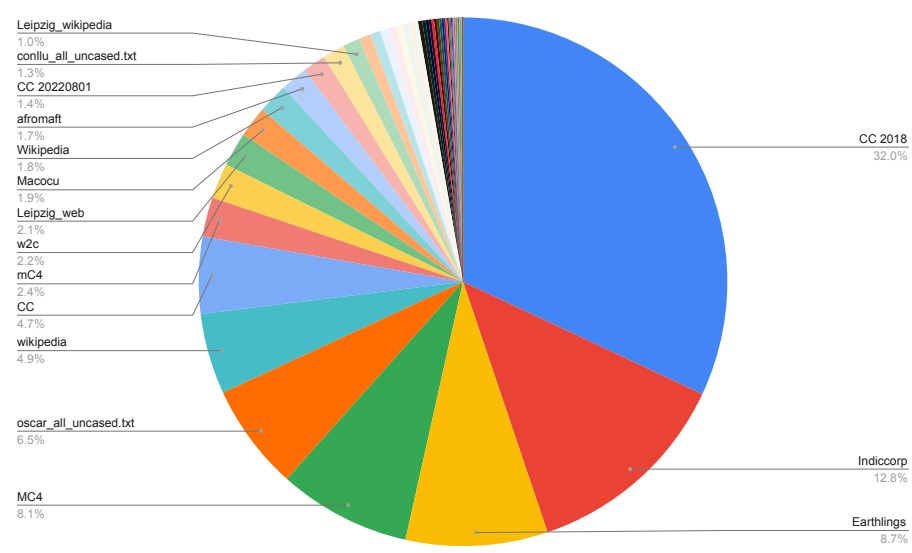

Figure 3: Data source distribution of MaLA corpus calculated by the number of documents

## D.2    TEXT CLASSIFICATION

The prompt template for SIB-200 is as follows:

```
Topic Classification: science/technology, travel, politics, sports,
    health, entertainment, geography.
{examples}
The topic of the news "${text}" is
```

For Taxi-1500, the prompt template is as follows:

```
Topic Classification: Recommendation, Faith, Description, Sin, Grace
    , Violence.
{examples}
The topic of the verse "${text}" is
```

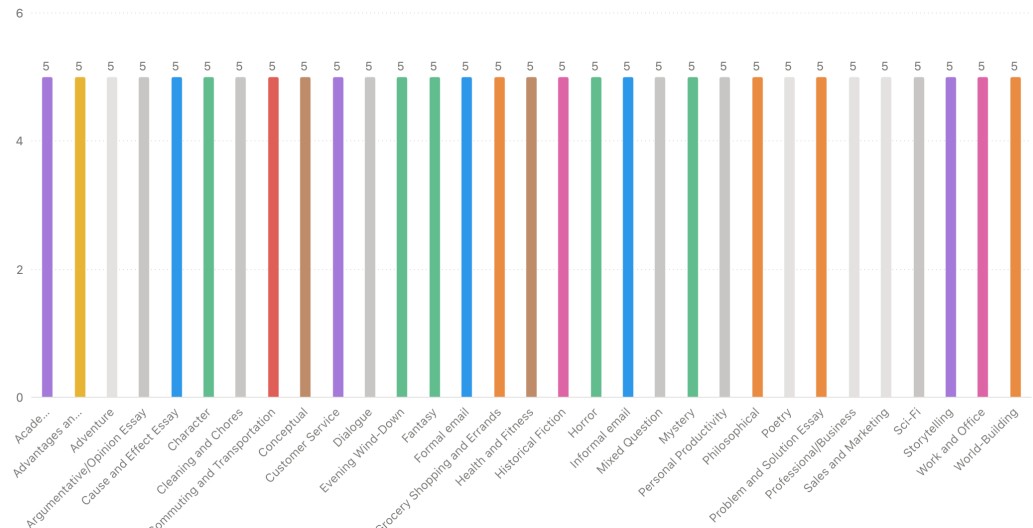

Figure 4: Writing tasks in the PolyWrite dataset.

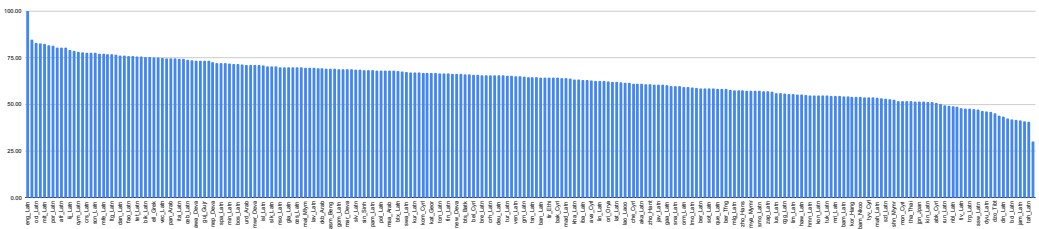

Figure 5: Mean BLEU scores per language in the PolyWrite dataset.

# E    DETAILED EVALUATION

This section presents detailed results of the evaluation. For benchmarks consisting of multiple languages, we host the results on GitHub[40]. Those benchmarks include ARC multilingual, BELEBELE, Glot500-c test set, PBC, SIB-200, Taxi-1500, FLORES200, XLSum, Aya evaluation suite, and PolyWrite. For FLORES200, XLSum, Aya evaluation suite, and PolyWrite, we also release the generated texts of all compared models.

**Evaluation Software** We use the Language Model Evaluation Harness (`lm-evaluation-harness`) framework (Gao et al., 2023) for benchmarking test sets that are already ingested in the framework. For other benchmarks, we use in-house developed evaluation scripts and other open-source implementations. In text classification tasks such as SIB-200 and Taxi-1500, the evaluation protocol involves calculating the probability of the next output token for each candidate category. These probabilities are then sorted in descending order, with the category having the highest probability being selected as the model's prediction. This process is implemented using the Transformers (Wolf et al., 2019) library. In tasks like Machine Translation and Open-Ended Generation, the vLLM library (Kwon et al., 2023) is used to accelerate inference, providing significant speed improvements. Similarly, for code generation tasks, we use a VLLM-enabled evaluation harness package (`vllm-code-harness`)[41] for the execution-tested evaluations. For massively multilingual benchmarks with more than 100 languages, we categorize languages into five groups as listed in Table 8 in Appendix B.1.

---

[40]https://github.com/MaLA-LM/emma-500/tree/main/evaluation_results

[41]https://github.com/iNeil77/vllm-code-harness

Table 9: NLL on Glot500-c test. EMMA-500 Llama 2 7B has better average performance than all baselines.

| Model | Avg | High | Medium-High | Medium | Medium-Low | Low |
|---|---|---|---|---|---|---|
| Llama 2 7B | 190.58 | 146.43 | 176.30 | 205.86 | 210.41 | 196.45 |
| Llama 2 7B Chat | 218.87 | 173.67 | 204.78 | 239.18 | 240.97 | 223.86 |
| CodeLlama 2 7B | 193.96 | 146.43 | 180.20 | 210.09 | 212.70 | 200.60 |
| LLaMAX Llama 2 7B | 187.37 | 108.58 | 142.84 | 197.74 | 212.22 | 203.83 |
| LLaMAX Llama 2 7B Alpaca | 169.12 | 94.84 | 123.90 | 173.54 | 193.83 | 187.34 |
| MaLA-500 Llama 2 10B v1 | 155.62 | 127.51 | 153.93 | 173.02 | 166.01 | 158.12 |
| MaLA-500 Llama 2 10B v2 | 151.25 | 123.82 | 147.69 | 167.46 | 161.20 | 155.13 |
| TowerBase Llama 2 7B | 192.98 | 150.41 | 180.19 | 209.12 | 212.18 | 198.89 |
| TowerInstruct Llama 2 7B | 199.44 | 157.33 | 186.42 | 216.92 | 218.82 | 204.93 |
| Occiglot Mistral 7B v0.1 | 191.11 | 159.48 | 185.53 | 209.26 | 207.67 | 194.07 |
| Occiglot Mistral 7B v0.1 Instruct | 193.83 | 162.31 | 188.20 | 212.41 | 210.81 | 196.58 |
| BLOOM 7B | 202.95 | 160.33 | 195.01 | 216.15 | 220.46 | 206.89 |
| BLOOMZ 7B | 217.32 | 178.12 | 210.99 | 235.53 | 235.60 | 220.65 |
| mGPT | 340.37 | 311.14 | 337.29 | 388.97 | 367.45 | 343.14 |
| XGLM 7.5B | 205.07 | 199.08 | 210.22 | 225.53 | 214.92 | 205.67 |
| Yayi 7B | 226.67 | 181.48 | 217.34 | 243.42 | 246.58 | 231.65 |
| Llama 3 8B | 156.36 | 102.78 | 129.36 | 153.11 | 167.26 | 173.56 |
| Llama 3.1 8B | 154.59 | 101.23 | 127.57 | 150.87 | 164.81 | 172.05 |
| Gemma 7B | 692.25 | 583.39 | 721.77 | 817.40 | 729.60 | 689.60 |
| Gemma 2 9B | 320.81 | 348.26 | 351.08 | 380.49 | 338.00 | 303.62 |
| Qwen 2 7B | 188.55 | 132.47 | 171.50 | 200.26 | 210.21 | 196.78 |
| Qwen 1.5 7B | 195.52 | 141.37 | 181.41 | 212.10 | 217.16 | 202.46 |
| EMMA-500 Llama 2 7B | 112.20 | 81.78 | 100.89 | 122.53 | 99.28 | 109.25 |

## E.1 INTRINSIC EVALUATION

For intrinsic evaluation, we compute the negative log-likelihood (NLL) of the test text given by the tested LLMs instead of using length-normalized perplexity due to different text tokenization schemes across models. We concatenate the test set as a single input and run a sliding-window approach.[42] To make a comparison across models, we use the Glot500-c test set, which covers 534 evaluated languages. We also test on the Parallel Bible Corpus (PBC) which could yield NLL more comparable across languages. Table 9 and Table 10 show the intrinsic evaluation results on Glot500-c test and PBC, respectively. As shown, our model attains lower NLL compared to all other models on both test sets and languages with different resource availability. This suggests that the test sets are more similar to the underlying training data of our model and it can be interpreted that EMMA-500 has learned to compress massive multilingual text more efficiently.

## E.2 SUMMARIZATION

XL-Sum (Hasan et al., 2021) is a large-scale multilingual abstractive summarization dataset that covers 44 languages. To evaluate the quality of the generated summaries, we use ROUGE-L (Lin, 2004) and BERTScore (Zhang et al., 2019) as evaluation metrics. ROUGE-L measures the longest common subsequence (LCS) between the reference and generated summaries. Recall and precision are calculated by dividing the LCS length by the reference length and the generated summary length, respectively. An F-score is then used to combine these two aspects into a single metric. BERTScore computes the semantic similarity between summaries by leveraging contextual embeddings from pre-trained language models. Specifically, we use the `bert-base-multilingual-cased`[43] model for BERTScore to accommodate multiple languages.

The evaluation results are presented in Table 11, we observe that our model, EMMA-500, performs comparably to other Llama2-based models like TowerInstruct Llama 2 7B. While EMMA-500 shows strength in certain language categories, it does not significantly outperform these models overall. This suggests that, although EMMA-500 is effective in multilingual summarization tasks, there is room for improvement to achieve more consistent performance across all languages. It is also important to note that this test set includes only 44 languages, limiting the evaluation of EMMA-500's capabilities in low-resource languages.

---

[42] https://huggingface.co/docs/transformers/en/perplexity
[43] https://huggingface.co/google-bert/bert-base-multilingual-cased

Table 10: NLL on PBC. EMMA-500 Llama 2 7B has better average performance than all baselines.

| Model | Avg | High | Medium-High | Medium | Medium-Low | Low |
|---|---|---|---|---|---|---|
| Llama 2 7B | 122.10 | 91.30 | 99.41 | 112.31 | 133.08 | 135.34 |
| Llama 2 7B Chat | 139.14 | 108.40 | 115.78 | 129.79 | 149.54 | 152.82 |
| CodeLlama 2 7B | 123.98 | 93.27 | 101.83 | 113.47 | 134.65 | 137.52 |
| LLaMAX Llama 2 7B | 117.39 | 69.41 | 79.06 | 103.74 | 131.90 | 138.03 |
| LLaMAX Llama 2 7B Alpaca | 107.81 | 60.05 | 69.36 | 93.39 | 122.44 | 128.77 |
| MaLA-500 Llama 2 10B v1 | 103.20 | 94.04 | 98.60 | 100.53 | 105.04 | 107.65 |
| MaLA-500 Llama 2 10B v2 | 101.67 | 92.42 | 96.30 | 98.88 | 103.34 | 106.62 |
| TowerBase Llama 2 7B | 123.70 | 93.64 | 101.44 | 114.72 | 134.41 | 136.77 |
| TowerInstruct Llama 2 7B | 127.30 | 98.21 | 105.17 | 118.65 | 137.53 | 140.28 |
| Occiglot Mistral 7B v0.1 | 121.64 | 95.15 | 101.86 | 114.37 | 131.49 | 132.93 |
| Occiglot Mistral 7B v0.1 Instruct | 123.18 | 96.86 | 103.41 | 115.88 | 132.98 | 134.48 |
| BLOOM 7B | 129.55 | 96.62 | 111.22 | 115.33 | 138.19 | 143.03 |
| BLOOMZ 7B | 137.72 | 107.03 | 119.89 | 125.85 | 145.27 | 150.95 |
| mGPT | 225.14 | 211.35 | 203.84 | 219.75 | 229.91 | 239.82 |
| XGLM 7.5B | 131.31 | 116.86 | 117.69 | 125.15 | 136.47 | 140.53 |
| Yayi 7B | 143.80 | 108.79 | 123.20 | 130.60 | 152.37 | 158.73 |
| Llama 3 8B | 102.55 | 64.20 | 71.82 | 85.22 | 114.79 | 121.29 |
| Llama 3.1 8B | 101.43 | 62.98 | 70.68 | 83.83 | 113.36 | 120.33 |
| Gemma 7B | 460.86 | 399.22 | 427.14 | 468.66 | 463.11 | 483.29 |
| Gemma 2 9B | 197.41 | 200.76 | 192.07 | 197.88 | 196.56 | 202.69 |
| Qwen 2 7B | 120.44 | 83.34 | 94.87 | 107.84 | 133.38 | 136.52 |
| Qwen 1.5 7B | 124.02 | 89.36 | 100.55 | 113.08 | 135.54 | 139.13 |
| EMMA-500 Llama 2 7B | 68.11 | 50.12 | 55.62 | 64.78 | 60.53 | 65.68 |

Table 11: Results on XL-Sum (ROUGE-L/BERTScore).

| Model | Avg | High | Medium-High | Medium | Medium-Low | Low |
|---|---|---|---|---|---|---|
| Llama 2 7B | 0.07/0.67 | 0.07/0.67 | 0.07/0.66 | 0.06/0.65 | 0.10/0.62 | 0.08/0.71 |
| Llama 2 7B Chat | 0.09/0.68 | 0.09/0.68 | 0.08/0.68 | 0.08/0.67 | 0.12/0.64 | 0.10/0.74 |
| CodeLlama 2 7B | 0.07/0.66 | 0.07/0.65 | 0.07/0.66 | 0.06/0.63 | 0.10/0.64 | 0.08/0.71 |
| LLaMAX Llama 2 7B | 0.05/0.65 | 0.05/0.65 | 0.05/0.65 | 0.05/0.61 | 0.07/0.62 | 0.06/0.69 |
| LLaMAX Llama 2 7B Alpaca | 0.10/0.69 | 0.10/0.70 | 0.09/0.69 | 0.09/0.68 | 0.13/0.65 | 0.12/0.74 |
| MaLA-500 Llama 2 10B v1 | 0.05/0.64 | 0.06/0.64 | 0.05/0.64 | 0.05/0.61 | 0.07/0.64 | 0.06/0.68 |
| MaLA-500 Llama 2 10B v2 | 0.05/0.64 | 0.06/0.64 | 0.05/0.64 | 0.05/0.61 | 0.07/0.64 | 0.06/0.69 |
| Yayi Llama 2 7B | 0.08/0.67 | 0.09/0.67 | 0.07/0.67 | 0.07/0.66 | 0.11/0.61 | 0.09/0.71 |
| TowerBase Llama 2 7B | 0.08/0.67 | 0.08/0.67 | 0.07/0.67 | 0.07/0.65 | 0.10/0.63 | 0.09/0.71 |
| TowerInstruct Llama 2 7B | 0.09/0.68 | 0.09/0.69 | 0.09/0.69 | 0.08/0.66 | 0.12/0.64 | 0.09/0.73 |
| Occiglot Mistral 7B v0.1 | 0.07/0.66 | 0.08/0.66 | 0.07/0.66 | 0.07/0.64 | 0.10/0.63 | 0.09/0.71 |
| Occiglot Mistral 7B v0.1 Instruct | 0.08/0.67 | 0.09/0.67 | 0.08/0.67 | 0.08/0.64 | 0.10/0.64 | 0.09/0.73 |
| BLOOM 7B | 0.07/0.65 | 0.07/0.65 | 0.07/0.65 | 0.06/0.62 | 0.10/0.58 | 0.08/0.71 |
| BLOOMZ 7B | 0.11/0.70 | 0.14/0.69 | 0.09/0.70 | 0.10/0.69 | 0.12/0.66 | 0.15/0.74 |
| mGPT | 0.04/0.60 | 0.05/0.60 | 0.04/0.60 | 0.02/0.56 | 0.07/0.59 | 0.04/0.63 |
| mGPT-13B | 0.05/0.62 | 0.06/0.62 | 0.05/0.63 | 0.04/0.59 | 0.08/0.59 | 0.05/0.67 |
| Yayi 7B | 0.12/0.70 | 0.14/0.69 | 0.10/0.69 | 0.11/0.69 | 0.12/0.66 | 0.15/0.75 |
| Llama 3 8B | 0.08/0.67 | 0.08/0.67 | 0.08/0.67 | 0.08/0.65 | 0.10/0.65 | 0.11/0.72 |
| Llama 3.1 8B | 0.09/0.67 | 0.08/0.66 | 0.08/0.67 | 0.08/0.65 | 0.09/0.66 | 0.10/0.72 |
| Gemma 2 9B | 0.07/0.65 | 0.07/0.64 | 0.07/0.66 | 0.07/0.63 | 0.09/0.64 | 0.09/0.71 |
| Gemma 7B | 0.07/0.65 | 0.07/0.63 | 0.06/0.65 | 0.07/0.67 | 0.07/0.60 | 0.08/0.63 |
| Qwen 1.5 7B | 0.10/0.69 | 0.11/0.69 | 0.09/0.69 | 0.10/0.68 | 0.11/0.65 | 0.11/0.74 |
| Qwen 2 7B | 0.10/0.69 | 0.12/0.69 | 0.09/0.69 | 0.10/0.68 | 0.11/0.65 | 0.11/0.74 |
| EMMA-500 Llama 2 7B | 0.09/0.67 | 0.08/0.66 | 0.08/0.67 | 0.08/0.69 | 0.11/0.66 | 0.10/0.67 |

We assess the models' commonsense reasoning ability using three multilingual benchmarks. XCOPA (Ponti et al., 2020) is a dataset covering 11 languages that focuses on causal commonsense reasoning across multiple languages. XStoryCloze (Lin et al., 2022) is derived from the English StoryCloze dataset (Mostafazadeh et al., 2017) and translated into 10 non-English languages, testing commonsense reasoning within a story. In this task, the system must choose the correct ending for a four-sentence narrative. XWinograd (Tikhonov & Ryabinin, 2021) is a multilingual collection of Winograd Schemas (Levesque et al., 2012) available in six languages, aimed at evaluating cross-lingual commonsense reasoning.

### E.3 COMMONSENSE REASONING

We perform zero-shot evaluations. Accuracy is used as the evaluation metric. We categorize languages into different resource groups based on language availability, accessibility, and possible corpus size. For XCOPA, we have three groups, i.e., high-resource (Italian, Turkish, Vietnamese, and Chinese), medium-resource (Swahili due to its regional influence, Estonian, Haitian Creole, Indonesian, Thai, and Tamil), and low-resource languages (Cusco-Collao Quechua).[44] For XStoryCloze, the resource group is high-resource (Arabic, English, Spanish, Russian, and Chinese), medium (Hindi, Indonesian, Swahili, and Telugu), and low (Basque and Burmese). For XWinograd, there is only one group for high-resource languages. Table 12 shows the evaluation results of zero-shot commonsense reasoning.

Compared with Llama 2-based models on XCOPA, our model improves the average performance by a large margin—up to a 5% increase when compared with the best-performing TowerInstruct based on Llama 2 7B.[45] Our model also outperforms all the multilingual LLMs. Recent LLMs such as Gemma and Llama 3 have stronger performance than Llama 2 models, and Gemma 2 9B performs the best. However, our model achieves better performance than Qwen, Llama 3, and 3.1. We gain similar results on XStoryCloze, outperforming all the models except Gemma 2 9B. As for XWinograd, a multilingual benchmark with high-resource only, our model achieves improved performance than Llama 2, despite not being as good as Tower models, which target high-resource languages. However, our model is comparable to other multilingual LLMs. For low-resource languages, our model outperforms all the compared LLMs except LLaMAX Llama 2 7B Alpaca on XCOPA, where we achieve the same accuracy as it.

### E.5 NATURAL LANGUAGE INFERENCE

We evaluate on the XNLI (Cross-lingual Natural Language Inference) benchmark (Conneau et al., 2018) where sentence pairs in different languages need to be classified as entailment, contradiction, or neutral. We categorize the languages in XNLI into 3 groups, i.e., high-resource (German, English, Spanish, French, Russian, and Chinese), medium-resource (Arabic, Bulgarian, Greek, Hindi, Turkish, and Vietnamese), and low-resource (Swahili, Thai, and Urdu).[46] Table 17 shows the aggregated accuracy. Our model outperforms most baselines including Llama 2-based models and multilingual LLMs. We achieve the second-best average accuracy, slightly behind the Llama 3.1 8B model. On the low-resource end, we perform the second-best, slightly behind the Gemma 2 9B model.

### E.6 MATH

We evaluate the ability of LLMs to solve grade-school math problems across multiple languages on the MGSM (Multilingual Grade School Math) benchmark (Shi et al., 2022). MGSM is an extension of the GSM8K (Grade School Math 8K) dataset (Cobbe et al., 2021) by translating 250 of the original GSM8K problems into ten languages. We also split these ten languages into three groups, i.e., high-resource (Spanish, French, German, Russian, Chinese, and Japanese), medium-resource (Thai, Swahili, and Bengali), and low-resource (Telugu). Table 19 shows the results for 3-shot prompting with a flexible match to obtain the answers in model generation. We evaluate all the models by directly prompting the questions (denoted as direct) and the questions with answers followed by Chain-of-Thoughts prompt in the same languages as the subset being evaluated (denoted as CoT)

---

[44]Note that there is no perfect categorization for language resource groups.

[45]The biggest improvements are on medium-resource languages. However, we note that the resource categorization is not perfect.

[46]Again, the categorization is not perfect.

Table 12: 0-shot results (ACC) on commonsense reasoning: XCOPA, XStoryCloze, and XWinograd. EMMA-500 Llama 2 7B has better average performance than Llama 2 models and multilingual LLMs on XCOPA, XStoryCloze, and comparable performance on XWinograd.

| Model | XCOPA | | | | XStoryCloze | | | | XWinograd |
|---|---|---|---|---|---|---|---|---|---|
| | Avg | High | Medium | Low | Avg | High | Medium | Low | Avg |
| Llama 2 7B | 0.5667 | 0.6210 | 0.5390 | 0.5160 | 0.5755 | 0.6338 | 0.5445 | 0.4921 | 0.7247 |
| Llama 2 7B Chat | 0.5585 | 0.6125 | 0.5313 | 0.5060 | 0.5841 | 0.6480 | 0.5477 | 0.4974 | 0.6945 |
| CodeLlama 2 7B | 0.5469 | 0.5870 | 0.5253 | 0.5160 | 0.5568 | 0.6068 | 0.5233 | 0.4990 | 0.7092 |
| LLaMAX Llama 2 7B | 0.5438 | 0.5550 | 0.5413 | 0.5140 | 0.6036 | 0.6434 | 0.5882 | 0.5347 | 0.6749 |
| LLaMAX Llama 2 7B Alpaca | 0.5660 | 0.5980 | 0.5517 | 0.5240 | 0.6383 | 0.6908 | 0.6201 | 0.5433 | 0.6986 |
| MaLA-500 Llama 2 10B v1 | 0.5309 | 0.5355 | 0.5327 | 0.5020 | 0.5307 | 0.5815 | 0.4922 | 0.4808 | 0.6589 |
| MaLA-500 Llama 2 10B v2 | 0.5309 | 0.5355 | 0.5327 | 0.5020 | 0.5307 | 0.5815 | 0.4922 | 0.4808 | 0.6589 |
| YaYi Llama 2 7B | 0.5671 | 0.6210 | 0.5413 | 0.5060 | 0.5842 | 0.6498 | 0.5491 | 0.4904 | 0.7450 |
| TowerBase Llama 2 7B | 0.5633 | 0.6250 | 0.5290 | 0.5220 | 0.5778 | 0.6435 | 0.5367 | 0.4957 | 0.7429 |
| TowerInstruct Llama 2 7B | 0.5705 | 0.6290 | 0.5400 | 0.5200 | 0.5924 | 0.6683 | 0.5453 | 0.4970 | 0.7400 |
| Occiglot Mistral 7B v0.1 | 0.5667 | 0.6280 | 0.5337 | 0.5200 | 0.5810 | 0.6518 | 0.5328 | 0.5003 | 0.7461 |
| Occiglot Mistral 7B v0.1 Instruct | 0.5655 | 0.6285 | 0.5297 | 0.5280 | 0.5939 | 0.6694 | 0.5433 | 0.5063 | 0.7293 |
| BLOOM 7B | 0.5689 | 0.5995 | 0.5587 | 0.5080 | 0.5930 | 0.6199 | 0.5905 | 0.5308 | 0.7013 |
| BLOOMZ 7B | 0.5487 | 0.5635 | 0.5460 | 0.5060 | 0.5712 | 0.6114 | 0.5582 | 0.4967 | 0.6795 |
| mGPT | 0.5504 | 0.5710 | 0.5440 | 0.5060 | 0.5443 | 0.5496 | 0.5453 | 0.5291 | 0.5969 |
| mGPT 13B | 0.5618 | 0.5975 | 0.5513 | 0.5060 | 0.5644 | 0.5776 | 0.5635 | 0.5331 | 0.6359 |
| XGLM 7.5B | 0.6064 | 0.6400 | 0.6037 | 0.4880 | 0.6075 | 0.6242 | 0.6036 | 0.5738 | 0.6884 |
| YaYi 7B | 0.5664 | 0.5955 | 0.5550 | 0.5180 | 0.6067 | 0.6490 | 0.5940 | 0.5265 | 0.6979 |
| Llama 3 8B | 0.6171 | 0.6835 | 0.5903 | 0.5120 | 0.6341 | 0.6850 | 0.6203 | 0.5344 | 0.7684 |
| Llama 3.1 8B | 0.6171 | 0.6930 | 0.5880 | 0.4880 | 0.6358 | 0.6866 | 0.6209 | 0.5387 | 0.7552 |
| Gemma 7B | 0.6364 | 0.7035 | 0.6143 | 0.5000 | 0.6501 | 0.6946 | 0.6449 | 0.5493 | 0.7741 |
| Gemma 2 9B | 0.6633 | 0.7340 | 0.6427 | 0.5040 | 0.6767 | 0.7247 | 0.6669 | 0.5764 | 0.8007 |
| Qwen 2 7B | 0.6031 | 0.6865 | 0.5640 | 0.5040 | 0.6146 | 0.6945 | 0.5697 | 0.5050 | 0.7644 |
| Qwen 1.5 7B | 0.5944 | 0.6685 | 0.5590 | 0.5100 | 0.5985 | 0.6662 | 0.5604 | 0.5056 | 0.7259 |
| EMMA-500 Llama 2 7B | 0.6311 | 0.6660 | 0.6257 | 0.5240 | 0.6638 | 0.6892 | 0.6573 | 0.6132 | 0.7280 |

(Shi et al., 2022). The results show that Llama 2 7B is a weak model in the MGSM math task. The base model and its variants failed in most settings. Multilingual LLMs such as BLOOM, XGLM, and YaYi are also subpar at this task. Recent LLMs like Llama 3, Qwen, and Gemma obtain reasonable performance, and Qwen series models are the best. Our model improves the Llama 2 7B model remarkably in both prompt strategies. For direct prompting, our model has an average accuracy of 0.1702, 7% higher than the Llama 2 7B chat model. For CoT-based prompting, our model increases the score of the Llama 2 7B chat model from 0.1 to 0.3 (20% higher) and slightly outperforms Llama 3 and 3.1 8B models.

### E.7 MACHINE READING COMPREHENSION

BELEBELE (Bandarkar et al., 2023) is a machine reading comprehension dataset covering 122 languages including high- and low-resource languages. Each question offers four multiple-choice answers based on a short passage from the FLORES-200 dataset. This benchmark is very challenging, even the English version of it presents remarkable challenges for advanced models. Table 22 shows the zero-shot results in different resource groups. Our continual pre-training improves the Llama 2 7B base model. But Llama 2 7 B-based models mostly fail in this task and get quasi-random results. Recent advanced models like Llama 3.1 and Qwen 2 get reasonable results.

We then move to a more challenging task, the ARC multilingual test, which is a machine-translated benchmark (Lai et al., 2023) from the ARC dataset (Clark et al., 2018) that contains English science exam questions for multiple grade levels. We test the five-shot performance, with results shown in Table 23. The evaluation results show a similar pattern to BELEBELE that EMMA-500 improves Llama 2 but the 7B model is not capable of this challenging task, all Llama2 7B-based models obtain close to random results, and recent advances like Llama 3, Gemma, and Qwen get much better results.

### E.8 CODE GENERATION

We conduct code generation evaluations on the Multipl-E (Cassano et al., 2022) benchmark in the interest of measuring the effects of massively multilingual continual pre-training on a model's code generation utility and detecting if any catastrophic forgetting (Luo et al., 2023) has occurred on this front. Importantly, this also has implications for a model's reasoning (Yang et al., 2024b) and entity-tracking abilities (Kim et al., 2024).

Table 13: 0-shot results (ACC) on XCOPA in all languages

| Model | Avg | et-acc | stderr | ht-acc | stderr | id-acc | stderr | it-acc | stderr | qu-acc | stderr | sw-acc | stderr | ta-acc | stderr | th-acc | stderr | tr-acc | stderr | vi-acc | stderr | zh-acc | stderr |
|---|---|---|---|---|---|---|---|---|---|---|---|---|---|---|---|---|---|---|---|---|---|---|---|
| Llama 2 7B | 0.5667 | 0.4860 | 0.0224 | 0.5060 | 0.0224 | 0.6240 | 0.0217 | 0.6580 | 0.0212 | 0.5160 | 0.0224 | 0.5220 | 0.0224 | 0.5340 | 0.0223 | 0.5620 | 0.0222 | 0.5480 | 0.0223 | 0.6280 | 0.0216 | 0.6500 | 0.0214 |
| Llama 2 7B Chat | 0.5585 | 0.4780 | 0.0224 | 0.5080 | 0.0224 | 0.6240 | 0.0217 | 0.6720 | 0.0210 | 0.5060 | 0.0224 | 0.5220 | 0.0224 | 0.5060 | 0.0224 | 0.5500 | 0.0223 | 0.5520 | 0.0223 | 0.6120 | 0.0218 | 0.6140 | 0.0218 |
| CodeLlama 2 7B | 0.5469 | 0.4680 | 0.0223 | 0.5180 | 0.0224 | 0.5740 | 0.0221 | 0.6300 | 0.0216 | 0.5160 | 0.0224 | 0.4880 | 0.0224 | 0.5800 | 0.0221 | 0.5720 | 0.0221 | 0.5300 | 0.0223 | 0.5580 | 0.0222 | 0.6230 | 0.0217 |
| LLaMAX Llama 2 7B | 0.5438 | 0.4920 | 0.0224 | 0.5260 | 0.0224 | 0.5380 | 0.0223 | 0.5260 | 0.0224 | 0.5140 | 0.0224 | 0.5400 | 0.0223 | 0.5800 | 0.0221 | 0.5720 | 0.0221 | 0.5300 | 0.0223 | 0.5300 | 0.0223 | 0.6340 | 0.0216 |
| LLaMAX Llama 2 7B Alpaca | 0.5660 | 0.5120 | 0.0224 | 0.5420 | 0.0223 | 0.5720 | 0.0221 | 0.6100 | 0.0218 | 0.5240 | 0.0224 | 0.5700 | 0.0224 | 0.5700 | 0.0223 | 0.5520 | 0.0223 | 0.5520 | 0.0223 | 0.5600 | 0.0220 | 0.6780 | 0.0209 |
| MaLA-500 Llama 2 10B v1 | 0.5309 | 0.4860 | 0.0224 | 0.5340 | 0.0223 | 0.5300 | 0.0223 | 0.5940 | 0.0220 | 0.5020 | 0.0224 | 0.5280 | 0.0223 | 0.5760 | 0.0221 | 0.5420 | 0.0223 | 0.5160 | 0.0224 | 0.5240 | 0.0224 | 0.5080 | 0.0224 |
| MaLA-500 Llama 2 10B v2 | 0.5309 | 0.4860 | 0.0224 | 0.5340 | 0.0223 | 0.5300 | 0.0223 | 0.5940 | 0.0220 | 0.5020 | 0.0224 | 0.5280 | 0.0223 | 0.5760 | 0.0221 | 0.5420 | 0.0223 | 0.5160 | 0.0224 | 0.5240 | 0.0224 | 0.5080 | 0.0224 |
| YaYi Llama 2 7B | 0.5671 | 0.4880 | 0.0224 | 0.5080 | 0.0224 | 0.6260 | 0.0217 | 0.6700 | 0.0210 | 0.5060 | 0.0224 | 0.5320 | 0.0223 | 0.5520 | 0.0223 | 0.5940 | 0.0220 | 0.5580 | 0.0220 | 0.6320 | 0.0216 | 0.6280 | 0.0216 |
| TowerBase Llama 2 7B | 0.5633 | 0.4600 | 0.0223 | 0.5020 | 0.0224 | 0.6020 | 0.0219 | 0.7080 | 0.0204 | 0.5220 | 0.0224 | 0.5060 | 0.0224 | 0.5440 | 0.0223 | 0.5600 | 0.0222 | 0.5380 | 0.0223 | 0.5920 | 0.0220 | 0.6620 | 0.0212 |
| TowerInstruct Llama 2 7B | 0.5705 | 0.4880 | 0.0224 | 0.5160 | 0.0224 | 0.6260 | 0.0217 | 0.7100 | 0.0203 | 0.5200 | 0.0224 | 0.5100 | 0.0224 | 0.5420 | 0.0223 | 0.5600 | 0.0222 | 0.5640 | 0.0222 | 0.5860 | 0.0220 | 0.6740 | 0.0210 |
| Occiglot Mistral 7B v0.1 | 0.5667 | 0.4720 | 0.0223 | 0.5140 | 0.0224 | 0.5700 | 0.0222 | 0.7460 | 0.0195 | 0.5200 | 0.0224 | 0.5160 | 0.0224 | 0.5720 | 0.0221 | 0.5580 | 0.0222 | 0.5440 | 0.0223 | 0.5520 | 0.0223 | 0.6700 | 0.0210 |
| Occiglot Mistral 7B v0.1 Instruct | 0.5655 | 0.4680 | 0.0223 | 0.5100 | 0.0224 | 0.5840 | 0.0221 | 0.7380 | 0.0197 | 0.5280 | 0.0223 | 0.5660 | 0.0222 | 0.5660 | 0.0222 | 0.5900 | 0.0223 | 0.5600 | 0.0223 | 0.5580 | 0.0222 | 0.6560 | 0.0213 |
| BLOOM 7B | 0.5689 | 0.4820 | 0.0224 | 0.5080 | 0.0224 | 0.6980 | 0.0206 | 0.5280 | 0.0223 | 0.5080 | 0.0223 | 0.5180 | 0.0224 | 0.5920 | 0.0220 | 0.5540 | 0.0223 | 0.5100 | 0.0224 | 0.7080 | 0.0204 | 0.6520 | 0.0213 |
| BLOOMZ 7B | 0.5487 | 0.4920 | 0.0224 | 0.5400 | 0.0223 | 0.6060 | 0.0219 | 0.5140 | 0.0224 | 0.5060 | 0.0224 | 0.5340 | 0.0223 | 0.5740 | 0.0223 | 0.5300 | 0.0223 | 0.5220 | 0.0224 | 0.5980 | 0.0219 | 0.6200 | 0.0217 |
| mGPT | 0.5504 | 0.5300 | 0.0223 | 0.4980 | 0.0224 | 0.5880 | 0.0220 | 0.5820 | 0.0221 | 0.5060 | 0.0224 | 0.5640 | 0.0222 | 0.5320 | 0.0223 | 0.5520 | 0.0223 | 0.5600 | 0.0223 | 0.6020 | 0.0219 | 0.5400 | 0.0223 |
| mGPT 13B | 0.5618 | 0.5080 | 0.0224 | 0.5180 | 0.0224 | 0.6260 | 0.0217 | 0.6080 | 0.0219 | 0.4820 | 0.0224 | 0.5800 | 0.0221 | 0.5480 | 0.0223 | 0.5280 | 0.0223 | 0.5680 | 0.0222 | 0.6340 | 0.0216 | 0.5800 | 0.0221 |
| XGLM 7.5B | 0.6064 | 0.6140 | 0.0218 | 0.5740 | 0.0221 | 0.6940 | 0.0206 | 0.6360 | 0.0215 | 0.4880 | 0.0224 | 0.6200 | 0.0219 | 0.5460 | 0.0223 | 0.5940 | 0.0220 | 0.5460 | 0.0223 | 0.7020 | 0.0205 | 0.6380 | 0.0215 |
| YaYi 7B | 0.5664 | 0.5040 | 0.0224 | 0.5300 | 0.0223 | 0.6340 | 0.0216 | 0.5180 | 0.0224 | 0.5180 | 0.0224 | 0.5540 | 0.0223 | 0.5620 | 0.0222 | 0.5460 | 0.0223 | 0.5200 | 0.0224 | 0.6640 | 0.0211 | 0.6800 | 0.0209 |
| Llama 3 8B | 0.6171 | 0.5340 | 0.0223 | 0.5280 | 0.0223 | 0.7140 | 0.0202 | 0.7160 | 0.0202 | 0.5120 | 0.0224 | 0.5120 | 0.0224 | 0.5800 | 0.0221 | 0.6000 | 0.0219 | 0.5860 | 0.0220 | 0.6240 | 0.0217 | 0.6780 | 0.0209 |
| Llama 3.1 8B | 0.6171 | 0.5300 | 0.0223 | 0.5380 | 0.0223 | 0.7160 | 0.0202 | 0.7260 | 0.0200 | 0.4880 | 0.0224 | 0.5540 | 0.0223 | 0.5780 | 0.0221 | 0.6180 | 0.0218 | 0.5780 | 0.0221 | 0.7240 | 0.0200 | 0.7040 | 0.0204 |
| Gemma 7B | 0.6364 | 0.5920 | 0.0220 | 0.5480 | 0.0223 | 0.7200 | 0.0201 | 0.7280 | 0.0199 | 0.5000 | 0.0224 | 0.6060 | 0.0219 | 0.6160 | 0.0218 | 0.6040 | 0.0219 | 0.6560 | 0.0213 | 0.7420 | 0.0196 | 0.6880 | 0.0207 |
| Gemma 2 9B | 0.6633 | 0.6380 | 0.0215 | 0.5280 | 0.0223 | 0.7780 | 0.0186 | 0.7580 | 0.0192 | 0.5040 | 0.0224 | 0.6380 | 0.0215 | 0.6380 | 0.0215 | 0.6040 | 0.0215 | 0.6740 | 0.0210 | 0.7660 | 0.0190 | 0.7380 | 0.0197 |
| Qwen 2 7B | 0.6031 | 0.5080 | 0.0224 | 0.5080 | 0.0224 | 0.7060 | 0.0204 | 0.7140 | 0.0202 | 0.5040 | 0.0224 | 0.5240 | 0.0224 | 0.5280 | 0.0223 | 0.6100 | 0.0218 | 0.5720 | 0.0221 | 0.6940 | 0.0206 | 0.7660 | 0.0190 |
| Qwen 1.5 7B | 0.5944 | 0.5200 | 0.0224 | 0.5300 | 0.0223 | 0.6440 | 0.0214 | 0.6520 | 0.0213 | 0.5100 | 0.0224 | 0.5580 | 0.0222 | 0.5760 | 0.0222 | 0.5880 | 0.0220 | 0.5600 | 0.0220 | 0.6920 | 0.0207 | 0.7420 | 0.0196 |
| EMMA-500 Llama 2 7B | 0.6311 | 0.6140 | 0.0218 | 0.5800 | 0.0221 | 0.7420 | 0.0196 | 0.6940 | 0.0206 | 0.5240 | 0.0224 | 0.6620 | 0.0212 | 0.6000 | 0.0219 | 0.5560 | 0.0222 | 0.6200 | 0.0217 | 0.7020 | 0.0205 | 0.6480 | 0.0214 |

Table 14: 0-shot results (ACC) on XStoryCloze in all languages

| Model | Avg | ar-acc | stderr | en-acc | stderr | es-acc | stderr | eu-acc | stderr | hi-acc | stderr | id-acc | stderr | my-acc | stderr | ru-acc | stderr | sw-acc | stderr | te-acc | stderr | zh-acc | stderr |
|---|---|---|---|---|---|---|---|---|---|---|---|---|---|---|---|---|---|---|---|---|---|---|---|
| Llama 2 7B | 0.5755 | 0.4990 | 0.0129 | 0.7704 | 0.0108 | 0.6737 | 0.0121 | 0.5036 | 0.0129 | 0.5374 | 0.0128 | 0.5923 | 0.0126 | 0.4805 | 0.0129 | 0.6300 | 0.0124 | 0.5050 | 0.0129 | 0.5433 | 0.0128 | 0.5956 | 0.0126 |
| Llama 2 7B Chat | 0.5841 | 0.5050 | 0.0129 | 0.7869 | 0.0105 | 0.6711 | 0.0121 | 0.5083 | 0.0129 | 0.5407 | 0.0128 | 0.5963 | 0.0126 | 0.4864 | 0.0129 | 0.6552 | 0.0122 | 0.5202 | 0.0129 | 0.5334 | 0.0128 | 0.6221 | 0.0125 |
| CodeLlama 2 7B | 0.5568 | 0.5010 | 0.0129 | 0.7148 | 0.0116 | 0.6340 | 0.0124 | 0.5043 | 0.0129 | 0.4970 | 0.0129 | 0.5586 | 0.0128 | 0.4937 | 0.0129 | 0.5923 | 0.0126 | 0.5003 | 0.0129 | 0.5374 | 0.0128 | 0.5917 | 0.0126 |
| LLaMAX Llama 2 7B | 0.6036 | 0.5890 | 0.0127 | 0.7551 | 0.0111 | 0.6525 | 0.0123 | 0.5447 | 0.0128 | 0.5817 | 0.0127 | 0.6062 | 0.0126 | 0.5248 | 0.0129 | 0.6122 | 0.0125 | 0.5718 | 0.0127 | 0.5930 | 0.0126 | 0.6082 | 0.0126 |
| LLaMAX Llama 2 7B Alpaca | 0.6383 | 0.6036 | 0.0126 | 0.8147 | 0.0100 | 0.7068 | 0.0117 | 0.5486 | 0.0128 | 0.6214 | 0.0125 | 0.6645 | 0.0122 | 0.5381 | 0.0128 | 0.6744 | 0.0121 | 0.6016 | 0.0126 | 0.5930 | 0.0126 | 0.6545 | 0.0122 |
| MaLA-500 Llama 2 10B v1 | 0.5307 | 0.4818 | 0.0129 | 0.7353 | 0.0114 | 0.6241 | 0.0125 | 0.4990 | 0.0129 | 0.4765 | 0.0129 | 0.4792 | 0.0129 | 0.4626 | 0.0128 | 0.5493 | 0.0128 | 0.4871 | 0.0129 | 0.5261 | 0.0128 | 0.5169 | 0.0129 |
| MaLA-500 Llama 2 10B v2 | 0.5307 | 0.4818 | 0.0129 | 0.7353 | 0.0114 | 0.6241 | 0.0125 | 0.4990 | 0.0129 | 0.4765 | 0.0129 | 0.4792 | 0.0129 | 0.4626 | 0.0128 | 0.5493 | 0.0128 | 0.4871 | 0.0129 | 0.5261 | 0.0128 | 0.5169 | 0.0129 |
| YaYi Llama 2 7B | 0.5842 | 0.4997 | 0.0129 | 0.7909 | 0.0105 | 0.6870 | 0.0119 | 0.5063 | 0.0129 | 0.5427 | 0.0128 | 0.6142 | 0.0125 | 0.4745 | 0.0129 | 0.6479 | 0.0123 | 0.5003 | 0.0129 | 0.5394 | 0.0128 | 0.6234 | 0.0125 |
| TowerBase Llama 2 7B | 0.5778 | 0.4917 | 0.0129 | 0.7723 | 0.0108 | 0.6982 | 0.0118 | 0.5076 | 0.0129 | 0.5288 | 0.0128 | 0.5831 | 0.0127 | 0.4838 | 0.0129 | 0.6704 | 0.0121 | 0.5036 | 0.0129 | 0.5314 | 0.0128 | 0.5850 | 0.0127 |
| TowerInstruct Llama 2 7B | 0.5924 | 0.4931 | 0.0129 | 0.8087 | 0.0101 | 0.7161 | 0.0116 | 0.5069 | 0.0129 | 0.5275 | 0.0128 | 0.5956 | 0.0126 | 0.4871 | 0.0129 | 0.6936 | 0.0119 | 0.5149 | 0.0129 | 0.5414 | 0.0128 | 0.6300 | 0.0124 |
| Occiglot Mistral 7B v0.1 | 0.5810 | 0.5129 | 0.0129 | 0.7737 | 0.0108 | 0.7340 | 0.0114 | 0.5208 | 0.0129 | 0.5149 | 0.0129 | 0.5864 | 0.0127 | 0.4798 | 0.0129 | 0.6294 | 0.0124 | 0.4983 | 0.0129 | 0.5314 | 0.0128 | 0.6089 | 0.0126 |
| Occiglot Mistral 7B v0.1 Instruct | 0.5939 | 0.5268 | 0.0128 | 0.7942 | 0.0104 | 0.7419 | 0.0113 | 0.5301 | 0.0128 | 0.5275 | 0.0128 | 0.6036 | 0.0126 | 0.4825 | 0.0129 | 0.6506 | 0.0123 | 0.5043 | 0.0129 | 0.5381 | 0.0128 | 0.6334 | 0.0124 |
| BLOOM 7B | 0.5930 | 0.5857 | 0.0127 | 0.7055 | 0.0117 | 0.6618 | 0.0122 | 0.5725 | 0.0127 | 0.6042 | 0.0126 | 0.6453 | 0.0123 | 0.4891 | 0.0129 | 0.5275 | 0.0128 | 0.5394 | 0.0128 | 0.5731 | 0.0127 | 0.6188 | 0.0125 |
| BLOOMZ 7B | 0.5712 | 0.5652 | 0.0128 | 0.7300 | 0.0114 | 0.6459 | 0.0123 | 0.5764 | 0.0127 | 0.5533 | 0.0128 | 0.6285 | 0.0124 | 0.4825 | 0.0129 | 0.5275 | 0.0129 | 0.5215 | 0.0129 | 0.5817 | 0.0127 | 0.5943 | 0.0126 |
| mGPT | 0.5443 | 0.4931 | 0.0129 | 0.5996 | 0.0126 | 0.5546 | 0.0128 | 0.5460 | 0.0128 | 0.5275 | 0.0128 | 0.5314 | 0.0128 | 0.5122 | 0.0129 | 0.5665 | 0.0128 | 0.5500 | 0.0128 | 0.5725 | 0.0127 | 0.5341 | 0.0128 |
| mGPT 13B | 0.5644 | 0.5162 | 0.0129 | 0.6420 | 0.0123 | 0.5864 | 0.0127 | 0.5539 | 0.0128 | 0.5506 | 0.0128 | 0.5811 | 0.0127 | 0.5152 | 0.0129 | 0.5943 | 0.0126 | 0.5460 | 0.0128 | 0.5764 | 0.0127 | 0.5493 | 0.0128 |
| XGLM 7.5B | 0.6075 | 0.5612 | 0.0128 | 0.6982 | 0.0118 | 0.6386 | 0.0124 | 0.5771 | 0.0127 | 0.5884 | 0.0127 | 0.6300 | 0.0124 | 0.5705 | 0.0127 | 0.6340 | 0.0124 | 0.5936 | 0.0126 | 0.6023 | 0.0126 | 0.5890 | 0.0127 |
| YaYi 7B | 0.6067 | 0.6181 | 0.0125 | 0.7432 | 0.0112 | 0.6942 | 0.0119 | 0.5606 | 0.0128 | 0.6367 | 0.0124 | 0.6241 | 0.0125 | 0.4924 | 0.0129 | 0.5215 | 0.0129 | 0.5361 | 0.0128 | 0.5791 | 0.0127 | 0.6678 | 0.0121 |
| Llama 3 8B | 0.6341 | 0.5864 | 0.0127 | 0.7869 | 0.0105 | 0.7062 | 0.0117 | 0.5579 | 0.0128 | 0.6281 | 0.0124 | 0.6592 | 0.0122 | 0.5109 | 0.0129 | 0.6863 | 0.0119 | 0.5639 | 0.0128 | 0.6300 | 0.0124 | 0.6578 | 0.0122 |
| Llama 3.1 8B | 0.6358 | 0.5910 | 0.0127 | 0.7816 | 0.0106 | 0.7081 | 0.0117 | 0.5533 | 0.0128 | 0.6327 | 0.0124 | 0.6803 | 0.0120 | 0.5242 | 0.0129 | 0.6863 | 0.0119 | 0.5592 | 0.0128 | 0.6115 | 0.0125 | 0.6658 | 0.0121 |
| Gemma 7B | 0.6501 | 0.6042 | 0.0126 | 0.8015 | 0.0103 | 0.7062 | 0.0117 | 0.5758 | 0.0127 | 0.6492 | 0.0123 | 0.6764 | 0.0120 | 0.5228 | 0.0129 | 0.7055 | 0.0117 | 0.6214 | 0.0125 | 0.6327 | 0.0124 | 0.6559 | 0.0122 |
| Gemma 2 9B | 0.6767 | 0.6525 | 0.0123 | 0.8015 | 0.0103 | 0.7426 | 0.0113 | 0.6016 | 0.0126 | 0.6691 | 0.0121 | 0.7134 | 0.0116 | 0.5513 | 0.0128 | 0.7373 | 0.0113 | 0.6373 | 0.0124 | 0.6479 | 0.0123 | 0.6896 | 0.0119 |
| Qwen 2 7B | 0.6146 | 0.5996 | 0.0126 | 0.7895 | 0.0105 | 0.6976 | 0.0118 | 0.5202 | 0.0129 | 0.5797 | 0.0127 | 0.6439 | 0.0123 | 0.4897 | 0.0129 | 0.6956 | 0.0118 | 0.5142 | 0.0129 | 0.5407 | 0.0128 | 0.6903 | 0.0119 |
| Qwen 1.5 7B | 0.5985 | 0.5539 | 0.0128 | 0.7816 | 0.0106 | 0.6830 | 0.0120 | 0.5189 | 0.0129 | 0.5566 | 0.0128 | 0.6287 | 0.0124 | 0.4924 | 0.0129 | 0.6327 | 0.0124 | 0.5169 | 0.0129 | 0.5394 | 0.0128 | 0.6797 | 0.0120 |
| EMMA-500 Llama 2 7B | 0.6638 | 0.6625 | 0.0122 | 0.7644 | 0.0109 | 0.7002 | 0.0118 | 0.6473 | 0.0123 | 0.6492 | 0.0123 | 0.6863 | 0.0119 | 0.5791 | 0.0127 | 0.6850 | 0.0120 | 0.6473 | 0.0123 | 0.6466 | 0.0123 | 0.6340 | 0.0124 |

Table 15: 0-shot results (ACC) on XWinograd in all languages

| Model | Avg | en-acc | stderr | fr-acc | stderr | jp-acc | stderr | pt-acc | stderr | ru-acc | stderr | zh-acc | stderr |
|---|---|---|---|---|---|---|---|---|---|---|---|---|---|
| Llama 2 7B | 0.7247 | 0.8791 | 0.0068 | 0.6627 | 0.0522 | 0.7028 | 0.0148 | 0.7224 | 0.0277 | 0.6825 | 0.0263 | 0.6984 | 0.0205 |
| Llama 2 7B Chat | 0.6945 | 0.8555 | 0.0073 | 0.7108 | 0.0501 | 0.6840 | 0.0150 | 0.6464 | 0.0295 | 0.6571 | 0.0268 | 0.6131 | 0.0217 |
| CodeLlama 2 7B | 0.7092 | 0.8452 | 0.0075 | 0.6747 | 0.0517 | 0.6361 | 0.0155 | 0.7224 | 0.0277 | 0.6667 | 0.0266 | 0.7103 | 0.0202 |
| LLaMAX Llama 2 7B | 0.6749 | 0.7789 | 0.0086 | 0.6145 | 0.0537 | 0.7101 | 0.0147 | 0.6540 | 0.0294 | 0.5556 | 0.0280 | 0.7361 | 0.0197 |
| LLaMAX Llama 2 7B Alpaca | 0.6986 | 0.8275 | 0.0078 | 0.6627 | 0.0522 | 0.7237 | 0.0144 | 0.6616 | 0.0292 | 0.5937 | 0.0277 | 0.7222 | 0.0200 |
| MaLA-500 Llama 2 10B v1 | 0.6589 | 0.8366 | 0.0077 | 0.6386 | 0.0531 | 0.6017 | 0.0158 | 0.6920 | 0.0285 | 0.6190 | 0.0274 | 0.5655 | 0.0221 |
| MaLA-500 Llama 2 10B v2 | 0.6589 | 0.8366 | 0.0077 | 0.6386 | 0.0531 | 0.6017 | 0.0158 | 0.6920 | 0.0285 | 0.6190 | 0.0274 | 0.5655 | 0.0221 |
| YaYi Llama 2 7B | 0.7450 | 0.8852 | 0.0066 | 0.7108 | 0.0501 | 0.7226 | 0.0145 | 0.7414 | 0.0270 | 0.7175 | 0.0254 | 0.6925 | 0.0206 |
| TowerBase Llama 2 7B | 0.7429 | 0.8714 | 0.0069 | 0.7470 | 0.0480 | 0.6945 | 0.0149 | 0.7567 | 0.0259 | 0.6476 | 0.0270 | 0.7401 | 0.0196 |
| TowerInstruct Llama 2 7B | 0.7400 | 0.8628 | 0.0071 | 0.7711 | 0.0464 | 0.6840 | 0.0150 | 0.7719 | 0.0259 | 0.6635 | 0.0267 | 0.6865 | 0.0207 |
| Occiglot Mistral 7B v0.1 | 0.7461 | 0.8654 | 0.0071 | 0.7952 | 0.0446 | 0.6601 | 0.0153 | 0.7186 | 0.0278 | 0.6635 | 0.0267 | 0.7738 | 0.0187 |
| Occiglot Mistral 7B v0.1 Instruct | 0.7293 | 0.8589 | 0.0072 | 0.7470 | 0.0480 | 0.6455 | 0.0155 | 0.7072 | 0.0281 | 0.6476 | 0.0270 | 0.7698 | 0.0188 |
| BLOOM 7B | 0.7013 | 0.8224 | 0.0079 | 0.7108 | 0.0501 | 0.5881 | 0.0159 | 0.7681 | 0.0267 | 0.5746 | 0.0279 | 0.7440 | 0.0195 |
| BLOOMZ 7B | 0.6795 | 0.8340 | 0.0077 | 0.7349 | 0.0487 | 0.5756 | 0.0160 | 0.6730 | 0.0290 | 0.5492 | 0.0281 | 0.7103 | 0.0202 |
| mGPT | 0.5969 | 0.6267 | 0.0100 | 0.5904 | 0.0543 | 0.5349 | 0.0161 | 0.5741 | 0.0305 | 0.5810 | 0.0278 | 0.6746 | 0.0209 |
| mGPT 13B | 0.6359 | 0.7062 | 0.0094 | 0.6386 | 0.0531 | 0.5777 | 0.0160 | 0.6122 | 0.0301 | 0.6000 | 0.0276 | 0.6806 | 0.0208 |
| XGLM 7.5B | 0.6884 | 0.7940 | 0.0084 | 0.6506 | 0.0527 | 0.6496 | 0.0154 | 0.6768 | 0.0289 | 0.6349 | 0.0272 | 0.7242 | 0.0199 |
| YaYi 7B | 0.6979 | 0.8404 | 0.0076 | 0.7590 | 0.0472 | 0.5808 | 0.0159 | 0.7224 | 0.0277 | 0.5587 | 0.0280 | 0.7262 | 0.0199 |
| Llama 3 8B | 0.7684 | 0.8680 | 0.0070 | 0.7108 | 0.0501 | 0.7529 | 0.0139 | 0.7985 | 0.0248 | 0.7143 | 0.0255 | 0.7659 | 0.0189 |
| Llama 3.1 8B | 0.7552 | 0.8757 | 0.0068 | 0.6627 | 0.0522 | 0.7529 | 0.0139 | 0.8061 | 0.0244 | 0.6762 | 0.0264 | 0.7579 | 0.0191 |
| Gemma 7B | 0.7741 | 0.8800 | 0.0067 | 0.7349 | 0.0487 | 0.7539 | 0.0139 | 0.7833 | 0.0255 | 0.7206 | 0.0253 | 0.7718 | 0.0187 |
| Gemma 2 9B | 0.8007 | 0.8942 | 0.0064 | 0.7590 | 0.0472 | 0.7977 | 0.0130 | 0.8289 | 0.0233 | 0.7429 | 0.0247 | 0.7817 | 0.0184 |
| Qwen 2 7B | 0.7644 | 0.8632 | 0.0071 | 0.6988 | 0.0507 | 0.6882 | 0.0150 | 0.7871 | 0.0253 | 0.7175 | 0.0254 | 0.8313 | 0.0167 |
| Qwen 1.5 7B | 0.7259 | 0.8331 | 0.0077 | 0.6988 | 0.0507 | 0.6590 | 0.0153 | 0.7034 | 0.0282 | 0.6635 | 0.0267 | 0.7976 | 0.0179 |
| EMMA-500 Llama 2 7B | 0.7280 | 0.8245 | 0.0079 | 0.7229 | 0.0494 | 0.7164 | 0.0146 | 0.6920 | 0.0285 | 0.6921 | 0.0261 | 0.7202 | 0.0200 |

We perform test-case-based execution-tested evaluations using the `pass@k` metric (Chen et al., 2021). We benchmark for k values of 1,10, and 25 using a generation pool of 50 samples per problem. Table 24 outlines comparisons against strong, openly available multilingual models such as Qwen (Yang et al., 2024a) and Aya (Üstün et al., 2024) along with other continually pre-trained models. Our main takeaway is that by carefully curating high-quality code data as part of the data mix, it is possible to not only avoid the catastrophic forgetting that has plagued existing continually pre-trained models (MaLA-500, LLaMAX and TowerBase) but also surpass the base model itself. However, the pre-training phase is still the most important, and closing the gap with stronger base models like Qwen and Gemma remains elusive.

### E.4 MACHINE TRANSLATION

Table 16: 3-shot results on FLORES-200 (Eng-X, BLEU/chrF++). EMMA-500 Llama 2 7B has better average performance than all baselines.

| Model | Avg | High | Medium-High | Medium | Medium-Low | Low |
|---|---|---|---|---|---|---|
| Llama 2 7B | 4.62/ 15.13 | 10.77/ 24.38 | 8.56/ 21.4 | 2.55/ 13.72 | 0.74/ 8.72 | 0.7/ 7.92 |
| Llama 2 7B Chat | 4.95/ 16.95 | 10.87/ 24.51 | 8.54/ 22.69 | 3.25/ 15.5 | 1.52/ 12.08 | 0.94/ 10.03 |
| CodeLlama 2 7B | 4.27/ 14.94 | 10.04/ 23.48 | 7.79/ 20.79 | 2.57/ 14.2 | 0.71/ 9.27 | 0.58/ 7.49 |
| LLaMAX Llama 2 7B | 0.8/ 7.42 | 1.85/ 12.06 | 1.2/ 9.74 | 0.54/ 6.55 | 0.22/ 4.52 | 0.38/ 4.81 |
| LLaMAX Llama 2 7B Alpaca | 12.51/ 28.35 | 24.8/ 41.76 | 18.69/ 38.42 | 10.1/ 27.27 | 3.79/ 16.53 | 6.68/ 18.15 |
| MaLA-500 Llama 2 10B v1[‡] | 0.6/ 6.08 | 1.51/ 9.0 | 1.13/ 8.19 | 0.35/ 5.99 | 0.07/ 4.5 | 0.02/ 2.9 |
| MaLA-500 Llama 2 10B v2[‡] | 0.54/ 6.38 | 1.4/ 9.19 | 1.02/ 8.42 | 0.24/ 5.99 | 0.07/ 5.14 | 0.02/ 3.27 |
| Yayi Llama 2 7B | 4.41/ 14.87 | 10.49/ 24.0 | 8.21/ 21.27 | 2.52/ 13.57 | 0.6/ 8.49 | 0.53/ 7.42 |
| TowerBase Llama 2 7B | 4.83/ 16.03 | 11.89/ 24.15 | 8.33/ 21.46 | 2.57/ 14.49 | 1.38/ 11.6 | 0.74/ 8.9 |
| TowerInstruct Llama 2 7B | 3.23/ 15.64 | 7.22/ 22.65 | 4.99/ 20.0 | 2.2/ 14.9 | 1.62/ 12.31 | 0.73/ 8.97 |
| Occiglot Mistral 7B v0.1 | 4.32/ 16.1 | 10.5/ 23.74 | 6.95/ 20.91 | 2.87/ 15.44 | 1.47/ 12.0 | 0.79/ 9.09 |
| Occiglot Mistral 7B v0.1 Instruct | 3.99/ 15.8 | 9.46/ 23.17 | 6.46/ 20.73 | 2.68/ 15.29 | 1.31/ 11.31 | 0.84/ 9.04 |
| BLOOM 7B | 2.81/ 11.8 | 7.53/ 19.0 | 3.12/ 13.36 | 2.05/ 11.48 | 0.85/ 8.0 | 2.09/ 9.22 |
| BLOOMZ 7B[†] | 7.44/ 16.1 | 23.64/ 32.22 | 7.46/ 16.62 | 6.98/ 16.05 | 1.28/ 9.99 | 4.17/ 11.77 |
| mGPT | 2.59/ 12.56 | 5.24/ 17.04 | 4.75/ 16.92 | 1.14/ 9.75 | 0.78/ 9.24 | 0.84/ 9.08 |
| mGPT-13B | 3.88/ 14.57 | 8.32/ 21.7 | 6.84/ 20.55 | 2.06/ 12.23 | 0.9/ 8.4 | 1.33/ 9.58 |
| Yayi 7B | 4.37/ 13.5 | 13.72/ 26.28 | 4.68/ 14.31 | 3.35/ 12.89 | 0.91/ 8.51 | 2.55/ 10.08 |
| Llama 3 8B | 9.93/ 24.08 | 20.38/ 36.87 | 14.95/ 32.05 | 8.89/ 24.28 | 2.83/ 14.26 | 4.2/ 14.29 |
| Llama 3.1 8B | 10.11/ 24.69 | 20.82/ 37.39 | 15.3/ 32.82 | 8.85/ 24.85 | 2.9/ 14.83 | 4.23/ 14.81 |
| Gemma 2 9B | 12.09/ 26.48 | 24.62/ 40.69 | 17.82/ 35.51 | 10.68/ 26.58 | 3.38/ 15.02 | 5.94/ 15.98 |
| Gemma 7B | 9.05/ 23.05 | 17.58/ 34.5 | 13.62/ 30.16 | 7.96/ 22.85 | 2.64/ 14.11 | 4.47/ 14.82 |
| Qwen 1.5 7B | 5.87/ 17.77 | 14.05/ 28.6 | 8.88/ 23.57 | 3.85/ 17.07 | 1.7/ 10.85 | 2.35/ 10.21 |
| Qwen 2 7B | 5.56/ 17.17 | 13.22/ 27.65 | 8.21/ 22.36 | 4.15/ 16.93 | 1.56/ 10.47 | 2.19/ 10.11 |
| EMMA-500 Llama 2 7B | 15.58/ 33.25 | 26.37/ 42.4 | 21.96/ 41.98 | 13.4/ 32.06 | 9.15/ 27.99 | 7.92/ 21.25 |

Table 17: 0-shot results on XNLI (ACC).

| Model | Avg | High | Medium | Low |
|---|---|---|---|---|
| Llama 2 7B | 0.4019 | 0.4526 | 0.3772 | 0.3497 |
| Llama 2 7B Chat | 0.3858 | 0.4277 | 0.3675 | 0.3387 |
| CodeLlama 2 7B | 0.4019 | 0.4627 | 0.3729 | 0.3386 |
| LLaMAX Llama 2 7B | 0.4427 | 0.4653 | 0.4264 | 0.4303 |
| LLaMAX Llama 2 7B Alpaca | 0.4509 | 0.4847 | 0.4280 | 0.4289 |
| MaLA-500 Llama 2 10B v1 | 0.3811 | 0.4210 | 0.3585 | 0.3465 |
| MaLA-500 Llama 2 10B v2 | 0.3811 | 0.4210 | 0.3585 | 0.3465 |
| YaYi Llama 2 7B | 0.4128 | 0.4732 | 0.3841 | 0.3494 |
| TowerBase Llama 2 7B | 0.3984 | 0.4608 | 0.3633 | 0.3439 |
| TowerInstruct Llama 2 7B | 0.4036 | 0.4707 | 0.3692 | 0.3379 |
| Occiglot Mistral 7B v0.1 | 0.4235 | 0.4990 | 0.3839 | 0.3519 |
| Occiglot Mistral 7B v0.1 Instruct | 0.4081 | 0.4758 | 0.3718 | 0.3452 |
| BLOOM 7B | 0.4160 | 0.4513 | 0.3969 | 0.3838 |
| BLOOMZ 7B | 0.3713 | 0.4002 | 0.3556 | 0.3451 |
| mGPT | 0.4051 | 0.4297 | 0.3965 | 0.3730 |
| XGLM 7.5B | 0.4375 | 0.4572 | 0.4216 | 0.4300 |
| YaYi 7B | 0.3987 | 0.4385 | 0.3824 | 0.3515 |
| Llama 3 8B | 0.4497 | 0.4882 | 0.4384 | 0.3956 |
| Llama 3.1 8B | 0.4562 | 0.4961 | 0.4404 | 0.4083 |
| Gemma 7B | 0.4258 | 0.4644 | 0.4100 | 0.3801 |
| Gemma 2 9B | 0.4674 | 0.4850 | 0.4511 | 0.4649 |
| Qwen 2 7B | 0.4277 | 0.4731 | 0.4135 | 0.3653 |
| Qwen 1.5 7B | 0.3947 | 0.4095 | 0.3880 | 0.3783 |
| EMMA-500 Llama 2 7B | 0.4514 | 0.4609 | 0.4440 | 0.4471 |

## F RELATED WORK

**Multilingual LLMs**    Multilingual large language models (LLMs) have made significant progress in processing and understanding multiple languages within a unified framework. Models like mT5 (Xue et al., 2021) and XGLM (Lin et al., 2022) leverage both monolingual and multilingual datasets to perform tasks such as translation and text summarization across a wide spectrum of languages. However, the predominant focus on English has led to disparities in performance, particularly for low-resource languages. Recent work on multilingual LLMs, such as BLOOM (Scao et al., 2022), has

Table 18: 0-shot results (ACC) on XNLI in all languages.

| Model | Avg | ar-acc | stderr | bg-acc | stderr | de-acc | stderr | el-acc | stderr | en-acc | stderr | es-acc | stderr | fr-acc | stderr | hi-acc | stderr | ru-acc | stderr | sw-acc | stderr | th-acc | stderr | tr-acc | stderr | ur-acc | stderr | vi-acc | stderr | zh-acc | stderr |
|---|---|---|---|---|---|---|---|---|---|---|---|---|---|---|---|---|---|---|---|---|---|---|---|---|---|---|---|---|---|---|---|
| Llama 2 7B | 0.4019 | 0.3542 | 0.0096 | 0.4265 | 0.0099 | 0.4711 | 0.0100 | 0.3667 | 0.0097 | 0.5530 | 0.0100 | 0.4052 | 0.0098 | 0.5008 | 0.0100 | 0.3771 | 0.0097 | 0.4237 | 0.0099 | 0.3494 | 0.0096 | 0.3635 | 0.0096 | 0.3727 | 0.0097 | 0.3361 | 0.0095 | 0.3663 | 0.0097 | 0.3618 | 0.0097 |
| Llama 2 7B Chat | 0.3858 | 0.3442 | 0.0095 | 0.3707 | 0.0097 | 0.4309 | 0.0099 | 0.3815 | 0.0097 | 0.5024 | 0.0100 | 0.3944 | 0.0098 | 0.4482 | 0.0100 | 0.3578 | 0.0096 | 0.4209 | 0.0099 | 0.3422 | 0.0095 | 0.3349 | 0.0095 | 0.3695 | 0.0098 | 0.3390 | 0.0095 | 0.3811 | 0.0097 | 0.3695 | 0.0097 |
| CodeLlama 2 7B | 0.4019 | 0.3341 | 0.0095 | 0.3775 | 0.0097 | 0.4723 | 0.0100 | 0.3763 | 0.0097 | 0.5024 | 0.0100 | 0.3944 | 0.0098 | 0.4438 | 0.0100 | 0.3594 | 0.0096 | 0.4920 | 0.0100 | 0.4606 | 0.0100 | 0.3329 | 0.0095 | 0.3502 | 0.0096 | 0.3859 | 0.0098 | 0.3325 | 0.0094 | 0.3594 | 0.0096 |
| LLaMAX Llama 2 7B | 0.4427 | 0.3378 | 0.0095 | 0.4683 | 0.0100 | 0.4896 | 0.0100 | 0.4257 | 0.0099 | 0.5490 | 0.0100 | 0.4759 | 0.0100 | 0.5197 | 0.0100 | 0.4562 | 0.0100 | 0.4627 | 0.0100 | 0.4305 | 0.0099 | 0.4185 | 0.0099 | 0.4329 | 0.0099 | 0.4418 | 0.0100 | 0.4309 | 0.0099 | 0.3438 | 0.0095 |
| LLaMAX Llama 2 7B Alpaca | 0.4509 | 0.3442 | 0.0095 | 0.4639 | 0.0100 | 0.4976 | 0.0100 | 0.4341 | 0.0099 | 0.5811 | 0.0099 | 0.4896 | 0.0100 | 0.5197 | 0.0100 | 0.4562 | 0.0100 | 0.4627 | 0.0100 | 0.4357 | 0.0099 | 0.4080 | 0.0099 | 0.4386 | 0.0099 | 0.4430 | 0.0100 | 0.4309 | 0.0099 | 0.3578 | 0.0096 |
| MaLA-500 Llama 2 10B v1 | 0.3811 | 0.3594 | 0.0096 | 0.4120 | 0.0099 | 0.4751 | 0.0100 | 0.3446 | 0.0095 | 0.5618 | 0.0099 | 0.3410 | 0.0095 | 0.4759 | 0.0100 | 0.3365 | 0.0095 | 0.3394 | 0.0095 | 0.3522 | 0.0096 | 0.3369 | 0.0095 | 0.3382 | 0.0095 | 0.3502 | 0.0096 | 0.3602 | 0.0096 | 0.3325 | 0.0094 |
| MaLA-500 Llama 2 10B v2 | 0.3811 | 0.3594 | 0.0096 | 0.4120 | 0.0099 | 0.4751 | 0.0100 | 0.3446 | 0.0095 | 0.5618 | 0.0099 | 0.3410 | 0.0095 | 0.4759 | 0.0100 | 0.3365 | 0.0095 | 0.3394 | 0.0095 | 0.3522 | 0.0096 | 0.3369 | 0.0095 | 0.3382 | 0.0095 | 0.3502 | 0.0096 | 0.3602 | 0.0096 | 0.3325 | 0.0094 |
| YaYi Llama 2 7B | 0.4128 | 0.3414 | 0.0095 | 0.4261 | 0.0099 | 0.4884 | 0.0100 | 0.3735 | 0.0097 | 0.5647 | 0.0099 | 0.4578 | 0.0100 | 0.5104 | 0.0100 | 0.3948 | 0.0098 | 0.4627 | 0.0100 | 0.3570 | 0.0096 | 0.3562 | 0.0096 | 0.3936 | 0.0098 | 0.3349 | 0.0095 | 0.3751 | 0.0097 | 0.3554 | 0.0096 |
| TowerBase Llama 2 7B | 0.3984 | 0.3390 | 0.0095 | 0.4137 | 0.0099 | 0.4787 | 0.0100 | 0.3526 | 0.0096 | 0.5635 | 0.0099 | 0.4169 | 0.0099 | 0.4944 | 0.0100 | 0.3434 | 0.0095 | 0.4594 | 0.0100 | 0.3502 | 0.0096 | 0.3574 | 0.0096 | 0.3337 | 0.0095 | 0.3719 | 0.0097 | 0.3522 | 0.0096 |
| TowerInstruct Llama 2 7B | 0.4036 | 0.3365 | 0.0095 | 0.4293 | 0.0099 | 0.4884 | 0.0100 | 0.3498 | 0.0096 | 0.5695 | 0.0099 | 0.4651 | 0.0100 | 0.4643 | 0.0100 | 0.3474 | 0.0095 | 0.4627 | 0.0100 | 0.3394 | 0.0095 | 0.3390 | 0.0095 | 0.3787 | 0.0097 | 0.3353 | 0.0095 | 0.3735 | 0.0097 | 0.3747 | 0.0097 |
| Occiglot Mistral 7B v0.1 | 0.4235 | 0.3386 | 0.0095 | 0.4137 | 0.0099 | 0.5177 | 0.0100 | 0.3771 | 0.0097 | 0.5586 | 0.0096 | 0.5165 | 0.0100 | 0.5193 | 0.0100 | 0.3574 | 0.0096 | 0.4763 | 0.0100 | 0.3574 | 0.0096 | 0.3739 | 0.0097 | 0.4301 | 0.0099 | 0.3349 | 0.0095 | 0.3863 | 0.0098 | 0.4056 | 0.0098 |
| Occiglot Mistral 7B v0.1 Instruct | 0.4081 | 0.3438 | 0.0095 | 0.3884 | 0.0098 | 0.5084 | 0.0100 | 0.4000 | 0.0098 | 0.5566 | 0.0100 | 0.4863 | 0.0100 | 0.5169 | 0.0100 | 0.3430 | 0.0095 | 0.4004 | 0.0098 | 0.3317 | 0.0094 | 0.3635 | 0.0096 | 0.3799 | 0.0097 | 0.3406 | 0.0095 | 0.3753 | 0.0097 | 0.4739 | 0.0071 |
| BLOOM 7B | 0.4160 | 0.3385 | 0.0067 | 0.3992 | 0.0069 | 0.3978 | 0.0069 | 0.3537 | 0.0068 | 0.5397 | 0.0070 | 0.4882 | 0.0071 | 0.4980 | 0.0071 | 0.4651 | 0.0070 | 0.4303 | 0.0070 | 0.3788 | 0.0069 | 0.3505 | 0.0067 | 0.3509 | 0.0067 | 0.3683 | 0.0097 | 0.3667 | 0.0097 | 0.3535 | 0.0068 |
| BLOOMZ 7B | 0.3713 | 0.3269 | 0.0094 | 0.3402 | 0.0095 | 0.4169 | 0.0099 | 0.3582 | 0.0096 | 0.4687 | 0.0100 | 0.3602 | 0.0096 | 0.4305 | 0.0099 | 0.4040 | 0.0098 | 0.3747 | 0.0097 | 0.3361 | 0.0095 | 0.3309 | 0.0094 | 0.3373 | 0.0095 | 0.3683 | 0.0097 | 0.3667 | 0.0097 | 0.3502 | 0.0096 |
| mGPT | 0.4051 | 0.3382 | 0.0095 | 0.4173 | 0.0099 | 0.4478 | 0.0100 | 0.3542 | 0.0096 | 0.4936 | 0.0100 | 0.4281 | 0.0099 | 0.4406 | 0.0100 | 0.4133 | 0.0099 | 0.4321 | 0.0099 | 0.4189 | 0.0099 | 0.3578 | 0.0096 | 0.4012 | 0.0098 | 0.3422 | 0.0095 | 0.4550 | 0.0100 | 0.3361 | 0.0095 |
| XGLM 7.5B | 0.4375 | 0.3349 | 0.0095 | 0.4365 | 0.0099 | 0.4755 | 0.0100 | 0.4040 | 0.0098 | 0.5309 | 0.0100 | 0.4707 | 0.0100 | 0.4542 | 0.0100 | 0.4574 | 0.0100 | 0.4598 | 0.0100 | 0.4570 | 0.0099 | 0.4137 | 0.0099 | 0.4679 | 0.0100 | 0.4193 | 0.0099 | 0.4289 | 0.0099 | 0.3522 | 0.0096 |
| YaYi 7B | 0.3987 | 0.3980 | 0.0098 | 0.3578 | 0.0096 | 0.4261 | 0.0099 | 0.3647 | 0.0096 | 0.5052 | 0.0100 | 0.4771 | 0.0100 | 0.4819 | 0.0100 | 0.4004 | 0.0098 | 0.3912 | 0.0098 | 0.3406 | 0.0095 | 0.3434 | 0.0095 | 0.3317 | 0.0095 | 0.3707 | 0.0097 | 0.4418 | 0.0100 | 0.3494 | 0.0096 |
| Llama 3 8B | 0.4497 | 0.3365 | 0.0095 | 0.4534 | 0.0100 | 0.5048 | 0.0100 | 0.3928 | 0.0098 | 0.5502 | 0.0100 | 0.4952 | 0.0100 | 0.5056 | 0.0100 | 0.4755 | 0.0100 | 0.4916 | 0.0100 | 0.3892 | 0.0098 | 0.4627 | 0.0100 | 0.4823 | 0.0100 | 0.3349 | 0.0095 | 0.4900 | 0.0100 | 0.3815 | 0.0097 |
| Llama 3.1 8B | 0.4562 | 0.3386 | 0.0095 | 0.4558 | 0.0100 | 0.5145 | 0.0100 | 0.3896 | 0.0098 | 0.5522 | 0.0100 | 0.5028 | 0.0100 | 0.5177 | 0.0100 | 0.4940 | 0.0100 | 0.4916 | 0.0100 | 0.3936 | 0.0098 | 0.4807 | 0.0100 | 0.4932 | 0.0100 | 0.3506 | 0.0096 | 0.4711 | 0.0100 | 0.3976 | 0.0098 |
| Gemma 7B | 0.4258 | 0.3349 | 0.0095 | 0.4549 | 0.0099 | 0.4863 | 0.0100 | 0.3811 | 0.0097 | 0.5205 | 0.0100 | 0.4414 | 0.0100 | 0.4976 | 0.0100 | 0.4434 | 0.0100 | 0.4739 | 0.0100 | 0.4064 | 0.0098 | 0.4558 | 0.0100 | 0.4988 | 0.0100 | 0.3795 | 0.0097 | 0.4325 | 0.0099 | 0.3542 | 0.0096 |
| Gemma 2 9B | 0.4674 | 0.3418 | 0.0095 | 0.4952 | 0.0100 | 0.5137 | 0.0100 | 0.4325 | 0.0099 | 0.5545 | 0.0100 | 0.5345 | 0.0100 | 0.5229 | 0.0100 | 0.4731 | 0.0100 | 0.4956 | 0.0100 | 0.4558 | 0.0100 | 0.4988 | 0.0100 | 0.5072 | 0.0100 | 0.4402 | 0.0100 | 0.4570 | 0.0100 | 0.3293 | 0.0094 |
| Qwen 2 7B | 0.4277 | 0.3369 | 0.0095 | 0.4546 | 0.0100 | 0.4819 | 0.0100 | 0.3683 | 0.0097 | 0.5426 | 0.0100 | 0.4723 | 0.0100 | 0.5141 | 0.0100 | 0.4498 | 0.0100 | 0.4739 | 0.0100 | 0.3731 | 0.0097 | 0.3827 | 0.0097 | 0.4353 | 0.0099 | 0.3402 | 0.0095 | 0.3538 | 0.0096 | - |
| Qwen 1.5 7B | 0.3947 | 0.3434 | 0.0095 | 0.4092 | 0.0099 | 0.4277 | 0.0099 | 0.3618 | 0.0096 | 0.4908 | 0.0100 | 0.3787 | 0.0097 | 0.4313 | 0.0099 | 0.3815 | 0.0097 | 0.3888 | 0.0098 | 0.3542 | 0.0096 | 0.4422 | 0.0100 | 0.3884 | 0.0098 | 0.3386 | 0.0095 | 0.4438 | 0.0100 | 0.3398 | 0.0095 |
| EMMA-500 Llama 2 7B | 0.4514 | 0.3478 | 0.0095 | 0.4627 | 0.0100 | 0.4707 | 0.0100 | 0.4586 | 0.0100 | 0.5378 | 0.0100 | 0.4707 | 0.0100 | 0.4687 | 0.0100 | 0.4759 | 0.0100 | 0.4655 | 0.0100 | 0.4618 | 0.0100 | 0.4197 | 0.0099 | 0.4486 | 0.0100 | 0.4598 | 0.0100 | 0.4703 | 0.0100 | 0.3522 | 0.0096 |

Table 19: 3-shot results (ACC) on MGSM obtained by direct and CoT prompting.

| Model | Direct Prompting | | | | CoT Prompting | | | |
|---|---|---|---|---|---|---|---|---|
| | Avg | High | Medium | Low | Avg | High | Medium | Low |
| Llama 2 7B | 0.0669 | 0.0807 | 0.0213 | 0.0120 | 0.0636 | 0.0760 | 0.0213 | 0.0080 |
| Llama 2 7B Chat | 0.1022 | 0.1373 | 0.0213 | 0.0080 | 0.1091 | 0.1353 | 0.0280 | 0.0160 |
| CodeLlama 2 7B | 0.0593 | 0.0707 | 0.0293 | 0.0120 | 0.0664 | 0.0873 | 0.0267 | 0.0200 |
| LLaMAX Llama 2 7B | 0.0335 | 0.0400 | 0.0200 | 0.0080 | 0.0362 | 0.0433 | 0.0227 | 0.0240 |
| LLaMAX Llama 2 7B Alpaca | 0.0505 | 0.0520 | 0.0400 | 0.0160 | 0.0635 | 0.0807 | 0.0413 | 0.0080 |
| MaLA-500 Llama 2 10B v1 | 0.0091 | 0.0133 | 0.0027 | 0.0000 | 0.0073 | 0.0127 | 0.0000 | 0.0000 |
| MaLA-500 Llama 2 10B v2 | 0.0091 | 0.0133 | 0.0027 | 0.0000 | 0.0073 | 0.0127 | 0.0000 | 0.0000 |
| TowerBase Llama 2 7B | 0.0615 | 0.0833 | 0.0173 | 0.0080 | 0.0616 | 0.0860 | 0.0240 | 0.0080 |
| TowerInstruct Llama 2 7B | 0.0724 | 0.0953 | 0.0173 | 0.0200 | 0.0824 | 0.1047 | 0.0187 | 0.0120 |
| Occiglot Mistral 7B v0.1 | 0.1331 | 0.1687 | 0.0453 | 0.0160 | 0.1407 | 0.1880 | 0.0360 | 0.0120 |
| Occiglot Mistral 7B v0.1 Instruct | 0.2276 | 0.2980 | 0.0747 | 0.0280 | 0.2216 | 0.3040 | 0.0787 | 0.0280 |
| BLOOM 7B | 0.0287 | 0.0260 | 0.0280 | 0.0360 | 0.0229 | 0.0220 | 0.0147 | 0.0200 |
| BLOOMZ 7B | 0.0255 | 0.0267 | 0.0240 | 0.0120 | 0.0215 | 0.0167 | 0.0307 | 0.0200 |
| mGPT | 0.0135 | 0.0167 | 0.0053 | 0.0000 | 0.0142 | 0.0193 | 0.0067 | 0.0000 |
| mGPT 13B | 0.0131 | 0.0180 | 0.0067 | 0.0000 | 0.0153 | 0.0167 | 0.0120 | 0.0000 |
| XGLM 7.5B | 0.0102 | 0.0120 | 0.0067 | 0.0200 | 0.0116 | 0.0107 | 0.0120 | 0.0280 |
| Yayi 7B | 0.0276 | 0.0293 | 0.0147 | 0.0240 | 0.0302 | 0.0293 | 0.0240 | 0.0200 |
| Llama 3 8B | 0.2745 | 0.2787 | 0.2613 | 0.0560 | 0.2813 | 0.2853 | 0.2667 | 0.0520 |
| Llama 3.1 8B | 0.2836 | 0.2900 | 0.2613 | 0.0440 | 0.2731 | 0.2727 | 0.2547 | 0.0840 |
| Gemma 7B | 0.3822 | 0.3660 | 0.3827 | 0.2720 | 0.3578 | 0.3467 | 0.3707 | 0.2680 |
| Gemma 2 9B | 0.3295 | 0.2800 | 0.3573 | 0.3080 | 0.4469 | 0.3607 | 0.5200 | 0.4720 |
| Qwen 2 7B | 0.4895 | 0.5440 | 0.3880 | 0.1480 | 0.4469 | 0.3607 | 0.5200 | 0.4720 |
| Qwen 1.5 7B | 0.3156 | 0.4000 | 0.1600 | 0.0400 | 0.5147 | 0.5893 | 0.3907 | 0.1440 |
| EMMA-500 Llama 2 7B | 0.1702 | 0.1920 | 0.1187 | 0.0240 | 0.3036 | 0.4060 | 0.1480 | 0.0240 |

shown that adapting these English-centric models through vocabulary extension based on multilingual corpora and continual pre-training (CPT) can improve performance across languages, especially low-resource ones. These models highlight the importance of efficient tokenization and adaptation, which can bridge the performance gap between high-resource and low-resource languages.

**Multilingual Corpora** The availability and use of multilingual corpora play a crucial role in training multilingual LLMs. CC100 Corpus (Conneau et al., 2020), launched in 2020, encompasses hundreds of billions of tokens and over 100 languages. Further, CC100-XL Corpus (Lin et al., 2022), created for the training of XGLM, extends across 68 Common Crawl Snapshots and 134 languages, aiming to balance language presentation and improve performance in few-shot and zero-shot tasks. The ROOTS Corpus (Laurençon et al., 2022), released in July 2022, supports BLOOM with approximately 341 billion tokens across 46 natural languages. It emphasizes underrepresented languages such as Swahili and Catalan, drawing from diverse sources including web crawls, books and academic publications. Besides, Occiglot Fineweb [47], which began to be released in early 2024, consists of around 230 million documents in 10 European languages. It combines curated and cleaned web data to support efficient training for both high- and low-resource European languages. Additionally, recent efforts parallel corpus construction from web crawls, such as ParaCrawl (Bañón et al., 2020) and CCMatrix (Schwenk et al., 2021b), have contributed to large-scale multilingual training too.

[47] https://occiglot.eu/posts/occiglot-fineweb/

Tables 20 and 21 show 3-shot results on MGSM by direct and Chain-of-Thought prompting respectively. All the scores are obtained by flexible matching.

Table 20: 3-shot results (ACC) on MGSM by direct prompting and flexible matching.

| Model | Avg | bn | bn-stderr | de | de-stderr | en | en-stderr | es | es-stderr | fr | fr-stderr | ja | ja-stderr | ru | ru-stderr | sw | sw-stderr | te | te-stderr | th | th-stderr | zh | zh-stderr |
|---|---|---|---|---|---|---|---|---|---|---|---|---|---|---|---|---|---|---|---|---|---|---|---|
| Llama 2 7B | 0.0669 | 0.0280 | 0.0105 | 0.0800 | 0.0172 | 0.1760 | 0.0241 | 0.1120 | 0.0200 | 0.1200 | 0.0206 | 0.0240 | 0.0097 | 0.0800 | 0.0172 | 0.0280 | 0.0105 | 0.0120 | 0.0069 | 0.0080 | 0.0056 | 0.0680 | 0.0160 |
| Llama 2 7B Chat | 0.1022 | 0.0280 | 0.0105 | 0.1680 | 0.0237 | 0.2780 | 0.0266 | 0.1960 | 0.0252 | 0.1920 | 0.0250 | 0.0240 | 0.0097 | 0.1440 | 0.0222 | 0.0080 | 0.0056 | 0.0280 | 0.0105 | 0.1000 | 0.0105 | 0.1000 | 0.0190 |
| CodeLlama 2 7B | 0.0593 | 0.0160 | 0.0080 | 0.0880 | 0.0180 | 0.1280 | 0.0212 | 0.0880 | 0.0193 | 0.1040 | 0.0193 | 0.0560 | 0.0146 | 0.0600 | 0.0151 | 0.0200 | 0.0089 | 0.0120 | 0.0069 | 0.0520 | 0.0141 | 0.0280 | 0.0105 |
| LLaMAX Llama 2 7B | 0.0335 | 0.0280 | 0.0105 | 0.0360 | 0.0118 | 0.0600 | 0.0151 | 0.0200 | 0.0089 | 0.0720 | 0.0164 | 0.0320 | 0.0112 | 0.0240 | 0.0097 | 0.0160 | 0.0080 | 0.0080 | 0.0056 | 0.0160 | 0.0080 | 0.0560 | 0.0146 |
| LLaMAX Llama 2 7B Alpaca | 0.0505 | 0.0400 | 0.0124 | 0.0360 | 0.0118 | 0.1080 | 0.0197 | 0.0600 | 0.0151 | 0.0640 | 0.0155 | 0.0320 | 0.0112 | 0.0440 | 0.0130 | 0.0480 | 0.0135 | 0.0160 | 0.0089 | 0.0320 | 0.0112 | 0.0760 | 0.0168 |
| MaLA-500 Llama 2 10B v1 | 0.0091 | 0.0000 | 0.0000 | 0.0000 | 0.0000 | 0.0120 | 0.0069 | 0.0200 | 0.0089 | 0.0240 | 0.0097 | 0.0120 | 0.0069 | 0.0200 | 0.0089 | 0.0040 | 0.0040 | 0.0000 | 0.0000 | 0.0040 | 0.0040 | 0.0040 | 0.0040 |
| MaLA-500 Llama 2 10B v2 | 0.0091 | 0.0000 | 0.0000 | 0.0080 | 0.0056 | 0.0000 | 0.0000 | 0.0120 | 0.0089 | 0.0240 | 0.0097 | 0.0120 | 0.0069 | 0.0200 | 0.0089 | 0.0040 | 0.0040 | 0.0000 | 0.0000 | 0.0040 | 0.0040 | 0.0040 | 0.0040 |
| YaYi Llama 2 7B | 0.0709 | 0.0320 | 0.0112 | 0.0840 | 0.0176 | 0.1560 | 0.0230 | 0.1600 | 0.0232 | 0.1040 | 0.0193 | 0.0320 | 0.0141 | 0.0560 | 0.0146 | 0.0080 | 0.0056 | 0.0120 | 0.0069 | 0.0040 | 0.0040 | 0.1120 | 0.0200 |
| TowerBase Llama 2 7B | 0.0615 | 0.0240 | 0.0097 | 0.0840 | 0.0176 | 0.1160 | 0.0203 | 0.0920 | 0.0183 | 0.0880 | 0.0180 | 0.0480 | 0.0135 | 0.1000 | 0.0190 | 0.0120 | 0.0069 | 0.0080 | 0.0056 | 0.0160 | 0.0080 | 0.0880 | 0.0180 |
| TowerInstruct Llama 2 7B | 0.0724 | 0.0160 | 0.0080 | 0.1000 | 0.0190 | 0.1520 | 0.0228 | 0.1560 | 0.0230 | 0.1280 | 0.0212 | 0.0160 | 0.0080 | 0.1000 | 0.0190 | 0.0160 | 0.0080 | 0.0200 | 0.0089 | 0.0200 | 0.0089 | 0.0720 | 0.0164 |
| Occiglot Mistral 7B v0.1 | 0.1331 | 0.0320 | 0.0112 | 0.2120 | 0.0259 | 0.3000 | 0.0290 | 0.2720 | 0.0282 | 0.2160 | 0.0261 | 0.0640 | 0.0155 | 0.1520 | 0.0228 | 0.0240 | 0.0097 | 0.0160 | 0.0080 | 0.0800 | 0.0172 | 0.0960 | 0.0187 |
| Occiglot Mistral 7B v0.1 Instruct | 0.2276 | 0.0480 | 0.0135 | 0.3400 | 0.0300 | 0.4640 | 0.0316 | 0.4000 | 0.0310 | 0.3160 | 0.0295 | 0.1840 | 0.0246 | 0.2360 | 0.0269 | 0.0640 | 0.0155 | 0.0280 | 0.0105 | 0.1120 | 0.0200 | 0.3120 | 0.0294 |
| BLOOM 7B | 0.0287 | 0.0240 | 0.0097 | 0.0160 | 0.0080 | 0.0400 | 0.0124 | 0.0360 | 0.0118 | 0.0120 | 0.0069 | 0.0200 | 0.0089 | 0.0360 | 0.0118 | 0.0160 | 0.0080 | 0.0124 | 0.0360 | 0.0089 | 0.0360 | 0.0118 |  |
| BLOOMZ 7B | 0.0255 | 0.0320 | 0.0112 | 0.0160 | 0.0080 | 0.0360 | 0.0118 | 0.0240 | 0.0097 | 0.0320 | 0.0112 | 0.0280 | 0.0105 | 0.0360 | 0.0118 | 0.0280 | 0.0105 | 0.0120 | 0.0069 | 0.0120 | 0.0069 | 0.0240 | 0.0097 |
| nGPT | 0.0135 | 0.0000 | 0.0000 | 0.0200 | 0.0089 | 0.0320 | 0.0112 | 0.0080 | 0.0056 | 0.0240 | 0.0097 | 0.0160 | 0.0080 | 0.0240 | 0.0097 | 0.0160 | 0.0080 | 0.0000 | 0.0000 | 0.0000 | 0.0000 | 0.0000 | 0.0000 |
| nGPT 13B | 0.0131 | 0.0000 | 0.0000 | 0.0160 | 0.0080 | 0.0160 | 0.0080 | 0.0080 | 0.0056 | 0.0360 | 0.0118 | 0.0160 | 0.0080 | 0.0200 | 0.0089 | 0.0120 | 0.0069 | 0.0000 | 0.0000 | 0.0080 | 0.0056 | 0.0120 | 0.0069 |
| XGLM 7.5B | 0.0102 | 0.0000 | 0.0000 | 0.0080 | 0.0056 | 0.0000 | 0.0000 | 0.0120 | 0.0089 | 0.0000 | 0.0000 | 0.0040 | 0.0200 | 0.0040 | 0.0040 | 0.0200 | 0.0089 | 0.0000 | 0.0000 | 0.0080 | 0.0056 | 0.0280 | 0.0105 |
| YaYi 7B | 0.0276 | 0.0240 | 0.0097 | 0.0200 | 0.0089 | 0.0600 | 0.0151 | 0.0240 | 0.0097 | 0.0560 | 0.0146 | 0.0160 | 0.0080 | 0.0120 | 0.0069 | 0.0120 | 0.0069 | 0.0097 | 0.0080 | 0.0480 | 0.0135 |  |  |
| Llama 3 8B | 0.2745 | 0.1760 | 0.0241 | 0.3960 | 0.0310 | 0.5080 | 0.0317 | 0.4720 | 0.0316 | 0.3640 | 0.0305 | 0.0360 | 0.0118 | 0.3760 | 0.0307 | 0.2400 | 0.0271 | 0.0560 | 0.0146 | 0.3680 | 0.0306 | 0.0280 | 0.0105 |
| Llama 3.1 8B | 0.2836 | 0.2000 | 0.0253 | 0.4120 | 0.0312 | 0.5520 | 0.0315 | 0.4840 | 0.0317 | 0.3840 | 0.0308 | 0.0360 | 0.0118 | 0.4000 | 0.0310 | 0.0280 | 0.0257 | 0.1040 | 0.0130 | 0.3760 | 0.0307 | 0.0240 | 0.0097 |
| Gemma 7B | 0.3822 | 0.3440 | 0.0301 | 0.4480 | 0.0315 | 0.5880 | 0.0312 | 0.4800 | 0.0317 | 0.3960 | 0.0310 | 0.1680 | 0.0237 | 0.4120 | 0.0312 | 0.3760 | 0.0307 | 0.2720 | 0.0282 | 0.4280 | 0.0314 | 0.2920 | 0.0288 |
| Gemma 2 9B | 0.4895 | 0.4440 | 0.0315 | 0.6560 | 0.0301 | 0.8080 | 0.0250 | 0.7600 | 0.0271 | 0.6960 | 0.0292 | 0.0160 | 0.0080 | 0.0672 | 0.0298 | 0.1600 | 0.0232 | 0.1480 | 0.0225 | 0.5600 | 0.0315 | 0.4640 | 0.0316 |
| Qwen 2 7B | 0.3295 | 0.2960 | 0.0289 | 0.4040 | 0.0311 | 0.5640 | 0.0314 | 0.5080 | 0.0317 | 0.3600 | 0.0304 | 0.0120 | 0.0069 | 0.3520 | 0.0303 | 0.3760 | 0.0307 | 0.3080 | 0.0293 | 0.4000 | 0.0310 | 0.0440 | 0.0130 |
| Qwen 1.5 7B | 0.3156 | 0.1240 | 0.0209 | 0.4400 | 0.0315 | 0.5520 | 0.0315 | 0.4840 | 0.0317 | 0.4520 | 0.0315 | 0.1680 | 0.0237 | 0.4320 | 0.0720 | 0.0164 | 0.0400 | 0.0124 | 0.2840 | 0.0286 | 0.4240 | 0.0313 |  |
| EMMA-500 Llama 2 7B | 0.1702 | 0.0880 | 0.0180 | 0.2320 | 0.0268 | 0.3400 | 0.0300 | 0.2800 | 0.0285 | 0.2560 | 0.0277 | 0.0920 | 0.0183 | 0.2280 | 0.0266 | 0.1680 | 0.0237 | 0.0240 | 0.0097 | 0.1000 | 0.0190 | 0.0640 | 0.0155 |

Table 21: 3-shot results (ACC) on MGSM by CoT prompting and flexible matching.

| Model | Avg | bn | bn-stderr | de | de-stderr | en | en-stderr | es | es-stderr | fr | fr-stderr | ja | ja-stderr | ru | ru-stderr | sw | sw-stderr | te | te-stderr | th | th-stderr | zh | zh-stderr |
|---|---|---|---|---|---|---|---|---|---|---|---|---|---|---|---|---|---|---|---|---|---|---|---|
| Llama 2 7B | 0.0636 | 0.0200 | 0.0089 | 0.0760 | 0.0168 | 0.1600 | 0.0232 | 0.1240 | 0.0209 | 0.0920 | 0.0183 | 0.0360 | 0.0118 | 0.0800 | 0.0172 | 0.0200 | 0.0089 | 0.0080 | 0.0056 | 0.0160 | 0.0105 | 0.0800 | 0.0172 |
| Llama 2 7B Chat | 0.1091 | 0.0240 | 0.0097 | 0.1760 | 0.0241 | 0.2720 | 0.0282 | 0.1920 | 0.0250 | 0.1880 | 0.0248 | 0.0400 | 0.0124 | 0.1520 | 0.0228 | 0.0200 | 0.0080 | 0.0056 | 0.0280 | 0.0105 | 0.1160 | 0.0203 |  |
| CodeLlama 2 7B | 0.0664 | 0.0160 | 0.0080 | 0.0920 | 0.0183 | 0.1320 | 0.0215 | 0.1080 | 0.0197 | 0.1040 | 0.0197 | 0.0440 | 0.0130 | 0.0760 | 0.0168 | 0.0120 | 0.0069 | 0.0160 | 0.0080 | 0.0600 | 0.0151 | 0.0400 | 0.0124 |
| LLaMAX Llama 2 7B | 0.0362 | 0.0360 | 0.0118 | 0.0360 | 0.0118 | 0.0720 | 0.0164 | 0.0360 | 0.0118 | 0.0520 | 0.0141 | 0.0280 | 0.0105 | 0.0280 | 0.0105 | 0.0080 | 0.0056 | 0.0080 | 0.0056 | 0.0200 | 0.0480 | 0.0135 |  |
| LLaMAX Llama 2 7B Alpaca | 0.0635 | 0.0320 | 0.0112 | 0.0480 | 0.0135 | 0.1520 | 0.0228 | 0.0960 | 0.0187 | 0.0520 | 0.0141 | 0.0440 | 0.0130 | 0.0360 | 0.0118 | 0.0480 | 0.0135 | 0.0040 | 0.0480 | 0.0135 | 0.0680 | 0.0160 |  |
| MaLA-500 Llama 2 10B v1 | 0.0073 | 0.0000 | 0.0000 | 0.0040 | 0.0040 | 0.0040 | 0.0040 | 0.0040 | 0.0040 | 0.0160 | 0.0080 | 0.0080 | 0.0240 | 0.0097 | 0.0040 | 0.0000 | 0.0000 | 0.0000 | 0.0080 | 0.0080 | 0.0056 |  |  |
| MaLA-500 Llama 2 10B v2 | 0.0073 | 0.0000 | 0.0000 | 0.0040 | 0.0040 | 0.0040 | 0.0040 | 0.0040 | 0.0040 | 0.0160 | 0.0080 | 0.0080 | 0.0240 | 0.0097 | 0.0040 | 0.0040 | 0.0000 | 0.0000 | 0.0000 | 0.0080 | 0.0056 |  |  |
| YaYi Llama 2 7B | 0.0722 | 0.0280 | 0.0105 | 0.0840 | 0.0176 | 0.1680 | 0.0237 | 0.1240 | 0.0209 | 0.1040 | 0.0193 | 0.0400 | 0.0135 | 0.0720 | 0.0164 | 0.0200 | 0.0089 | 0.0120 | 0.0069 | 0.0320 | 0.1240 | 0.0209 |  |
| TowerBase Llama 2 7B | 0.0616 | 0.0360 | 0.0118 | 0.0800 | 0.0172 | 0.1120 | 0.0200 | 0.0840 | 0.0176 | 0.0760 | 0.0168 | 0.0360 | 0.0118 | 0.0840 | 0.0176 | 0.0120 | 0.0069 | 0.0080 | 0.0056 | 0.0280 | 0.0105 | 0.0920 | 0.0183 |
| TowerInstruct Llama 2 7B | 0.0824 | 0.0120 | 0.0069 | 0.1280 | 0.0212 | 0.1800 | 0.0248 | 0.1520 | 0.0228 | 0.1200 | 0.0206 | 0.0240 | 0.0097 | 0.1360 | 0.0217 | 0.0160 | 0.0080 | 0.0120 | 0.0069 | 0.0320 | 0.1120 | 0.0200 |  |
| Occiglot Mistral 7B v0.1 | 0.1407 | 0.0240 | 0.0097 | 0.2160 | 0.0261 | 0.3320 | 0.0298 | 0.2680 | 0.0281 | 0.2240 | 0.0264 | 0.0640 | 0.0155 | 0.1720 | 0.0239 | 0.0320 | 0.0112 | 0.0120 | 0.0069 | 0.0600 | 0.0151 | 0.1120 | 0.0200 |
| Occiglot Mistral 7B v0.1 Instruct | 0.2216 | 0.0400 | 0.0124 | 0.3320 | 0.0298 | 0.4320 | 0.0314 | 0.4200 | 0.0313 | 0.3160 | 0.0295 | 0.1680 | 0.0237 | 0.2440 | 0.0272 | 0.0560 | 0.0146 | 0.0160 | 0.0080 | 0.0480 | 0.2480 | 0.0274 |  |
| BLOOM 7B | 0.0229 | 0.0200 | 0.0089 | 0.0160 | 0.0080 | 0.0480 | 0.0135 | 0.0240 | 0.0097 | 0.0240 | 0.0097 | 0.0040 | 0.0040 | 0.0400 | 0.0130 | 0.0160 | 0.0080 | 0.0200 | 0.0089 | 0.0200 | 0.0240 | 0.0097 |  |
| BLOOMZ 7B | 0.0215 | 0.0200 | 0.0089 | 0.0200 | 0.0089 | 0.0240 | 0.0097 | 0.0320 | 0.0112 | 0.0320 | 0.0112 | 0.0200 | 0.0089 | 0.0160 | 0.0080 | 0.0240 | 0.0097 | 0.0200 | 0.0089 | 0.0160 | 0.0080 | 0.0120 | 0.0069 |
| nGPT | 0.0142 | 0.0000 | 0.0000 | 0.0200 | 0.0089 | 0.0200 | 0.0089 | 0.0160 | 0.0080 | 0.0240 | 0.0097 | 0.0120 | 0.0069 | 0.0200 | 0.0089 | 0.0120 | 0.0069 | 0.0080 | 0.0105 | 0.0240 | 0.0160 | 0.0080 |  |
| nGPT 13B | 0.0153 | 0.0000 | 0.0000 | 0.0200 | 0.0089 | 0.0240 | 0.0097 | 0.0160 | 0.0080 | 0.0240 | 0.0097 | 0.0120 | 0.0069 | 0.0160 | 0.0080 | 0.0200 | 0.0089 | 0.0160 | 0.0080 | 0.0080 | 0.0280 | 0.0105 |  |
| XGLM 7.5B | 0.0116 | 0.0000 | 0.0000 | 0.0120 | 0.0000 | 0.0000 | 0.0000 | 0.0080 | 0.0056 | 0.0000 | 0.0000 | 0.0040 | 0.0160 | 0.0080 | 0.0240 | 0.0097 | 0.0160 | 0.0080 | 0.0080 | 0.0280 | 0.0105 |  |  |
| YaYi 7B | 0.0302 | 0.0480 | 0.0135 | 0.0320 | 0.0112 | 0.0480 | 0.0135 | 0.0360 | 0.0118 | 0.0400 | 0.0124 | 0.0200 | 0.0089 | 0.0120 | 0.0069 | 0.0160 | 0.0080 | 0.0080 | 0.0056 | 0.0600 | 0.0151 |  |  |
| Llama 3 8B | 0.2813 | 0.1920 | 0.0250 | 0.3840 | 0.0308 | 0.5400 | 0.0316 | 0.4600 | 0.0316 | 0.3600 | 0.0304 | 0.0200 | 0.0089 | 0.3800 | 0.0308 | 0.2520 | 0.0275 | 0.0640 | 0.0155 | 0.3920 | 0.0309 | 0.0400 | 0.0124 |
| Llama 3.1 8B | 0.2731 | 0.0250 | 0.0420 | 0.0313 | 0.5280 | 0.0316 | 0.4280 | 0.0316 | 0.3480 | 0.0302 | 0.0200 | 0.0105 | 0.3720 | 0.0306 | 0.2520 | 0.0275 | 0.0466 | 0.0146 | 0.3560 | 0.0303 | 0.0160 | 0.0080 |
| Gemma 7B | 0.3578 | 0.3640 | 0.0305 | 0.4080 | 0.0311 | 0.6080 | 0.0309 | 0.4520 | 0.0315 | 0.4200 | 0.0313 | 0.0600 | 0.0118 | 0.3920 | 0.0309 | 0.3880 | 0.0309 | 0.2480 | 0.0274 | 0.4440 | 0.0315 | 0.0480 | 0.0135 |
| Gemma 2 9B | 0.4469 | 0.4880 | 0.0317 | 0.5360 | 0.0316 | 0.7200 | 0.0285 | 0.6040 | 0.0310 | 0.5000 | 0.0317 | 0.0200 | 0.5000 | 0.0317 | 0.5360 | 0.0316 | 0.4440 | 0.0315 | 0.5520 | 0.0315 | 0.0160 | 0.0080 |  |
| Qwen 2 7B | 0.5147 | 0.4520 | 0.0315 | 0.6600 | 0.0300 | 0.8200 | 0.0243 | 0.7520 | 0.0277 | 0.7160 | 0.0286 | 0.0281 | 0.0105 | 0.6840 | 0.0295 | 0.1720 | 0.0239 | 0.1760 | 0.0241 | 0.5840 | 0.0312 | 0.6080 | 0.0309 |
| Qwen 1.5 7B | 0.3036 | 0.1280 | 0.0212 | 0.4320 | 0.0314 | 0.5840 | 0.0312 | 0.5040 | 0.0317 | 0.4200 | 0.0313 | 0.1120 | 0.0200 | 0.3680 | 0.0306 | 0.0880 | 0.0180 | 0.0400 | 0.0124 | 0.2560 | 0.0277 | 0.2600 | 0.0278 |
| EMMA-500 Llama 2 7B | 0.1809 | 0.0840 | 0.0176 | 0.2240 | 0.0264 | 0.3760 | 0.0307 | 0.2560 | 0.0277 | 0.2160 | 0.0261 | 0.0800 | 0.0172 | 0.2280 | 0.0266 | 0.2120 | 0.0259 | 0.0240 | 0.0097 | 0.1160 | 0.0203 | 0.1640 | 0.0235 |

Table 22: 0-shot results (ACC) on BELEBELE.

| Model | Avg | High | Medium-High | Medium | Medium-Low | Low |
|---|---|---|---|---|---|---|
| Llama 2 7B | 0.2627 | 0.2676 | 0.2635 | 0.2607 | 0.2641 | 0.2627 |
| Llama 2 7B Chat | 0.2905 | 0.3184 | 0.2997 | 0.2895 | 0.2947 | 0.2909 |
| CodeLlama 2 7B | 0.2738 | 0.2737 | 0.2733 | 0.2730 | 0.2730 | 0.2738 |
| LLaMAX Llama 2 7B | 0.2309 | 0.2323 | 0.2315 | 0.2307 | 0.2310 | 0.2308 |
| LLaMAX Llama 2 7B Alpaca | 0.2448 | 0.2546 | 0.2482 | 0.2441 | 0.2460 | 0.2449 |
| MaLA-500 Llama 2 10B v1 | 0.2296 | 0.2302 | 0.2298 | 0.2297 | 0.2298 | 0.2297 |
| MaLA-500 Llama 2 10B v2 | 0.2296 | 0.2302 | 0.2298 | 0.2297 | 0.2298 | 0.2297 |
| YaYi Llama 2 7B | 0.2832 | 0.2964 | 0.2867 | 0.2811 | 0.2837 | 0.2826 |
| TowerBase Llama 2 7B | 0.2636 | 0.2743 | 0.2685 | 0.2629 | 0.2648 | 0.2634 |
| TowerInstruct Llama 2 7B | 0.2793 | 0.2988 | 0.2851 | 0.2757 | 0.2819 | 0.2792 |
| Occiglot Mistral 7B v0.1 | 0.3016 | 0.3225 | 0.3094 | 0.3002 | 0.3040 | 0.3015 |
| Occiglot Mistral 7B v0.1 Instruct | 0.3205 | 0.3414 | 0.3262 | 0.3174 | 0.3240 | 0.3208 |
| BLOOM 7B | 0.2411 | 0.2425 | 0.2452 | 0.2412 | 0.2411 | 0.2408 |
| BLOOMZ 7B | 0.3932 | 0.4543 | 0.4367 | 0.4151 | 0.4008 | 0.3951 |
| mGPT | 0.2396 | 0.2401 | 0.2381 | 0.2400 | 0.2394 | 0.2398 |
| XGLM 7.5B | 0.2485 | 0.2552 | 0.2517 | 0.2489 | 0.2490 | 0.2485 |
| YaYi 7B | 0.3797 | 0.4437 | 0.4271 | 0.4049 | 0.3872 | 0.3809 |
| Llama 3 8B | 0.4073 | 0.4607 | 0.4292 | 0.4103 | 0.4187 | 0.4088 |
| Llama 3.1 8B | 0.4519 | 0.5250 | 0.4801 | 0.4565 | 0.4674 | 0.4534 |
| Gemma 7B | 0.4337 | 0.5263 | 0.4783 | 0.4482 | 0.4543 | 0.4394 |
| Gemma 2 9B | 0.5449 | 0.6410 | 0.5890 | 0.5583 | 0.5685 | 0.5505 |
| Qwen 2 7B | 0.4931 | 0.5762 | 0.5220 | 0.5004 | 0.5116 | 0.4948 |
| Qwen 1.5 7B | 0.4183 | 0.4886 | 0.4479 | 0.4218 | 0.4300 | 0.4178 |
| EMMA-500 Llama 2 7B | 0.2675 | 0.2832 | 0.2818 | 0.2758 | 0.2714 | 0.2694 |

**Continual Pre-training** Continual pre-training has become a popular strategy to adapt LLMs to new languages and domains without retraining from scratch. The process involves updating model parameters incrementally using a relatively small amount of new data from target languages. Recent studies, such as those by Tejaswi et al. (2024), have demonstrated the effectiveness of CPT in improving the adaptability of LLMs in low-resource language settings. Techniques such as vocabulary augmentation and targeted domain-specific pre-training have been shown to significantly improve both efficiency and performance, particularly when large multilingual models are adapted to new

Table 23: 5-shot results (ACC) on ARC multilingual.

| Model | Avg | High | Medium | Low |
|---|---|---|---|---|
| Llama 2 7B | 0.2756 | 0.3312 | 0.2731 | 0.2102 |
| Llama 2 7B Chat | 0.2802 | 0.3369 | 0.2779 | 0.2129 |
| CodeLlama 2 7B | 0.2523 | 0.2886 | 0.2464 | 0.2165 |
| LLaMAX Llama 2 7B | 0.2609 | 0.3000 | 0.2592 | 0.2148 |
| LLaMAX Llama 2 7B Alpaca | 0.3106 | 0.3689 | 0.3185 | 0.2249 |
| MaLA-500 Llama 2 10B v1 | 0.2116 | 0.2192 | 0.2048 | 0.2132 |
| MaLA-500 Llama 2 10B v2 | 0.2116 | 0.2192 | 0.2048 | 0.2132 |
| YaYi Llama 2 7B | 0.2840 | 0.3430 | 0.2835 | 0.2111 |
| TowerBase Llama 2 7B | 0.2794 | 0.3532 | 0.2682 | 0.2051 |
| TowerInstruct Llama 2 7B | 0.3010 | 0.3888 | 0.2885 | 0.2116 |
| Occiglot Mistral 7B v0.1 | 0.2977 | 0.3839 | 0.2851 | 0.2103 |
| Occiglot Mistral 7B v0.1 Instruct | 0.3088 | 0.4029 | 0.2965 | 0.2113 |
| BLOOM 7B | 0.2365 | 0.2627 | 0.2272 | 0.2189 |
| BLOOMZ 7B | 0.2395 | 0.2694 | 0.2274 | 0.2218 |
| mGPT | 0.2024 | 0.2011 | 0.1965 | 0.2137 |
| mGPT 13B | 0.2176 | 0.2299 | 0.2114 | 0.2123 |
| XGLM 7.5B | 0.2221 | 0.2461 | 0.2105 | 0.2110 |
| YaYi 7B | 0.2444 | 0.2796 | 0.2329 | 0.2191 |
| Llama 3 8B | 0.3480 | 0.4243 | 0.3553 | 0.2406 |
| Llama 3.1 8B | 0.3493 | 0.4243 | 0.3589 | 0.2400 |
| Gemma 7B | 0.3868 | 0.4646 | 0.4047 | 0.2606 |
| Gemma 2 9B | 0.4415 | 0.5459 | 0.4618 | 0.2782 |
| Qwen 2 7B | 0.3382 | 0.4388 | 0.3264 | 0.2317 |
| Qwen 1.5 7B | 0.2893 | 0.3555 | 0.2814 | 0.2192 |
| EMMA-500 Llama 2 7B | 0.2953 | 0.3410 | 0.2982 | 0.2334 |

Table 24: Results on Multipl-E. For language-level breakdowns, refer to Tables 25 to 27 in the appendix.

| Model | Avg Pass@1 | Avg Pass@10 | Avg Pass@25 |
|---|---|---|---|
| Llama 2 7B | 8.92% | 19.45% | 25.68% |
| CodeLlama 2 7B | 28.43% | 50.83% | 63.92% |
| LLaMAX 2 7B | 0.35% | 1.61% | 2.67% |
| MaLA-500 Llama 2 10B V2 | 0.0% | 0.0% | 0.0% |
| TowerBase Llama 2 7B | 3.61% | 6.65% | 8.97% |
| Occiglot Mistral 7B v0.1 | 21.26% | 31.37% | 45.86% |
| Bloom 7B | 5.34% | 10.49% | 14.65% |
| BloomZ 2 7B | 5.85% | 11.40% | 15.76% |
| Aya23 8B | 9.19% | 23.52% | 32.09% |
| Mistral 7B v0.3 | 26.10% | 48.68% | 59.05% |
| Llama 3 8B | 30.09% | 53.82% | 64.01% |
| LLaMAX 3 8B | 3.00% | 7.23% | 10.67% |
| Gemma 7B | 28.55% | 54.27% | 64.75% |
| CodeGemma 7B | 31.51% | 63.13% | 72.65% |
| Qwen 1.5 7B | 21.05% | 37.19% | 47.63% |
| Qwen 2 7B | 38.68% | 62.63% | 71.55% |
| EMMA-500 Llama 2 7B | 11.38% | 19.02% | 26.16% |

languages. Despite its benefits, CPT can lead to catastrophic forgetting, where models lose previously learned information. To address the potential degraded performance issue, Ibrahim et al. (2024) presents a simple yet effective approach to continual pre-train models, demonstrating that with a combination of learning rate re-warming, re-decaying, and replay of previous data, it is possible to match the performance of fully re-trained models. Also, recent studies have delved into the effectiveness of continual pre-training with parallel data. As highlighted in the study by Gilabert et al. (2024), the use of a Catalan-centric parallel dataset has enabled the training of models good at trasnalting in various directions. Besides, the research by Kondo et al. (2024), proposed a two-phase continual training approach with parallel data. In the first phase, a pre-trained LLM is continually pre-trained on parallel data, followed by a second phase of supervised fine-tuning with a small amount of high-quality parallel data. Their experiments with a 3.8B-parameter model across various data formats revealed that alternating between source and target sentences during continual pre-training is crucial for enhancing translation accuracy in the corresponding direction.

Table 25: Pass@1 on Multipl-E.

| Model | C++ | C# | Java | JavaScript | Python | Rust | TypeScript |
|---|---|---|---|---|---|---|---|
| Llama 2 7B | 6.74% | 5.65% | 8.54% | 11.34% | 11.98% | 6.12% | 12.04% |
| CodeLlama 2 7B | 26.70% | 20.44% | 30.56% | 32.89% | 28.76% | 26.23% | 33.45% |
| LLaMAX 2 7B | 0.0% | 0.0% | 0.02% | 0.68% | 0.0% | 0.0% | 1.78% |
| MaLA-500 Llama 2 10B V2 | 0.0% | 0.0% | 0.0% | 0.0% | 0.0% | 0.0% | 0.0% |
| TowerBase Llama 2 7B | 1.01% | 6.85% | 7.86% | 0.87% | 8.68% | 0.0% | 0.0% |
| Occiglot Mistral 7B v0.1 | 23.16% | 16.14% | 19.83% | 26.92% | 21.45% | 16.34% | 24.97% |
| Bloom 7B | 5.25% | 2.97% | 6.37% | 6.76% | 7.65% | 1.01% | 7.34% |
| BloomZ 2 7B | 6.87% | 3.59% | 6.02% | 6.96% | 7.33% | 2.13% | 8.04% |
| Aya23 8B | 16.03% | 7.91% | 14.29% | 6.23% | 3.56% | 11.43% | 4.90% |
| Mistral 7B v0.3 | 26.12% | 22.87% | 25.54% | 35.24% | 24.91% | 19.24% | 28.76% |
| Llama 3 8B | 34.12% | 21.06% | 26.56% | 36.12% | 30.22% | 25.19% | 37.34% |
| LLaMAX 3 8B | 0.0% | 0.0% | 0.0% | 9.73% | 0.91% | 0.21% | 10.12% |
| Gemma 7B | 29.66% | 21.40% | 27.35% | 35.29% | 30.11% | 25.48% | 30.57% |
| CodeGemma 7B | 33.91% | 21.05% | 29.43% | 37.78% | 32.56% | 28.70% | 37.14% |
| Qwen 1.5 7B | 22.04% | 12.42% | 17.68% | 27.58% | 32.11% | 8.32% | 27.21% |
| Qwen 2 7B | 43.51% | 21.47% | 38.95% | 46.31% | 37.49% | 34.61% | 48.41% |
| EMMA-500 Llama 2 7B | 11.34% | 8.94% | 11.93% | 11.67% | 18.97% | 6.86% | 9.94% |

Table 26: Pass@10 on Multipl-E.

| Model | C++ | C# | Java | JavaScript | Python | Rust | TypeScript |
|---|---|---|---|---|---|---|---|
| Llama 2 7B | 17.92% | 14.22% | 20.78% | 23.12% | 24.86% | 13.08% | 22.19% |
| CodeLlama 2 7B | 51.39% | 37.13% | 50.38% | 59.08% | 56.58% | 47.68% | 53.60% |
| LLaMAX 2 7B | 0.96% | 0.0% | 0.53% | 3.99% | 0.0% | 0.12% | 5.64% |
| MaLA-500 Llama 2 10B V2 | 0.0% | 0.0% | 0.0% | 0.0% | 0.0% | 0.0% | 0.0% |
| TowerBase Llama 2 7B | 1.01% | 6.85% | 7.86% | 0.87% | 8.68% | 0.0% | 0.0% |
| Occiglot Mistral 7B v0.1 | 36.08% | 27.49% | 35.87% | 47.45% | 41.76% | 29.80% | 43.13% |
| Bloom 7B | 11.35% | 6.86% | 12.79% | 12.44% | 14.24% | 3.68% | 12.08% |
| BloomZ 2 7B | 12.55% | 8.21% | 12.94% | 14.10% | 14.84% | 3.91% | 13.27% |
| Aya23 8B | 28.49% | 17.34% | 27.12% | 23.19% | 26.72% | 26.14% | 15.67% |
| Mistral 7B v0.3 | 50.23% | 35.18% | 46.83% | 57.23% | 54.29% | 42.77% | 54.22% |
| Llama 3 8B | 55.13% | 36.67% | 54.34% | 62.65% | 59.24% | 45.68% | 63.02% |
| LLaMAX 3 8B | 0.72% | 0.0% | 0.97% | 22.91% | 1.58% | 1.38% | 23.04% |
| Gemma 7B | 55.21% | 39.09% | 52.02% | 61.88% | 60.09% | 50.34% | 61.23% |
| CodeGemma 7B | 62.29% | 45.11% | 59.74% | 70.96% | 69.76% | 61.27% | 72.76% |
| Qwen 1.5 7B | 40.11% | 28.95% | 37.56% | 48.35% | 40.34% | 20.19% | 44.85% |
| Qwen 2 7B | 64.58% | 43.17% | 63.28% | 73.21% | 54.34% | 65.48% | 74.32% |
| EMMA-500 Llama 2 7B | 22.84% | 14.29% | 21.28% | 18.22% | 24.94% | 15.68% | 15.91% |

Table 27: Pass@25 on Multipl-E.

| Model | C++ | C# | Java | JavaScript | Python | Rust | TypeScript |
|---|---|---|---|---|---|---|---|
| Llama 2 7B | 24.89% | 18.77% | 26.45% | 30.31% | 31.09% | 18.70% | 29.55% |
| CodeLlama 2 7B | 64.12% | 45.96% | 61.98% | 72.76% | 71.77% | 59.98% | 70.89% |
| LLaMAX 2 7B | 1.47% | 0.0% | 1.04% | 6.78% | 0.0% | 0.34% | 9.04% |
| MaLA-500 Llama 2 10B V2 | 0.0% | 0.0% | 0.0% | 0.0% | 0.0% | 0.0% | 0.0% |
| TowerBase Llama 2 7B | 1.86% | 15.74% | 22.45% | 2.53% | 20.18% | 0.0% | 0.0% |
| Occiglot Mistral 7B v0.1 | 44.17% | 33.89% | 42.77% | 56.63% | 49.87% | 37.94% | 55.76% |
| Bloom 7B | 17.27% | 10.07% | 17.23% | 17.55% | 17.93% | 5.70% | 16.81% |
| BloomZ 2 7B | 17.41% | 11.98% | 18.66% | 18.37% | 18.78% | 6.83% | 18.31% |
| Aya23 8B | 37.56% | 21.87% | 36.45% | 33.57% | 36.09% | 35.34% | 23.75% |
| Mistral 7B v0.3 | 60.93% | 44.80% | 56.12% | 66.34% | 65.33% | 55.31% | 64.51% |
| Llama 3 8B | 64.42% | 43.34% | 62.14% | 73.51% | 71.82% | 57.78% | 75.09% |
| LLaMAX 3 8B | 1.71% | 0.0% | 2.32% | 35.61% | 2.77% | 2.08% | 30.19% |
| Gemma 7B | 66.14% | 47.08% | 63.24% | 70.22% | 70.47% | 63.52% | 72.56% |
| CodeGemma 7B | 73.66% | 50.79% | 69.96% | 79.48% | 78.68% | 74.87% | 81.12% |
| Qwen 1.5 7B | 51.87% | 37.69% | 48.71% | 59.71% | 49.69% | 29.78% | 55.98% |
| Qwen 2 7B | 74.33% | 51.87% | 72.03% | 81.67% | 65.38% | 74.97% | 80.59% |
| EMMA-500 Llama 2 7B | 31.93% | 18.69% | 29.85% | 26.45% | 32.55% | 21.32% | 22.34% |

