# OpenReview forum: "EMMA-500: Enhancing Massively Multilingual Adaptation of Large Language Models"
_ICLR.cc/2025/Conference — Submitted to ICLR 2025_

### Official Review · Reviewer_YEQS · 2024-10-26

**Soundness:** 3
**Presentation:** 2
**Contribution:** 3
**Rating:** 5
**Confidence:** 4

**Summary:**

This paper proposes a multilingual dataset named "the MaLA corpus" by augmenting and merging existing datasets, a multilingual model by continue pre-training on the processed dataset using LLaMA-2-7b, and a open-ended multilingual benchmark generated by ChatGPT. The main contents of this paper are used to illustrate the common pipeline of processing data and report results on multilingual benchmark. The only finding in this paper is that the multilingual capacities of LLaMA-2-7b can be improved by continue pre-training on multilingual corpus, which is trivial.

**Strengths:**

- The MaLA corpus compiled and PolyWrite multilingual benchmark will be beneficial to multilingual NLP researchers if they are public.
- Extensive experiments are conducted to show the effectiveness of EMMA-500.

**Weaknesses:**

- [Addressed] Limited novelty: the corpus are generated by merging and cleaning existing datasets. The method to create multilingual benchmark is not novel, which is commonly adopted in the multilingual NLP community. It is an incremental work on augmenting the multilingual corpus and benchmark.
- Lacking baseline: some models of continue pre-training on LLaMA-2-7b are missing like ALMA[1] and Bayling[2].

[1]. Xu, H., Kim, Y. J., Sharaf, A., & Awadalla, H. H. A Paradigm Shift in Machine Translation: Boosting Translation Performance of Large Language Models. ICLR 2024.

[2]. Zhang, S., Fang, Q., Zhang, Z., Ma, Z., Zhou, Y., Huang, L., ... & Feng, Y. (2023). Bayling: Bridging cross-lingual alignment and instruction following through interactive translation for large language models. arXiv preprint arXiv:2306.10968.

**Questions:**

1. Will this corpus, model weight, benchmark and processing scripts be public?
2. Line 121-122: "This issue is resolved ...", Which issue has been resolved?
3. Line 310-311: Why is the global batch size set to 16M, rather than the commonly used 4M tokens like LLaMA-2?

---

> ### Author Response · Authors · 2024-11-18
>
> **Q:** Regarding novelty
> **A:** While it's true that the corpus is constructed by merging and cleaning existing datasets, and the approach to creating a multilingual benchmark is not groundbreaking in itself, the contribution of this work should not be understated. The **scale and diversity of the MaLA corpus**, along with the strategic focus on low-resource languages, represent a significant advancement over prior efforts. Additionally, **the integration of different text types**, including code, scientific papers, and instructions, offers a more comprehensive and effective foundation for continual pre-training. The enhanced MaLA corpus **significantly boosts the multilingual capabilities of models**, particularly in low-resource languages, leading to measurable improvements in tasks like machine translation, text classification, and more. This work has the potential to drive substantial improvements in multilingual NLP, especially for underrepresented languages, which makes it highly relevant and impactful within the field.
>
> **Q:** Baselines like ALMA and Bayling
> **A:** ALMA and Bayling are heavily optimized for translation only, similar to the Tower model, for which we offered a comparison on MT. According to the Tower paper, Tower is slightly better than ALMA, so we omitted ALMA and Bayling. Further, the primary contribution of this work is the MaLA corpus and its impact on multilingual continued pre-training which yielded a massive multilingual base model beyond just the translation task.
>
> **Q:** Will this corpus, model weight, benchmark and processing scripts be public?
> **A:** Yes.
>
> **Q:** Line 121-122: "This issue is resolved ...", Which issue has been resolved?
> **A:** The issue involves text records consisting solely of date and timestamp information in the dataset for the Languages of Russia.
>
> **Q:** Line 310-311: Why is the global batch size set to 16M, rather than the commonly used 4M tokens like LLaMA-2?
> **A:** This is a technical choice that allows us to efficiently use our computing. We note that increasing batch size during training is seen in LLM pre-training now, e.g. Llama-3 doubled their batch size twice from 4M to 16M during training.

---

> > ### Comment · Reviewer_YEQS · 2024-11-26
> > **Response to Authors**
> >
> > Thanks for your response! It addresses my concerns, and my reviews have been updated correspondingly.

---

### Official Review · Reviewer_iDTk · 2024-10-28

**Soundness:** 2
**Presentation:** 2
**Contribution:** 2
**Rating:** 3
**Confidence:** 4

**Summary:**

This paper presents a continually-pretrained multilingual LLM that covers 546 languages. The paper focuses on the data mix (MaLA corpus) and the evaluation.
The model is based on Llama2-7B and compared to other Llama derivatives or multilingually pretrained models. It claims comparatively strong performance on commonsense reasoning, machine translation and open-ended generation.
The data combines multiple existing sources and adds preprocessing steps.
The evaluation includes existing benchmarks and new intrinsic benchmarks and a new automated open-generation benchmark that is using BLEU as a metric for comparing generations.

**Strengths:**

- Solid effort in including/focusing the data mix on low-resource languages that were excluded in previous works. This is ambitious and showcases that more languages are possibly “in reach” for more language-inclusive language modeling.
- Good highlight on iso code normalization and writing system treatment, which is often underestimated/undervalued in multilingual modeling but essential.

**Weaknesses:**

1. No discussion of related work beyond listing models for comparison and data sources that are part of MaLA. It would be important to communicate how this work differs from past efforts beyond covering more languages in continual training. What are the main achievements that have been unlocked? Where is the scientific novelty?
2. Presentation: The result tables are overcrowded with text, and some of the results that are mentioned as main model strengths are hidden away in the appendix. It is hard for the reader to understand which numbers the claims of performance are grounded in. Furthermore, it is not clear what the main goal behind the model is: is it to be good at tail languages or beat other models in their supported languages or both?
3. Evaluation:
    - Metric: Two of the three main evaluation results are based on BLEU, although it is known that BLEU is (1) not equally suitable for all 500+ languages (see e.g. https://arxiv.org/pdf/2205.03983, Section 4.2) (2) is purely surface-based, e.g. underestimates paraphrases, neglects semantics, etc. The evaluation with self-BLEU is questionable - what does high diversity mean in the absence of a metric of accuracy?
    - Human: There is no human evaluation of the final model, nor the data pipeline, hence it is not clear how good the generation is especially in the newly covered tail languages.
      - The only benchmark that covers 500+ languages is intrinsic modeling on Glot500-c, which might be biased as the same corpus was used during pre-training, hence data quality issues won’t show in evaluation.
      - The new PolyWrite benchmark is machine-generated and machine-translated, and machine-filtered. How reliable is it? Is there Western-centric bias in the prompts, or are they equally representative for all languages covered?
    - Safety: No safety-related evaluation. It is known that data quality especially in low-resource languages is often lacking, caused by quality issues in the data pipeline, which in turn can introduce unwanted biases (https://direct.mit.edu/tacl/article/doi/10.1162/tacl_a_00447/109285)
    - Comparison with other models: It is not fair to compare models that do not officially support certain languages on these specific languages. This distorts averages across languages and makes it impossible to compare where wins are due to covering languages at all vs truly outperforming other models that also include these languages. I would recommend separating these: evaluating against other models on their languages and benchmarks, and then evaluating on the newly covered languages, with a focus on quality of generations (some basic checks can be run: is the generation in the correct language? Does it have expected length?).

  I understand that this represents a “best effort with limited means” - building reliable evaluation for 500+ languages is a huge enterprise by itself. However, the paper should at least prominently discuss these shortcomings and be more explicit about evaluation gaps and how to interpret the presented results, and which purpose the model can serve for advancing NLP in these languages. To be concrete, let’s say I want to use the model for Nogai - what can I expect?
Expanding the set of languages in training without having reliable evaluation metrics for each of them (or at least discussing or being transparent about their shortcomings and evaluation gaps) is not sufficient for responsibly claiming an advance in multilingual modeling. I do believe that the effort put into data preparation and training is worth publishing, but needs to be evaluated and presented in a much more rigorous manner.

**Questions:**

- What is the relation to the MaLA-500 model/work? What are key improvements over it?
- How was the data mix tuned? What was the process of setting the weights, based on intuition or prior experience or reports? Were any ablations run? This would be nice to describe in the Appendix (if no space in the main paper) to help others understand the deciding factors in setting up a successful data mix.
- Table 4:
  - Why is tower-instruct performing so poorly (especially since it’s built for translation)?
  - How would the model perform with fewer or more shots? Is the number of shots optimized for EMMA-500?
- Since whitespace cleaning only applies to whitespace-tokenized languages, are the other languages cleaner per se or need cleaning in other ways?
- How were existing datasets and their languages to be included chosen?
- Table 6: Are BLEU scores in this low range even reliable? Is the difference between 1- and 2-BLEU outputs significant? This should be supported by at least some evidence, e.g. significance tests or qualitative examples.
- How is self-BLEU evaluation influenced by sampling hyperparameters? Please report sampling parameters and discuss how they might affect the self-BLEU.

---

> ### Author Response · Authors · 2024-11-18
>
> **Q:** related work and data sources
> **A:** Thank you for your comments. The related work and data sources are provided in Appendix A1 and F.
>
> **Q:** what are the main achievements and the scientific novelty? And the main goal
> **A:** The main goal is to improve multilingual language models by creating a massive, diverse multilingual corpus (MaLA) and applying continual pre-training to enhance linguistic inclusivity and task performance. Our main achievements and scientific novelty are written in the introduction section.
>
> **Q:** what does high diversity mean?
> **A:** High diversity in evaluating open-ended generation refers to assessing a model's ability to produce varied and creative outputs across a wide range of tasks and topics.
>
> **Q:** No human evaluation
> **A:** Human evaluation, while valuable, has limitations such as subjectivity, inconsistency, and scalability. Evaluators may have biases, leading to variability in judgments across individuals and cultures. Additionally, evaluating multilingual and open-ended tasks requires diverse, linguistically skilled annotators, which can be expensive and time-consuming. These challenges make automated metrics a practical complement, even though they may lack nuanced judgment.
>
> **Q:** The only benchmark that covers 500+ languages is intrinsic modeling on Glot500-c
> **A:** This comment is incorrect. Taxi1500 covers 1,507 languages. Additionally, we evaluated two benchmarks with more than 100 languages, three benchmarks with more than 200 languages, and one with more than 300 languages. Together, these should provide our evaluation with good language coverage.
>
> **Q:** how reliable is machine-generated benchmark?
> **A:** Machine-generated benchmarks **have significant value** as they offer scalability, consistency, and objectivity, making it feasible to evaluate models across numerous tasks and languages efficiently. They are particularly useful for comparative analysis and identifying measurable improvements.
> There are **also limitations**: they often miss nuanced qualities like creativity, contextual appropriateness, and cultural sensitivity, especially in open-ended or multilingual tasks. Biases in the generation process or evaluation metrics can also lead to misleading results.
>
> **Q:** No safety-related evaluation
> **A:** We find safety-related evaluation a separate research topic. This work looks at CPT and releases a corpus and a multilingual base model rather than an instruction-tuned model. Also, to our knowledge, no standardized benchmark with many languages exists for benchmarking *base models* on safety. However, we are open to exploring it further and including these; we appreciate the suggestions for relevant benchmarks from reviewers.
>
>
> **Q:** unfair to compare models that do not officially support certain languages
> **A:** Our evaluation focuses on the performance of models across a broad range of languages, including those that may not be officially supported by some models. This is primarily because *not many other models can support this many languages*. Our comparison represents the *best effort* we can have when comparing to prior works; otherwise, no empirical comparison could be made. For "unseen" languages, many models might have a certain level of transfer, even if they are not specifically optimized for them. Nonetheless, this is the "upper bound" that users can get from these models. There is no widely accepted standard for which models should be used as the "fair" baseline for such comparisons. For example, [reviewer YEQS suggests comparing with ALMA and Bayling](https://openreview.net/forum?id=DPynq6bSHn&noteId=EeGCULon6N), but this may not be a fair comparison given the criteria mentioned.
>
> **Q:** What is the relation to the MaLA-500 model/work? What are the key improvements over it?
> **A:** The key improvements include:
> - Larger, More Diverse Corpus: our corpus includes 939 languages and 74 billion tokens.
> - Higher Token Count per Document: our corpus features longer documents (average of 90 tokens per sequence) compared to Glot500-c used by the MaLA-500 model (average of 19 tokens), providing better context and enabling the model to capture long-range dependencies.
> - More Diverse Data: We also include a wide range of text types, including code, books, and scientific papers.
> - Better Performance: Evaluated across more benchmarks, MaLA-500 shows improvements in multilingual tasks like translation, commonsense reasoning, and open-ended generation.
>
> **Q:** How was the data mix tuned?
> **A:** The tuning of the data mix is a separate research topic. Our paper focuses on providing a valuable resource through data processing and model training. While we did not conduct specific ablations on the data mix, we carefully curated the corpus to ensure diversity and balance across languages and text types. Further exploration of this topic could be beneficial for future work.

---

> > ### Comment · Reviewer_iDTk · 2024-11-18
> >
> > Thanks for the response! I missed the comparison of MaLA-trained LLaMa models and the Taxi dataset - thanks for pointing these out.
> >
> > While you have addressed a subset of my questions I would still like to hear your thoughts on:
> > 1) BLEU - why not use ChrF? It's arguably more inclusive and doesn't rely on tokenizers. See https://arxiv.org/pdf/2205.03983 Section 4.2.
> >
> > 2) I understand what diversity means - but when diversity is decoupled from accuracy - it can mean that the model is just generating "creative" gibberish.
> >
> > 3) Regarding human evaluation: I understand that it can be a resource-intensive enterprise, but there are trade-offs to be explored. For example, I would prefer to see a subset of languages/datasets evaluated where possible (and especially for those languages added that no other work covers). Human evaluation on the most basic (audit) level "is this text/generation written in language x"? does not require linguistic training. In fact, automatic metrics, when entirely relying on automatic pipelines (such as in the generation and evaluation of PolyWrite), lack more than just nuances (https://direct.mit.edu/tacl/article/doi/10.1162/tacl_a_00447/109285).
> >
> > 4) Since you include instruction data in the pretraining, and evaluate on other instruction benchmarks - why not evaluate on safety benchmarks? There are several benchmarks that cover at least a few languages, like XSafety (https://github.com/Jarviswang94/Multilingual_safety_benchmark) or Aya Redteaming (https://huggingface.co/datasets/CohereForAI/aya_redteaming?not-for-all-audiences=true).
> >
> > 4) For the models where this information is available, can you add which languages they officially support and separate the aggregated metrics for officially supported and unseen languages? It would be great to understand (a) in the languages that other models cover, how is EMMA-500 comparing to them? (there's probably a trade-off), (b) in the languages that other models don't cover, how is EMMA-500 comparing to crosslingual transfer (i.e. what is the benefit of this new corpus/model)?.
> > This would help to understand if the "curse of multilinguality" is present.
> >
> > 5) In your response regarding MaLa500, there must have been a confusion in the last point - you mean EMMA-500?

---

> > > ### Author Response · Authors · 2024-11-19
> > >
> > > **Q1:** BLEU - why not use ChrF?
> > > **A:** We used both BLEU and chrF++ (ChrF++ includes word bi-grams in addition to chrF) to evaluate machine translation (Table 4). We also release all the translated text so the public can quantitatively or qualitatively evaluate these.
> > >
> > > **Q2:** Regarding diversity
> > > **A:** While diversity must be balanced with accuracy to ensure outputs are meaningful, it is particularly important for creative tasks like those in PolyEval, which include storytelling, sci-fi, poetry, fantasy, adventure, and mystery. In these contexts, diversity reflects the model's ability to generate varied and engaging content rather than repetitive or formulaic responses. As automatic evaluation serves as an indicative reference, we also release all generated texts for open evaluation, allowing the community to assess the balance between creativity and coherence.
> > >
> > >
> > > **Q3:** Regarding human evaluation
> > > **A:** While resource constraints have limited our ability to perform human evaluation for this work, we recognize its importance in complementing automatic metrics. However, we must emphasize the limitations and unsustainability of human evaluation, particularly in a massively multilingual setting. Even evaluating a subset of languages or datasets, most likely low-resource languages, remains prohibitively expensive. For basic (audit-level) evaluation, such as "is this text/generation written in language x?", we advocate using automatic methods, such as language identification tools, to address these challenges more efficiently.
> > >
> > > **Q4:** Safety evaluation
> > > **A:** Thank you for pointing out these benchmarks. While our work focuses on multilingual data expansion and model evaluation across a broad range of languages and tasks, safety evaluation was not a primary focus as the model is not directly developed for end users. We recognize the importance of safety benchmarks like XSafety and Aya Redteaming. As you also acknowledge that they only cover a few languages, it might not fit the evaluation in a massively multilingual setting best. We will discuss incorporating them in future evaluation runs to assess our model's safety and robustness.
> > >
> > > **Q5:** For the models where this information is available, can you add which languages they officially support and separate the aggregated metrics for officially supported and unseen languages?
> > > **A:** We can separate metrics for models that explicitly mention supported languages, but this is not possible for models like LLaMA and GEMMA 2, which do not provide such details. We report per-language performance for some benchmarks in the Appendix, allowing readers to categorize the performance into different groups of languages as they wish. To avoid making the Appendix unnecessarily long, we will also share all results, including models' generation, via GitHub, enabling the community to conduct further analyses with their preferred metrics.
> > >
> > > **Q:** In your response regarding MaLa500, there must have been a confusion in the last point - you mean EMMA-500?
> > > **A:** Apologies for the confusion. It was a typo. The correct statement is, "... EMMA-500 shows improvements in multilingual tasks ...".
> > >
> > >
> > > Overall, we thank you for your feedback. We wish to emphasize the core contributions of this paper: the corpus, the model, and the technical process. Many of the points raised, while important, could be considered "nice-to-haves" but are not the primary focus of our work.
> > >
> > > To summarize the key contributions of this paper: The **MaLA corpus's scale, diversity, and focus on low-resource languages** represent a significant advancement over previous efforts. By integrating a variety of text types—such as code, scientific papers, and instructions—this work provides a more comprehensive foundation for multilingual pre-training. These improvements lead to a better multilingual LLM (EMMA-500) that **boosts multilingual capabilities** evaluated on 10 tasks and 17 benchmarks. The **scale and impact** of this work make it a valuable contribution to the field of multilingual NLP, especially for low-resource languages.

---

> > > > ### Comment · Reviewer_iDTk · 2024-11-21
> > > >
> > > > Thanks for your response! I appreciate the release of the outputs and full results, it will be good to have community feedback.
> > > >
> > > > Q1 Wouldn't it also make sense to replace self-BLEU with self-ChrF++? It would also give you a larger spread of values that are perhaps a bit more distinctive and fair towards non-whitespace-tokenized languages.
> > > >
> > > > Q3 The problem with automatic tools (especially multilingual ones) is that they're usually lower-performing and less reliable in low-resource languages than for higher-resource languages, even for the most "basic" language id (https://arxiv.org/abs/2010.14571). If e.g. language id was imprecise in the data collection pipeline, using the same lang id tool for evaluation won't allow to spot any problems, even if humans were easily be able to do so.

---

> > > > > ### Author Response · Authors · 2024-11-22
> > > > >
> > > > > **Q1:** Wouldn't it also make sense to replace self-BLEU with self-ChrF++?
> > > > > **A:** You raise a good point. The original paper proposed self-BLEU, which naturally guided our choice to use it for consistency with prior works. However, replacing self-BLEU with self-ChrF++ could provide a good alternative. We will also use self-ChrF++ in the revision.
> > > > >
> > > > > **Q3**: The problem with automatic tools (especially multilingual ones)
> > > > > **A:** You raise an important concern about the limitations of automatic tools, as highlighted in the referenced work. However, we do not fully agree that humans would easily overcome this challenge. Human evaluation might also face significant difficulties when evaluating very low-resource languages; for example, the limitations mentioned in [our previous replies](https://openreview.net/forum?id=DPynq6bSHn&noteId=qepfFH8F2c) like subjectivity, inconsistency, and scalability. Also, finding qualified evaluators for low-resource languages is often impractical and costly. Automatic tools, while not perfect, offer a scalable and consistent method for data collection and evaluation. That said, **addressing the limitations of language identification tools is a broader research topic, and determining whether automatic or human evaluation is better is not the primary focus of this paper**.

---

> ### Author Response · Authors · 2024-11-18
>
> **Q:** Why is tower-instruct performing so poorly (especially since it’s built for translation)?
> **A:** According to its technical report, Tower is heavily optimized (through CPT and SFT) for only ~10 languages centred on translation-related data. Its language capabilities might not generalize well across low-resource languages or specific language pairs. This is essentially our model's advantage over Tower.
>
> **Q:** How would the model perform with fewer or more shots? Is the number of shots optimized for EMMA-500?
> **A:** The choice of 3-shot was based on results from LLaMA-2 models, where it showed reasonable performance on high-resource languages. The number of shots was not optimized through a grid search for EMMA-500. While fewer or more shots could impact performance, further experimentation would be needed to determine the optimal number for different tasks and languages. However, this is not the primary focus of our work, which is intended as a resource paper. The 3-shot choice already effectively showcases the model's performance across tasks, providing a good balance between data efficiency and task completion.
>
>
> **Q:** Since whitespace cleaning only applies to whitespace-tokenized languages, are the other languages cleaner per se or need cleaning in other ways?
> **A:** The whitespace cleaning process primarily applies to whitespace-tokenized languages, helping standardize the text for better processing. For languages that do not rely on whitespace tokenization (such as Chinese, Japanese, or Korean), the text may not require this specific cleaning but might need other preprocessing steps, such as character normalization or segmentation.
>
> **Q:** How were existing datasets and their languages to be included chosen?
> **A:** The selection of existing datasets and their languages for inclusion was based on several factors:
> - Language Coverage: We prioritized datasets that cover a wide range of languages, with a focus on including both high-resource and low-resource languages to ensure broad multilingual representation.
> - Data Quality: Datasets with high-quality, diverse content, such as books, scientific papers, and code, were selected to ensure the corpus has varied text types, improving model generalization.
> - Task Relevance: We considered datasets that aligned well with our goals for continual pre-training and multilingual adaptation across tasks like translation, classification, and reasoning.
> - Availability: Datasets that were publicly available and accessible were preferred, ensuring transparency and reproducibility.
>
> **Q:** Table 6: Are BLEU scores in this low range even reliable?
> **A:** BLEU scores in the low range can provide a general indication of performance, but they are not a perfect evaluation method, especially for open-ended generation. While BLEU helps assess text similarity, it may not capture the full quality or creativity of the generated responses. There is no single evaluation metric that fully reflects the complexity of open-ended generation. In fact, this remains a challenging research direction, especially in a multilingual setting. We have also discussed this from lines 488 to 494.
>
> **Q:** How is self-BLEU evaluation influenced by sampling hyperparameters?
> **A:** Self-BLEU evaluation can be influenced by sampling hyperparameters, but we do not perform a grid search on them. We use the same set of hyperparameters for all models. The parameters used are:
>
> - Temperature: 0.6
> - Top-p: 0.9
> - Max tokens: 512

---

### Official Review · Reviewer_Bad1 · 2024-11-01

**Soundness:** 3
**Presentation:** 2
**Contribution:** 3
**Rating:** 5
**Confidence:** 4

**Summary:**

EMMA-500 outlines work around three technical contributions:
* EMMA-500 -> A multilingual model trained on 546 languages
* MaLA Corpus -> A multilingual corpus spanning 939 languages (A subset of which that had >100k tokens were used in the EMMA-500 training)
* PolyWrite -> Open Ended generation benchmark

**Strengths:**

* Always great to see work that tackles the challenges in the long tail of languages.
* In-depth information for several preprocessing steps. This is really useful and important to allow future work to build on.
* Several evaluations performed on a plethora of datasets.

**Weaknesses:**

I would like to list the following weakness fully ensuring the authors that I am not unreasonable and am completely open to increasing my score if these are addressed/answered satisfactorily.

* The contributions section could be rewritten as bullet points and bold headings to emphasize the exact contributions. This was confusing for a first-time reader given that MaLA-500(I assume this is related to previous work) also exists.
* Evaluations lack benchmarking the recent Aya-23 model. They are even omitted when using the eval sets from the Aya paper. Surprisingly they are only benchmarked in the code evals without any further explanation.
* The categorization of “high” vs “mid” vs “low” resources is created by measuring token count in this specific training dataset, however there already exists widely accepted standards/taxonomies(like https://microsoft.github.io/linguisticdiversity/ – Joshi et al). No reference or explanation is provided as to why the authors deviated from this standard.
* For n-shot evals ( like 3-shot on FLORES), it would be useful to also see 0-shot evals to understand how big a role ICL plays in the performance gains.
* I would edit the tables to be easier to read( for ex: bold the important numbers; say lower/higher is better; add arrows)

**Questions:**

* Could you please explain why the Aya model was omitted from some evals despite being quite a popular model for multilingual benchmarking?

---

> ### Author Response · Authors · 2024-11-18
>
> **Q:** Comparing with Aya-23
> **A:** The Aya evaluation suite originates from Aya-101. We omitted a direct comparison with Aya-101 because our focus is on decoder-only models. Aya-23 is relatively new; we have compared it on code tasks but will include a more comprehensive comparison in the next revision. The current version already compares EMMA-500 with state-of-the-art (SOTA) models, such as Qwen and Llama 3/3.1, which outperform Aya-23. Therefore, the current comparison is sufficient to demonstrate the effectiveness of the EMMA-500 model.
>
> **Q:**  Joshi et al's categorization of “high” vs “mid” vs “low” resources
> **A:** The complete list of languages in Joshi et al.'s categorization is not publicly available to us, unfortunately.
>
> **Q:** n-shot / 0-shot evals
> **A:** The choice of 3-shot evaluation is based on the results of Llama 2 models, which achieve reasonably good scores on high-resource languages with this setup. We did not perform a comprehensive grid search. We can, however, check 0-shot results. However, we are concerned about reviewer iDTk's question regarding the reliability of BLEU scores in the low range, which could affect evaluations, particularly for low-resource languages.
>
> **Q:** The contributions section could be rewritten as bullet points and suggestion to edit table
> **A:** Thank you for the suggestion. We agree that rewriting the contributions section as bullet points would improve clarity. Additionally, we will review the table and incorporate edits to enhance its readability and presentation.

---

> > ### Comment · Reviewer_iDTk · 2024-11-21
> >
> > Joshi's full taxonomy is linked on the website that reviewer Bad1 shared: https://microsoft.github.io/linguisticdiversity/assets/lang2tax.txt - unfortunately without ISO codes.

---

> > > ### Author Response · Authors · 2024-11-22
> > >
> > > Thank you for sharing the link. Indeed, the lack of ISO codes (both language codes and writing systems) makes it challenging to directly align it with our dataset.

---

> ### Comment · Reviewer_Bad1 · 2024-11-25
>
> These are my comments regarding the author's rebuttal.
>
> **Regarding Aya-23** -> I think calling it "relatively new" with it being a multilingual-first decoder model coming out in May, while Llama 3.1 coming out much later (in July?) was instead used for evals still seems odd. If Aya-23 was completely omitted, I would understand it being intentionally left out. Again, using it for code generation blindly without including even a single other eval with the model doesn't bode well. Also, doesn't Llama 3.1 only support English, German, French, Italian, Portuguese, Hindi, Spanish, Thai? If there really is a specific reason behind this omittal, I am happy to engage.
>
> **Regarding Joshi taxonomy** -> I am disappointed that reviewer *iDTk* had to point out that **I already pasted the link** to the taxonomy. Albeit it doesn't contain the ISO codes, but saying its "not publicly available to us" wasn't particularly encouraging as a rebuttal comment.

---

> > ### Author Response · Authors · 2024-11-25
> >
> > **Q:** Regarding Aya-23
> > **A:** Thank you for pointing this out. There were no specific reasons for the limited use of Aya-23 in our evaluations. The models compared in coding tasks and other benchmarks differed because different authors contributed to the evaluation. For multilingual tasks, baselines such as MaLA-500 and LlamaX, which are continued pre-trained on LLaMA 2, were deemed more critical. While Aya-23 was not the primary focus of our evaluations, we can already find some results for comparison. Below is a summary of performance on three specific benchmarks:
> >
> > | **Benchmark**   | **EMMA-500** | **Aya-23** |
> > |------------------|--------------|------------|
> > | **XCOPA**       | 63.11        | 59.8       |
> > | **XStoryCloze** | 66.38        | 62.3       |
> > | **XWinograd**   | 72.80        | 80.7       |
> >
> > As seen, EMMA-500 outperforms Aya-23 on XCOPA and XStoryCloze, but lags behind on XWinograd. These results reflect the relative strengths of the models across tasks.
> > The primary focus of this work is **not on comparing with Aya-23 specifically but rather on advancing the multilingual NLP landscape with the MaLA corpus**, a continued pre-trained model based on LLaMA 2, and evaluations that demonstrate improved performance in many languages. Continue-pretrained models like MaLA-500 and LlamaX, and SOTA models like Qwen 2 and Llama 3//3.1 were prioritized as baselines.
> >
> > We acknowledge the value of including Aya-23 in a wider range of benchmarks and will consider incorporating it in future updates to provide a more comprehensive evaluation. However, the current version already compares EMMA-500 with state-of-the-art (SOTA) models, such as Qwen and Llama 3/3.1, and advanced continue-pretrained models like MaLA-500 and LlamaX. Therefore, the current comparison is sufficient to demonstrate the effectiveness of the EMMA-500 model.
> >
> > We encourage the reviewer to focus on **the key contributions of this paper: the MaLA corpus, the data mixing, the continued pre-trained model based on LLaMA 2, and evaluations showcasing significant improvements in multilingual tasks.**
> >
> > Regarding Llama 3.1’s language support, its official documentation mentions the languages you listed, but our evaluation sought to assess its generalization to a broader set of languages, even those not explicitly supported.
> >
> > **Q:** Regarding Joshi taxonomy
> > **A:** Thank you for your comment. Reviewer [iDTk](https://openreview.net/forum?id=DPynq6bSHn&noteId=MTzjES3Z5D) directly shared a different link to a text file that contains the language list and explicitly mentioned the lack of ISO codes. Unfortunately, we were unable to locate the text file from the link you shared earlier, which seems to have led to a misunderstanding.
> > Additionally, even if we had the file, it seems some languages in our covered set, such as kir_Cyrl (Kyrgyz), are not included in Joshi's list.

---

> > > ### Comment · Reviewer_iDTk · 2024-11-26
> > >
> > > Just to close the Joshi discussion:
> > > 1) The taxonomy is linked on the website that the reviewer shared.
> > > 2) The file I shared is found by clicking on that website on "taxonomy".
> > > 3) Kyrgyz is on the list, it's listed as "Kirghiz". Another oversight? I am doubting the diligence that went into this work. It is common for low-resource languages to be used with various names, language mapping is not fun but part of data preprocessing.
> > > 4) Iso codes missing is unfortunate, but other works, such as Aya, have overcome this hurdle too. It requires manual resolution of iso codes, but it's feasible if one cares about grounding ones language selection in this taxonomy.

---

> > > > ### Author Response · Authors · 2024-11-26
> > > >
> > > > Thank you for clarifying. While we acknowledge the importance of language taxonomies like Joshi’s, we choose not to rely on it for this work for several reasons:
> > > >
> > > > 1. **Focus on Practical Impact:** Our work emphasizes advancing multilingual NLP through **comprehensive corpus creation and improved model performance**, rather than adhering to a specific taxonomy. We focused on covering as many languages as possible, particularly underrepresented ones, using ISO codes for consistency in data processing.
> > > >
> > > > 2. **Feasibility vs. Contribution:** Manually aligning our language set with Joshi’s taxonomy, especially given the absence of ISO codes, is a labor-intensive task with limited relevance to the core contributions of this paper. While it is feasible, we believe our efforts are better directed toward demonstrating the utility of the MaLA corpus and models through tangible multilingual improvements.
> > > >
> > > > We appreciate your observation about language mapping challenges and agree that it’s an integral part of multilingual data preprocessing. However, **grounding our language selection in Joshi’s taxonomy is not a priority or requirement for this work**, as our contributions lie in practical advancements rather than taxonomy alignment.

---

### Meta-Review · Area_Chair_gU9K · 2024-12-19

**Metareview:**

The submission introduces EMMA-500, a multilingual language model trained across 546 languages via continual pre-training on the MaLA corpus, a multilingual dataset compiled from various existing sources. The paper highlight improvements in multilingual performance, particularly for low-resource languages, and propose a new open-ended generation benchmark, PolyWrite, alongside evaluations on multilingual tasks. The claimed contribution lies in improving Llama 2’s multilingual capacity, especially for underrepresented languages.

However, the submission falls short of significant novelty. While the compilation of the MaLA corpus and its integration into a training pipeline is valuable, the data cleaning and methodology lack rigorous analysis or clear advancements over existing multilingual approaches. Furthermore, the evaluation framework is incomplete and raises concerns: reliance on automatic metrics like BLEU and self-BLEU, known to be unreliable across diverse languages, is not sufficiently mitigated with human evaluations. PolyWrite, though a novel benchmark, lacks thorough validation, with unclear reliability and concerns about biases inherent in machine-generated prompts. Additionally, comparisons with state-of-the-art multilingual models, including Aya-23, are insufficient, and the justification for their exclusion in certain evaluations is unconvincing. The taxonomy for language categorization deviates from established standards (e.g., Joshi et al.), and the rebuttal failed to adequately address concerns regarding this choice.

Overall, despite the considerable effort to create a resource and improve multilingual inclusivity, the work does not present a substantial scientific contribution beyond aggregating and cleaning datasets for continual training. Given these weaknesses, I recommend rejecting the paper.

**Additional Comments On Reviewer Discussion:**

During the rebuttal period, the discussion centered around the novelty, evaluation methodology, baseline comparisons, and taxonomy alignment in the submission.

Novelty and Contributions: Multiple reviewers, including iDTk and YEQS, questioned the novelty of the work, emphasizing that merging and cleaning existing datasets lacks significant scientific innovation. The authors defended the contributions by highlighting the scale and diversity of the MaLA corpus and the resulting EMMA-500 model’s improved multilingual performance.

Evaluation Methodology: Reviewers iDTk and Bad1 raised concerns about the heavy reliance on BLEU and self-BLEU, metrics known to be unreliable for low-resource languages and open-ended generation tasks. The reviewers suggested using ChrF++ and incorporating human evaluation to validate the results. The authors agreed to add ChrF++ but defended the omission of human evaluations due to scalability issues. This response did not fully address concerns about the reliability of automatic metrics, particularly for low-resource languages.

Baseline Comparisons: Reviewers Bad1 and YEQS highlighted the omission of key baselines, such as Aya-23, which weakened the empirical comparisons. While the authors provided partial explanations for the exclusion of Aya-23 and argued that ALMA and Bayling focus on translation, the reviewers remained unconvinced. Bad1 specifically noted inconsistencies in the inclusion of models across evaluations.

Language Taxonomy and Diligence:Reviewer Bad1 criticized the authors for not aligning their work with Joshi et al.’s established taxonomy, questioning the thoroughness of the data preprocessing pipeline. While the authors cited difficulties in mapping languages without ISO codes, iDTk demonstrated that this issue was addressable, further questioning the authors’ diligence and transparency.

Safety and Reliability: iDTk and Bad1 pointed out the lack of safety-related evaluations and concerns about biases in machine-generated benchmarks like PolyWrite. The authors acknowledged the importance of safety benchmarks but argued it was outside their current scope, proposing to address it in future work. This was seen as a gap, given the potential risks in multilingual data pipelines.

In weighing these points, the lack of novelty and insufficient evaluation emerged as critical weaknesses. The authors’ rebuttals clarified some points but failed to address key concerns around incomplete baselines, reliance on questionable metrics, and the absence of rigorous validation. While the effort to expand multilingual data coverage is commendable, the reviewers’ concerns regarding scientific contribution, methodology rigor, and presentation remain unresolved. Consequently, these shortcomings led to the final decision to recommend rejecting the paper.

---

### Decision · Program_Chairs · 2025-01-22

Reject